# Proxy surrogate reconstructions for Europe and the estimation of their uncertainties

Oliver Bothe[1] and Eduardo Zorita[1]

[1]Helmholtz Zentrum Geesthacht, Institute of Coastal Research, 21502 Geesthacht, Germany

**Correspondence:** Oliver Bothe (ol.bothe@gmail.com)

**Abstract.** Combining proxy information and climate model simulations reconciles these sources of information about past climates. This, in turn, strengthens our understanding of past climatic changes. The analogue or proxy surrogate reconstruction method is a computationally cheap data assimilation approach, which searches in a pool of simulated climate states the best fit to proxy data. We use the approach to reconstruct European summer mean temperature from the 13th century until present using the Euro 2k set of proxy-records and a pool of global climate simulation output fields. Our focus is on quantifying the uncertainty of the reconstruction, because previous applications of the analogue method rarely provided uncertainty ranges. We show several ways of estimating reconstruction uncertainty for the analogue method, which take into account the non-climate part of the variability in each proxy record.

In general, our reconstruction agrees well at multi-decadal timescales with the Euro 2k reconstruction, which was conducted with two different statistical methods and no information from model simulations. In both methodological approaches, the decades around year 1600 CE were the coldest. However, the approaches disagree on the warmest preindustrial periods. The reconstructions from the analogue method also represent the local variations of the observed proxies.

The diverse uncertainty estimates obtained from our analogue approaches can be locally larger or smaller than the estimates from the Euro 2k effort. Local uncertainties of the temperature reconstructions tend to be large in areas that are poorly covered by the proxy records. Uncertainties highlight the ambiguity of field based reconstructions constrained by a limited set of proxies.

## 1 Introduction

There have been numerous efforts to reconstruct regional to global surface temperature for the last 500 to 2000 years. Many of the statistical reconstruction methods essentially assume a linear relationship between proxy information and temperature data. Here, we apply a non-linear method, the analogue method, to reconstruct the mean European summer temperature over the past 750 years in annual resolution. Our main goal is to provide a perspective on estimating uncertainties for reconstructions by analogue, which only few previous applications quantified. Our approach relies on a collection of dendroclimatological proxy-records and the output of paleoclimate simulations.

The core of the analogue method is searching for similar spatial patterns in simulated temperature data compared to the proxy records. That is, we search for simulated analogues of the climate anomalies indicated by the set of proxies at each available

date. The method originated during the Second World War when the US Air Force catalogued weather situations of previous decades as a means of long range weather forecasting. In this approach forecasters obtain forecasts by analogy between current observations and a past set of weather patterns (Namias, 1948). Lorenz (1969) was the first to mention the method in the wider academic literature. The analogue method found subsequent applications in downscaling and upscaling of climate information

(e.g., Zorita and von Storch, 1999; Schenk and Zorita, 2012). Modern analogue techniques of paleoecology follow a similar idea (e.g., Graumlich, 1993; Jackson and Williams, 2004).

The approach allows to reconcile the spatially sparse information from environmental and documentary proxy data with spatially complete and dynamically consistent information from observational data or long climate simulations in the sense of data assimilation (Graham et al., 2007; Trouet et al., 2009; Guiot et al., 2010; Franke et al., 2010; Luterbacher et al.,

2010; Schenk and Zorita, 2012; Gómez-Navarro et al., 2015b; Diaz et al., 2016; Gómez-Navarro et al., 2017; Jensen et al., 2018; Talento et al., 2019; Neukom et al., 2019; Wahl et al., 2019). It can provide an initial dynamic understanding of past climate variability. However, it is less sophisticated than full data assimilation procedures (compare, e.g. Tardif et al., 2019, and discussions in Gómez-Navarro et al., 2017). Graham et al. (2007) call reconstructions by analogue "Proxy Surrogate Reconstructions" in an early paleoclimatological application. Later studies use the approach for climate index and climate field

reconstructions (e.g., Franke et al., 2010; Trouet et al., 2009; Gómez-Navarro et al., 2015b, 2017; Jensen et al., 2018; Talento et al., 2019; Neukom et al., 2019).

The analogue method is generally found to perform well, e.g., for area averaged indices and also at the locations of the used predictors (compare, e.g., Franke et al., 2010). However, reducing the number of predictors prominently worsens the skill at remote locations, and reconstruction skill accumulates at the predictor locations (Franke et al., 2010; Gómez-Navarro et al.,

2015b). Annan and Hargreaves (2012) find a trade-off between accuracy and reliability of reconstructions dependent on quality and quantity of the available proxy-records. They test a particle-filter method. As simple analogue searches and particle filter methods share common assumptions, this also applies for analogue search reconstructions. Similarly, it is well established that applications of the analogue method have to deal with a trade-off between accuracy and variability (Gómez-Navarro et al., 2015b, 2017). Franke et al. (2010), Gómez-Navarro et al. (2015b), and Talento et al. (2019) discuss the influence of considering

more than one analogue to produce a composite reconstruction while Graham et al. (2007) and Trouet et al. (2009) consider only the single best analogue based on specific criteria. In any application, one has to consider the potential biases in the simulation data.

Previous analogue search reconstructions usually do not consider the uncertainty in the predictor data, and studies rarely provide an uncertainty estimate for the final reconstruction. This precludes to some extent a realistic evaluation of predictors

or reconstructions. Exceptions are the studies by Jensen et al. (2018) and Neukom et al. (2019). The former use age-uncertain proxies and obtain an uncertainty estimate of their reconstruction for the Marine Isotope Stage 3 through shifting the dates of individual proxies (Jensen et al., 2018). The latter use a subsampling approach to provide an ensemble of reconstructions for the Common Era of the last 2000 years (Neukom et al., 2019). Their study interprets the spread as uncertainty of the final reconstruction.

Here, we propose alternative means to estimate the uncertainty of analogue search reconstructions based on the calibration correlation of the proxy predictors with an appropriate observational data set. Our approach to estimating uncertainty ranges reduces the possibility of producing time series of reconstructed climate. On the other hand, it allows to provide alternative reconstructions that are compatible with the sparse information from the proxy records. The procedure further acknowledges the possibility that the analogue pool does not cover certain points in the predictor space. Our proposed uncertainty estimates
originate in the uncertainty of the individual proxies, whereas Neukom et al. (2019) quantify the variations in reconstruction results by using less information than available.

Recent continental proxy-based reconstructions (PAGES 2k Consortium, 2013) and the underlying proxy predictors are potential test cases and allow to assess the analogue method against more common reconstruction procedures. (Dis)agreement between the analogue reconstructions and previously published estimates helps to reevaluate our confidence in our understand-
ing of past climate changes. For the present purpose, we choose the European reconstruction from PAGES 2k Consortium (2013) as a single test case. See also the work by Luterbacher et al. (2016), who discuss the methods and the proxy-selection in more detail. Luterbacher et al. (2016) rigorously select proxy records of high quality for their reconstruction.

In the following, first, we introduce our approach to the analogue search uncertainty as well as the used proxy and model data. Then, we discuss the results for three different approaches to an analogue reconstruction. These are (i) using a single best
analogue, (ii) using a fixed number of good analogues, and (iii) considering all analogues complying with the proxies within a fixed level of uncertainty. We also consider estimates from an ensemble following the subsampling approach of Neukom et al. (2019). We compare the resulting uncertainties among each other, and we shortly compare reconstructions to records based on station data.

## 2    Methods & Data

## 2.1    Methods

### 2.1.1    Analogue Search Reconstructions

The paradigm that past analogues may provide information for anthropogenic climate changes is pervasive in climate science (Dahl-Jensen et al., 2015; Schmidt, 2010; Schmidt et al., 2014) but the origin of the analogue method lies in weather forecasting (see, e.g., Lorenz, 1969). Zorita and von Storch (1999) show the method's value for downscaling while others provide evidence
for its ability to upscale local information (e.g., Schenk and Zorita, 2012; Luterbacher et al., 2010; Franke et al., 2010).

Here, we obtain annually resolved large-scale fields of seasonal mean summer (June, July, August, JJA) temperature based on a pool of relevant candidate fields and a set of local data indices as predictors for the period 1260 to 2003 of the Common Era (CE). The reconstruction domain is Europe from -10E to 40E and from 35N to 70N (Figure 1). The predictors are proxy reconstructions in temperature units from PAGES 2k Consortium (2013) and the pool of candidate fields consists of more than
9000 summer temperature fields from simulations with an earth system model (Jungclaus et al., 2010).

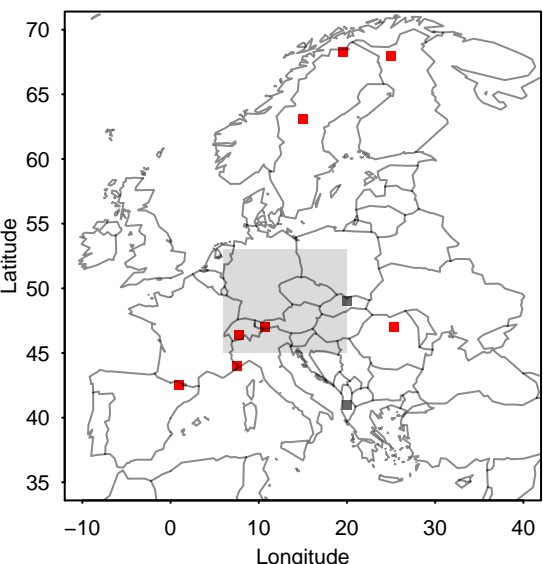

**Figure 1.** Reconstruction domain and locations of the included proxies. Red squares show the proxies included in our search, grey squares show the locations from the original Euro 2k setup, which we exclude. The grey shaded box shows the original domain of Dobrovolný et al. (2010).

The approach of an analogue search is usually that, for each set of predictors, i.e. each point in time, one ranks all potential analogues according to a criterion of similarity to the target proxy pattern. The criterion is traditionally the Euclidean distance and only the single pool-member with the smallest Euclidean (e.g., Franke et al., 2010) or a low number of so defined best analogues is considered. It is possible to weight the found analogues, e.g., according to their distance. This can provide more reliable posterior distributions about the climate state.

The approach presented here differs from previous applications in some important aspects. While we also show a single best-analogue reconstruction and a reconstruction based on a fixed number of analogues, we add a reconstruction that explicitly considers the uncertainty of the proxy records in the selection of the analogue fields.

The next subsection provides details on our three different approaches. In short: first, the single best reconstruction is the common application and our uncertainty estimates derive from the local correlation between gridded observation data and the local proxy series. Second, a further common approach is to use a fixed number of analogues. As we want to consider the uncertainty of the local predictors, we identify for a given uncertainty level of the proxies the smallest number of valid analogues for any date in our period of interest and then provide a reconstruction for each date using this minimum number of analogues. Finally, we fix the value of the uncertainty level around the predictors and consider all valid analogues within this uncertainty level.

We consider predictors and analogues normalized by their local standard deviation to conserve the interfield relations. The final reconstructions are rescaled by a chosen standard deviation, which is, here, usually the local full period standard deviation of one of the simulations.

### 2.1.2 Assumptions on uncertainty

Empirical reconstructions of past environmental conditions rely on proxy data, which may be documentary notations but more often are measurements of biological, geological, or chemical properties of the environment. Such proxy representations of the past conditions are naturally uncertain. The most obvious source of uncertainty is that the archives recorded signals from more than one climate or environmental variable (e.g. temperature and precipitation; compare Evans et al., 2013; Tolwinski-Ward et al., 2013; Evans et al., 2014; Tolwinski-Ward et al., 2015).

In the following, we describe our thinking on the uncertainty of an analogue reconstruction. We first provide general derivations before describing the three reconstruction approaches (i) best analogue, (ii) fixed number of analogues, and (iii) fixed uncertainty level. Our derivation of the uncertainty estimates relies on a number of assumptions, which we detail in the next paragraphs. Table 1 lists all mathematical expressions used in the following.

### Derivation of the uncertainty estimates

Correlations provide a simple measure of the relation between proxy-observations and the climatic environment over a period when reliable (instrumental) observations of the climatic variability exist. We assume we can derive the uncertainty of how well a local proxy record represents the local climate from the correlation coefficients. We denote this uncertainty hereafter as proxy uncertainty. We use correlations between the proxy records and the observational gridded CRU-data (Harris et al., 2014, Version CRU TS 3.10). Table 2 in section 2.2 lists the used proxies and their correlations to the observational data. These listed correlations enter our considerations on uncertainty.

In our present approach, we consider normalized proxy data. That is the variance of an individual proxy $i$ is $Var_i = 1$. We also consider normalized simulated records, and their local variance then also is $Var_{sim} = 1$. Our goal is to derive a simple criterion for the similarity between proxy patterns and simulated (analogue) patterns that takes into account the inherent uncertainty in the proxy records.

Assuming one can interpret the squared correlation coefficient ($R^2$) as explained variance, one can profit from the equivalence

$$R^2 = 1 - MSE_{res}/MSE_{tot} = 1 - Var_{res}/Var_{tot}$$

if we take the considered mean squared errors (MSE) as unbiased. The subscripts are *res* for residual and *tot* for total.

We can take the total variance $Var_{tot}$ to be equal to the variance of the sum of a signal (subscript *sig*) and the residual noise. If we assume these are uncorrelated, we obtain

$$1 - R^2 = Var_{noi}/(Var_{sig} + Var_{noi})$$

We replaced the residual variance by the noise variance (subscript *noi*) and reorganised the equation.

Because we consider normalized data, the total variance becomes one, $Var_{tot} = 1$. For a simulated climate record in a grid-cell of a climate model, there is no uncertainty and, then, it is indeed $Var_{tot} = Var_{sig} = 1$, i.e. the total variance is pure signal.

**Table 1.** List of mathematical expressions.

| Expression | Description |
| --- | --- |
| $r$ | Correlation coefficient |
| $R^2$ | Squared correlation coefficient |
| $MSE_{res}$ | Residual Mean squared error |
| $MSE_{tot}$ | Total Mean squared error |
| $Var_{res}$ | Residual Variance |
| $Var_{tot}$ | Total Variance |
| $Var_{sig}$ | Variance of the signal |
| $Var_{noi}$ | Variance of the noise |
| $Var_{noi_i}$ | Variance of the noise of an individual record |
| $Var_{sim}$ | Variance of a simulation record |
| $Var_i$ | Variance of an indidividual time series |
| $SD$ | Standard deviation |
| $SD_{noi}$ | Standard deviation of the noise |

For the case of a normalized proxy we take $Var_{tot} = 1 = Var_{sig} + Var_{noi}$ and thus

$$1 - R^2 = Var_{noi}$$

This is an expression for the noise variance of one local proxy record.

We want to use the local estimates of the proxy noise to formulate a criterion for finding analogues in simulated field records from climate simulations. Because we use simulated records with unit variance, we can consider the following as a noise standard deviation

$$SD_{noi} = \sqrt{1 - R^2}$$

Based on these assumptions, there are a number of possible ways to obtain uncertainty estimates for a reconstruction by analogue, which we describe next.

**Different reconstructions and uncertainty estimates**

First, we consider the case of a reconstruction from the single best analogue. We use the normalized data for this reconstruction.

For this case, we assume that we can obtain one standard deviation uncertainties as the square root of the sum over the individual proxy noise variances ($Var_{noi_i}$) divided by the number $N$ of proxies: $\sqrt{\sum_1^N (1 - Var_{noi})/N}$. These are only an approximation of the uncertainty. If we want to plot the time-series in temperature units, we have to rescale these estimates. We do this simply by multiplying the noise variances in the square root by the grid-point variance from a selected simulation. Our

visualisations for the single best analogue reconstruction add an alternative uncertainty envelope. This is given by the mean squared error between the proxy-values and the best-analogue values at the closest grid-point.

From our point of view, the real benefit of our derivation of uncertainty is to use only analogues which comply with a certain tolerance criterion. That is, a second way towards an uncertainty estimate assumes that we can obtain a similarity criterion between proxy data and simulation pool by considering the noise standard deviation for an individual proxy as local noise tolerance threshold. A candidate field has to comply with all local thresholds to be considered a valid analogue. We then can limit our analogue search to only those analogues within a certain tolerance range at each location, i.e. within plus and minus one, two, or three $SD_{noi}$ around the proxy value.

In the following we only consider analogues within traditional 90%, 95%, 99% and 99.9% intervals. We consider two cases: (A) we use a fixed number of analogues, and (B) we use a fixed noise level $SD_{noi}$. For the fixed number approach, we ad hoc require that there are at least ten valid analogues for all years.

For a defined noise tolerance criterion, there may be at best a few locally tolerable analogues for a certain date. For example, if we consider a criterion of one $SD_{noi}$, that is a ~68%-interval, this criterion is so strict that we do not find any tolerable analogues for 35 years in our period of interest. Similarly ~$1.64\,SD_{noi}$ (90%) and ~$1.96\,SD_{noi}$ (95%) criteria still imply that we find less than ten analogues for one year (2003 CE).

However, we want to provide a reconstruction at each date in the period 1260 to 2003 CE and want to consider a fixed number of analogues. We find that among the tested levels, a tolerance criterion of $2.57\,SD_{noi}$, i.e. a 99% interval, is the smallest noise level that provides more than 10 analogues for every year in the full period. The minimal number of analogues is 39 for this criterion if we include the year 2003. It increases to 156 excluding the year 2003. We do not test additional noise levels between ~$1.96\,SD_{noi}$ and $2.57\,SD_{noi}$ as we further, ad hoc, decide that 39 analogues is still a reasonably small number of analogues for the reconstruction with a constant number of analogues. Thus, our reconstruction with a constant number of analogues uses 39 analogues.

Considering a fixed standard-deviation criterion, the number of valid analogues can become large for individual years. For example, the largest number of analogues for a single year for a one-standard deviation is 2105 in our approach. We regard this still a subjectively reasonable maximal number. Thus, we choose a one $SD_{noi}$ interval to discuss results for a fixed $SD_{noi}$ criterion. As the previous paragraphs highlight, such a $1\,SD_{noi}$ criterion will fail to find analogues for certain years.

We later show the results for these reconstructions in comparison to the single best-analogue reconstruction. For ensembles of analogues, uncertainty estimates are the full range of the ensemble and an uncertainty envelope based on the intra-ensemble variance.

As a side note, we could also use the individual local values for all proxies to construct a maximally tolerated Euclidean distance. The obvious caveat of this approach is that the analogues may locally lie outside the tolerance range of some of the proxy records although the Euclidean distance is smaller than the maximally tolerated value. On the other hand, the criterion that the analogue should lie within each individual proxy tolerance may exclude the overall best analogue according to the minimal Euclidean distance. We consider this downside acceptable and only consider these. Furthermore, we do not weight

**Table 2.** Proxies considered, their geographic position, and the correlations between the proxy records and the summer (June, July, August; JJA) mean temperature observations from the CRU-TS-3.10 data (Harris et al., 2014) over the period 1901 to 2003. The proxy record data is from PAGES 2k Consortium (2013).

| Proxyname, Country & ID | Lon | Lat | Correlation |
|---|---|---|---|
| Torneträsk, Sweden, Tor92 | 19.6 E | 68.25 N | 0.79 |
| Jämtland, Sweden, Jae11 | 15 E | 63.1 N | 0.65 |
| Northern Scandinavia, Nsc12 | 25 E | 68 N | 0.74 |
| greater Tatra region, Slovakia, Tat12 | 20 E | 49 N | 0.16 |
| Carpathian, Romania, Car09 | 25.3 E | 47 N | 0.56 |
| Alps, Austria, Aus11 | 10.7 E | 47 N | 0.75 |
| Alps, Switzerland, Swi06 | 7.8 E | 46.4 N | 0.68 |
| Alps, France, Fra12 | 7.5 E | 44 N | 0.52 |
| Pyrenees, Spain, Pyr12 | 1 E | 42.5 N | 0.41 |
| Albania, Alb12 | 20 E | 41 N | -0.16 |

the analogues, e.g., according to their distance, because our approach of explicitly considering the uncertainty in the proxies already accounts for the mismatch between proxies and candidate pool.

Recently, Neukom et al. (2019) used a subsampling strategy to assess the uncertainty of reconstructions from an analogue
search. To compare our uncertainty estimates to such an ensemble based uncertainty, we also apply their approach. That is we produce an ensemble of reconstructions by using only half of the available proxy records and half of the available simulation pool. Such an ensemble estimate of the reconstruction uncertainty mainly measures the uncertainty due to sampling variability in the available proxy and simulation data.

More specifically, our set of 8 proxies (see next section) allows for 70 combinations of 4 proxies. We exclude those combi-
nations without any information in Northern Europe. Thereby we obtain 65 combinations of 4 proxies. In addition, we choose 100 sets of simulated candidate fields. Each set includes 4824 candidate fields. We then produce 100 reconstructions for each of the 65 combinations of proxies. That is, our ensemble has in total 6500 reconstructions. We use the same 100 sets of candidate fields for all 65 combinations of proxies. For each date and each reconstruction, we only consider the single best field according to the Euclidean distance.

## 2.2   Data

### 2.2.1   Proxies

The target of our application of the analogue method is a representation of European temperature in summer, June, July, August (JJA), equivalent to the original Euro 2k-reconstruction by the PAGES 2k Consortium (2013). Therefore, we rely on the proxy-

**Table 3.** Simulations in our pool of analogue candidates: ID, forcing components, data reference. We consider for all eight simulations the period 800 to 2005 CE, i.e. 1206 simulated years. Forcings are stratospheric sulphate aerosols from volcanic eruptions (V), variations of total solar irradiance (large amplitude: S, small amplitude: s), changes in earth's orbit (O), land use change (L), greenhouse gases (G); note, only methane and nitrous oxide were prescribed, the carbon dioxide concentration was calculated interactively. For details see data references and Jungclaus et al. (2010).

| ID | Forcing | Reference |
|---|---|---|
| mil0010 | VsOLG | Jungclaus (2008a) |
| mil0012 | VsOLG | Jungclaus (2008b) |
| mil0013 | VsOLG | Jungclaus (2008c) |
| mil0014 | VsOLG | Jungclaus (2008d) |
| mil0015 | VsOLG | Jungclaus (2008e) |
| mil0021 | VSOLG | Jungclaus and Esch (2009) |
| mil0025 | VSOLG | Jungclaus (2009a) |
| mil0026 | VSOLG | Jungclaus (2009b) |

selection of the Euro-Med 2k Consortium (PAGES 2k Consortium, 2013; Luterbacher et al., 2016). PAGES 2k Consortium (2013) and Luterbacher et al. (2016) provide individual references for the proxy records. Table 2 gives the correlations between the proxy series and the CRU-data over the period 1901 to 2003. These correlations enter our considerations on uncertainty as detailed in section 2.1.2. Figure 1 shows the proxy locations.

Since neither the Albanian nor the Slovakian proxy records provided by the PAGES 2k Consortium (2013) explain a relevant portion of the variability of the CRU-TS-3.10 (Harris et al., 2014) summer temperature data at the closest grid-point, we exclude them from the following reconstruction efforts. Already Luterbacher et al. (2016) noted this and, therefore, did not consider these two proxies in their reconstruction effort. That is, we, as Luterbacher et al. (2016), exclude these proxies because there is not a clear relation to temperature. Furthermore, since the Dobrovolný et al. (2010) Central European data is a spatial

average, we also do not consider it in the reconstruction. All three excluded records, however, are subsequently compared to the reconstructed local series.

We describe results for the period 1260 to 2003 CE, although two of the Euro 2k proxy series extend back to the year 138 BC, and the analogue approach is suited to use variable numbers of proxies. The latest start date of any of the used eight proxy indices is the year 1260 CE, and, thus, all eight records cover the period 1260 to 2003 CE. We decide against using uneven

numbers of proxies and against extending the reconstruction further back to ease the comparison of the results and our different uncertainty estimates.

### 2.2.2 Model simulations

Thanks to the PMIP3-effort (Paleoclimate Modelling Intercomparison Project phase 3, e.g., Schmidt et al., 2012) there exists a multi-model ensemble of climate simulations for the last approximately 1100 years. A number of additional simulations comply with the PMIP3 protocol but are not included in the effort (Jungclaus et al., 2010; Fernández-Donado et al., 2013; Lohmann et al., 2015; Otto-Bliesner et al., 2016). Wagner (personal communication, 2016, 2019) has performed a simulation for the last 2,000 years, and Gómez-Navarro et al. (2013, see also Gómez-Navarro et al., 2015a) and Wagner (personal communication, 2014, 2018, 2019, see also Bierstedt et al., 2016, Bothe et al., 2019) have performed regional simulations for Europe for approximately the last 500 years. All these simulations would be suitable as pool of analogues. Especially the PMIP3-ensemble is easily available.

We opt here for a single model ensemble predating the PMIP3-effort but compliant with its protocol, i.e. the millennium simulations with the COSMOS-setup of the Max-Planck-Institute Earth System Model (MPI-ESM) by Jungclaus et al. (2010). This choice bases not least on the assumption that the simulations provide a very similar internal variability. This is beneficial in our case because we rescale the final reconstructions by a chosen standard deviation, which is usually the local full period standard deviation of one of the simulations. Furthermore, one may assume that the single model ensemble provides data with a consistent bias throughout the ensemble, which may ease comparison of the results. On the other hand, such consistent biases may translate to the reconstruction, i.e. a biased reconstruction. This could be avoided by using a pool of simulations from structurally different climate models. Obviously, the shortcomings in simulating ENSO (Jungclaus et al., 2006) are prominent in the MPI-ESM-COSMOS ensemble and affect the results.

We use data centered on the full period 1260 to 2003 CE and the data is normalized with the standard deviation over the same period. Jungclaus et al. (2010) provide details on the simulations (see also data references in Table 3). We use simulation output from the ensemble members including all forcing components for the period 800 to 2005 CE (Table 3). Thereby we have a pool of 9648 candidate fields. Forcings are solar, volcanic, greenhouse gas, orbital, and land use; the carbon dioxide concentration was calculated interactively (compare Jungclaus et al., 2010).

## 3 Results

### 3.1 Single best-analogue reconstruction

Figures 2 and 3 compare the single best-analogue reconstruction to the Euro 2k-reconstruction and the observational data relative to the full period 1260 to 2003 CE. There is generally good agreement between the Euro 2k-reconstruction and the analogue reconstruction but the latter appears to overestimate the warming since the early 19th century (Figure 2a). Note that the observational data is plotted relative to the mean of the Euro 2k-reconstruction over the observational period and solely provides a qualitative comparison. We evaluate our analogue reconstruction against the Euro 2k reconstruction, because we regard the former reconstruction as the main benchmark for the analogue uncertainty estimation. Appendix Figure A1 makes the comparison relative to the period of the observational data. We note that differences between the observations and the

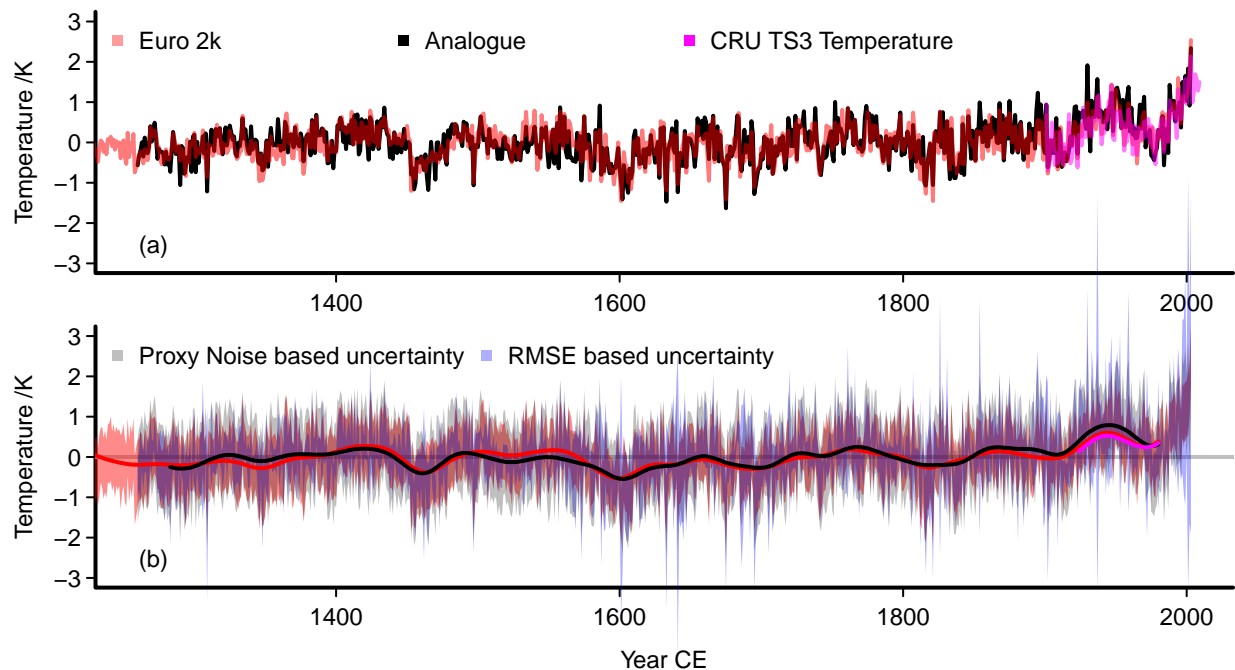

**Figure 2.** The best-analogue reconstruction relative to its full period mean: **(a)** the interannual temperature reconstruction in black, the red line is the area mean Euro 2k-reconstruction, magenta is the observational CRU temperature adjusted to the mean of the reconstruction over its time-range. The analogue reconstruction is rescaled by the variability from one of the simulations. **(b)**: as (a) but for 47-point Hamming filtered data; we further add the uncertainty for estimates for the interannual data: red is the unsmoothed Euro 2k-uncertainty, the grey envelope is a 2 standard-deviation uncertainty based on the correlation between the proxies and the observations at the proxy locations, the blue envelope is a 2 standard-deviation uncertainty based on a MSE-estimate. Panel (b) adds a zero line as visual assistance.

reconstructions are larger for the best analogue approach compared to the Euro-2k reconstruction (Figure 3a and Appendix
Figure A2).

The analogue reconstruction shows rather small centennial variations as does the Euro 2k-reconstruction (Figure 2). We note that the Bayesian Hierarchichal Modelling (BHM) reconstruction by Luterbacher et al. (2016) shows larger variations compared to their composite-plus-scaling reconstruction in the early part of the last millennium prior to our study period. The larger warming since about 1800 in the analogue reconstruction is in line with a slightly larger warming in the BHM-
reconstruction by Luterbacher et al. (2016). Appendix Figure A3 shows a comparison of the best analogues reconstruction to the two European summer temperature reconstructions of Luterbacher et al. (2016). This complements Figure 2 where we show the comparison to the Euro 2k-reconstruction of PAGES 2k Consortium (2013).

Figure 3a shows differences between different data sets as swarm plots. Swarm plots are categorical scatter plots, where the data points are adjusted to avoid overlap between points. Thereby, swarm plots provide information on the distribution of the
data plotted. The differences between the Euro 2k composite-plus-scaling reconstruction and the best-analogue reconstruction

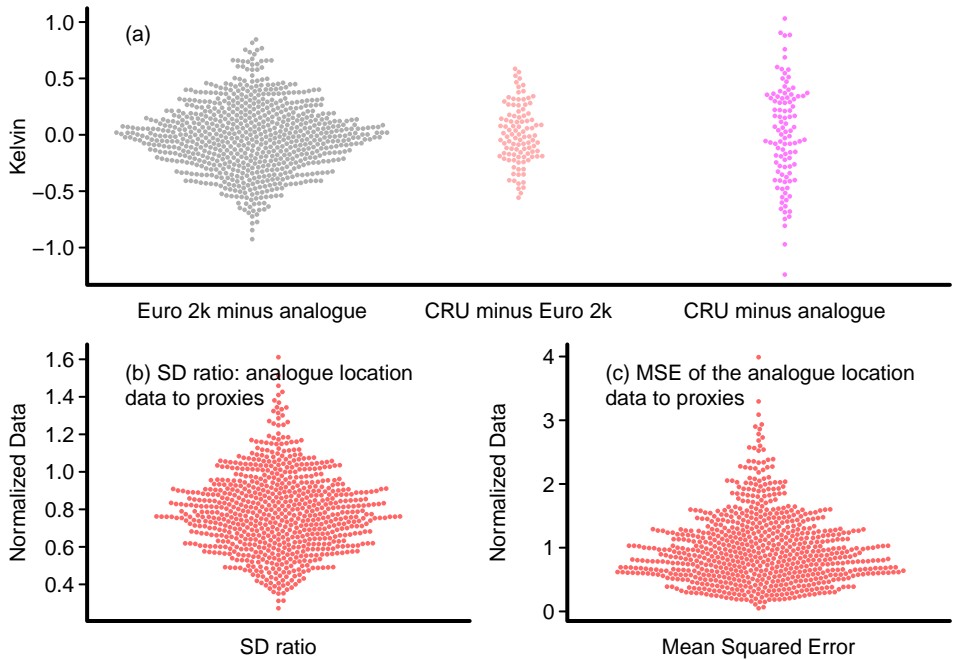

**Figure 3.** Further information about the single best analogue reconstruction: **(a)** swarm plots for the differences between different data sets relative to their periods of overlap: grey is the Euro 2k reconstruction minus the single beste analogue, red is the CRU TS data minus the Euro 2k reconstruction, and magenta is the CRU TS data minus the single best analogue. Since the data are relative to their overlapping period, the visualisation hides potential biases. **(b)**: Ratio between the standard deviations of the normalized analogue values at the closest grid-points to the normalized proxy values. **(c)**: Mean squared error between the normalized analogue grid-point values and the normalized proxies, i.e. the basis for one of the uncertainty estimates in Figure 2. Swarm plots are categorical scatter plots that ensure that points do not overlap.

highlight again their reasonable agreement (Figure 3a on the left in grey dots). These differences do not exceed 1 Kelvin. Time-series plots of the smoothed differences reveal temporal structure with periods of over- and underestimation (not shown). Differences are especially large in periods before the 1600s and since about 1800 (not shown). Differences between the reconstructions and the observational data emphasize that the analogue reconstruction (magenta dots in Figure 3a) disagrees more

with the observations than the Euro 2k reconstruction does (red dots).

Warmest and coldest periods help to characterize the reconstruction. Note that the start date in 1260 CE prevents an assessment of the Medieval Climate Anomaly for the best analogue data. For the period from 1260 to 1850 CE, the Euro 2k-reconstruction and the best analogue both have the warmest 100-year period from 1353 until 1452 CE. Considering the full period until 2003, the last hundred years were warmest. The coldest 100-year period was from 1549 to 1648 CE according

to the best-analogue reconstruction but from 1579 to 1678 CE in the Euro 2k record. Estimates of coldest decades and thirty year periods fall within this coldest century and overlap between both reconstructions. Estimates for shorter warmest periods disagree more.

We now consider the response to volcanic forcing, as volcanoes are considered to be the most important external forcing over the pre-industrial period. They are also the best constrained past climate forcing for the last 500 to 2000 years (e.g., Sigl et al., 2015; Wilson et al., 2016). The period of our reconstructions includes only a few of the large tropical eruptions of the last millennium. We consider a subselection of tropical eruption events in 1286, 1345, 1458, 1601, 1641, 1695, 1809, and 1815. We performed a superposed epoch analysis but we do not graphically show the results. We considered fields and area averages. We chose the five calendar years before an eruption year as reference period, which is a common approach (compare, e.g. Sigl et al., 2015).

Individual eruptions show usually some cooling though it may be quite small (not shown). Noteworthy is the lack of a clear response for, e.g., the Kuwae eruption, which took place in 1458 CE according to Sigl et al. (2015, but see Hartman et al., 2019). The lack of a response in the reconstruction mainly reflects the lack of a clear signature of this event in the proxies entering the reconstruction (not shown). Considering fields for these events, some may show summer cooling, but, e.g., the year 1459 shows widespread slightly warmer conditions.

Considering uncertainties for the reconstructions, Figure 2b shows unsmoothed two standard deviation uncertainties for the Euro 2k-reconstruction and the single best analogue reconstruction together with the smoothed records. We show two uncertainty estimates for the analogue reconstruction. The first is calculated as the square root of the sum over the $Var_{noi}$ for the $N$ invdidual proxies divided by the number of proxies $N$: $\sqrt{\sum_1^N (1 - Var_{noi})/N}$. We assume these represent one standard deviation uncertainties. However, they are only an approximation of the uncertainty. From these we calculate the assumed two standard deviation intervals. These are constant estimates over the full period. The second, time-varying envelope in Figure 2b bases on the mean squared errors between the proxy-values and the best-analogue values at each date. The Euro 2k uncertainty intervals are derived from the data provided by the PAGES 2k Consortium (2013), and they base on the range of a nested composite-plus-scale reconstruction ensemble and the standard-deviation of the reconstruction-validation residuals (see supplement to PAGES 2k Consortium, 2013).

The noise variance based envelope for the best analogue reconstruction is generally wider than the uncertainty of the Euro 2k-reconstruction while the MSE-based analogue uncertainty is usually narrower. The MSE-based uncertainty is also generally narrower than the noise based uncertainty but can become occasionally very wide. The latter widening reflects that the best analogues may fit badly to the proxy records. The mean squared error based uncertainty estimates become particularly wide in the late 20th century highlighting that the single best analogues found for this period do not match the proxy data well. The best analogue reconstruction is generally within the two standard-deviation uncertainty of the Euro 2k-reconstruction. Similarly, the noise based uncertainty estimate for the analogue reconstruction usually includes the Euro 2k-data.

Both uncertainty measures for the analogue reconstruction describe different but not mutually exclusive parts of the uncertainty of the reconstruction. The variance based envelope estimates the reconstruction uncertainty based on the local agreement between proxies and observations over the period when instrumental data is available. Thus, it is unlikely that the uncertainty of the reconstruction at any time is smaller than this estimate because we can assume that the quality of the proxies is best in the recent period. The proxy based noise uncertainty estimate includes local information but extrapolates these over the period without instrumental data. On the other hand, the mean square error captures the misfit between the uncertain proxies and the

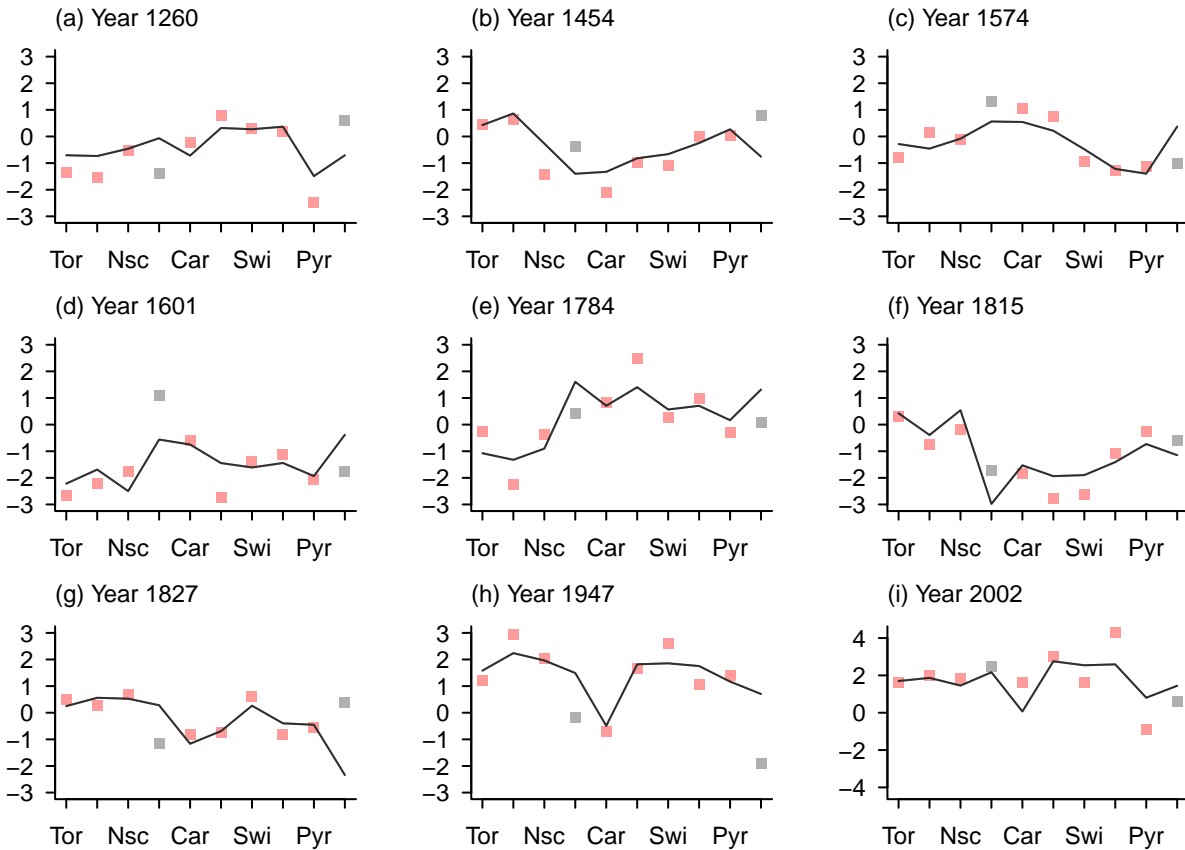

**Figure 4.** Normalised proxy values (squares) for proxies included (red) and excluded (grey) and the values of the best analogue for selected years (lines). Proxy locations on x-axes are from PAGES 2k Consortium (2013): Tor92 (Torneträsk, Sweden), Jae11 (Jämtland, Sweden), Nsc12 (Northern Scandinavia), Tat12 (greater Tatra region, Slovakia), Car09 (Carpathian, Romania), Aus11 (Alps, Austria), Swi06 (Alps, Switzerland), Fra12 (Alps, France), Pyr12 (Pyrenees, Spain), Alb12 (Albania).

final reconstruction product. Where it is smaller than the variance based estimate, we would call it unrealistic. When it exceeds this estimate, it is preferable.

Both measures of reconstruction uncertainty rely on the level of agreement between reconstructed and observed data. In the following, we particularly look at the agreement between the reconstructed data and the proxy data as it enters the MSE-based uncertainty estimate. Figure 4 plots both the proxy-values as squares and the best-analogue values at the closest grid-points as lines for years of interest and arbitrarily selected years. Proxies excluded from the reconstruction are grey and proxies included are red. The analogues agree well with the proxies, e.g., for the year 1827, but notable differences occur as well, e.g., for the

5    years 1601 or 2002. The analogues even appear to occasionally capture the relation between the proxies included and those excluded, which obviously might be by chance. Overall, this small selection indicates that the considered simulation ensemble

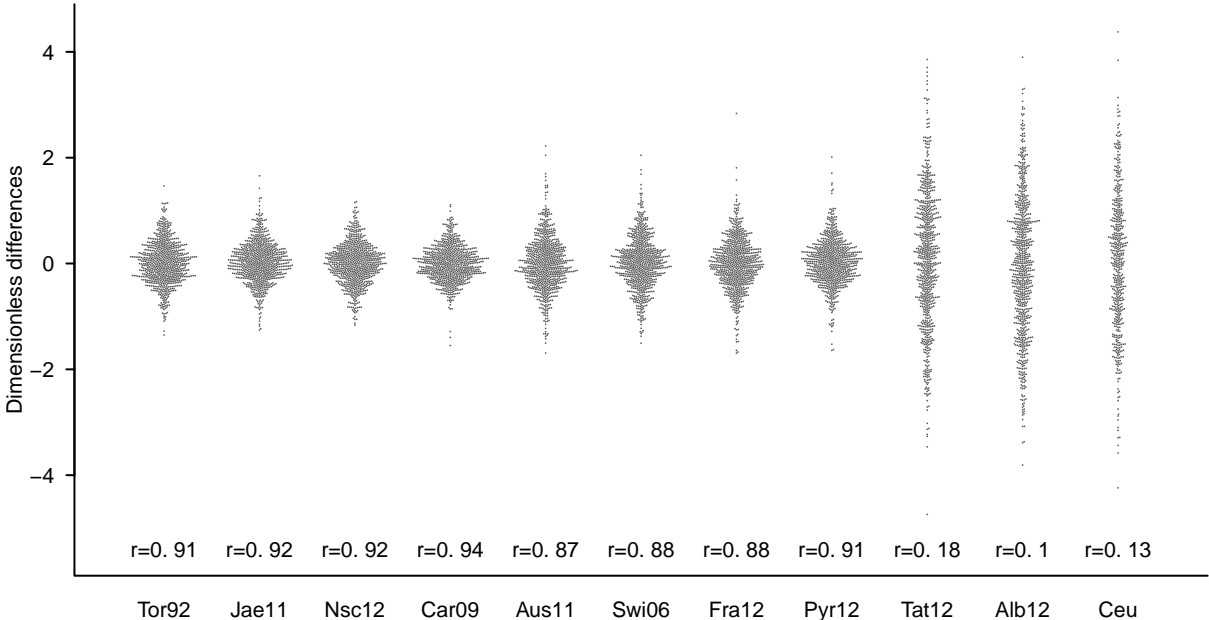

**Figure 5.** Differences between local grid-point series for the single best analogue and the proxy series as swarm plots. Numbers above the x-axis are correlation coefficients between the proxy series and the grid-point records. Proxies are: Torneträsk Tor92, Jämtland Jae11, Northern Scandinavia Nsc12, greater Tatra region Tat12, Carpathian Car09, Austrian Alps Aus11, Swiss Alps Swi06, French Alps Fra12, Pyrenees Pyr12, Albania Alb12. CEu is the Central Europe data. All data is from the normalised series and thus dimensionless.

represents well the relation between the considered regions. We note that for these years and the selected analogues, it is not necessarily the case that spatial clustering of proxies in the Alps or Scandinavia results in close agreement.

A slightly disconcerting feature is visible for, e.g., the year 1947, where the analogue appears to underestimate the intra-location variability. Figure 3b shows the relation between the standard deviation of the best-analogue locations and the standard deviation of the proxy records as swarm plot. While the intra-grid-point variability can be larger than the intra-proxy variation, it is apparent that the ratio is more often smaller than one indicating that the intra-proxy variation is larger.

Figure 3c adds the mean squared error of the best-analogue locations and the proxy values. As already seen in Figure 2 for the mean squared error based uncertainty envelope, the errors are often rather small, but there are times when they become very large. This stresses again that the best analogue may occasionally fit the proxies rather badly.

We do not investigate the differences in intra-location variability in detail. There are a number of explanations on which we only very shortly touch here. First, the noisy proxy series may overestimate the true intra-location variability. Second, our selected simulations may be spatially too smooth. This, thirdly, might be due to the low resolution and simulations with higher resolutions might help then. Fourth, the chosen distance measure may result in such a feature dependent on the characteristics of the simulation pool, which however should usually not be the case. Including a more diverse set of simulations may be the simplest way to investigate this in future applications.

Figure 5 provides a summary evaluation of the local differences by showing swarm plots for the various proxy locations. The figure also gives the correlations between the proxies and the local records from the analogues. Differences are well constrained for the proxies included but become very large for the excluded records. Indeed, correlations are also very small for the excluded records while generally being larger than 0.85 for the included records. The swarm plots hide strong low frequency temporal variability in the differences for the Albanian and Tatra proxies. Some such structures are also apparent in the differences for the proxies included in the analogue search (not shown). Indeed, the Swiss Alps data show a small amplitude multicentennial variation in their local differences.

In summarising, the general agreement between the Euro 2k and the analogue reconstruction as seen in Figure 2 and this section is another encouraging sign that the analogue method is a valid reconstruction tool at least for the considered time-period and regional focus. We give two uncertainty estimates for the single best analogue reconstruction. The potential of the mean squared error based uncertainties to become very large emphasizes that a best analogue may be a very bad fit for the underlying proxies. Indeed, uncertainty levels generally include the other reconstruction, which helps to build confidence in the estimates. We regard this convergence of evidence important for confidence in our understanding of past climates. The strong local deviations at excluded locations (compare Figures 4 and 5) challenge how well the included proxies really represent the European domain and its intra-regional relations.

## 3.2 A set of 'good' analogues

As described above, we also consider a reconstruction based on a set of good analogues. One could base such a selection on an arbitrary number of, e.g., 10 analogues. However, we base our choice of the number of analogues on our considerations in section 2.1.2 on the uncertainty of the local proxies and on the number of analogues available for different uncertainty levels of the proxies. That is in our case, a $2.57 SD_{noi}$ uncertainty interval for the proxy values allows for at least 39 analogues for each date. Thus, we select 39 analogues at the locations of the grid-points closest to the proxy-locations.

Figure 6 presents local results for the analogue search reconstruction for the case of a fixed number of analogues. We display the correlation coefficients between the proxies and the reconstructed local series medians at the top of each panel next to the proxy ID. They are between 0.84 and 0.98 for the anchor locations of the reconstruction. They are weak for the two locations excluded, i.e., Tatra and Albania. The correlation coefficients are larger than the correlations to the observational data. That is, the proxies included in our analogue search constrain the search effectively towards the proxy values. This holds especially for the median, which is a filter for the data of the reconstruction ensemble members. The aim of the analogue search is to match the observations, i.e. the proxies, closely with the simulated data. We stress that these correlations only indicate agreement with the proxy records not with the true temperature.

The good agreement between the proxies included in our analogue search and our reconstructed local series extends be-yond correlations. The range of reconstructed values usually is narrow for these proxies. However, there are also obvious mismatches, e.g., 16th century warmth in the Austrian Alps and, more frequently, individual very cold excursions, which are not matched in the analogues (Figure 6). Plotting local analogue data against the proxy series highlights how commonly the reconstruction median and random individual analogue members do not match the extreme values of the proxies (not shown).

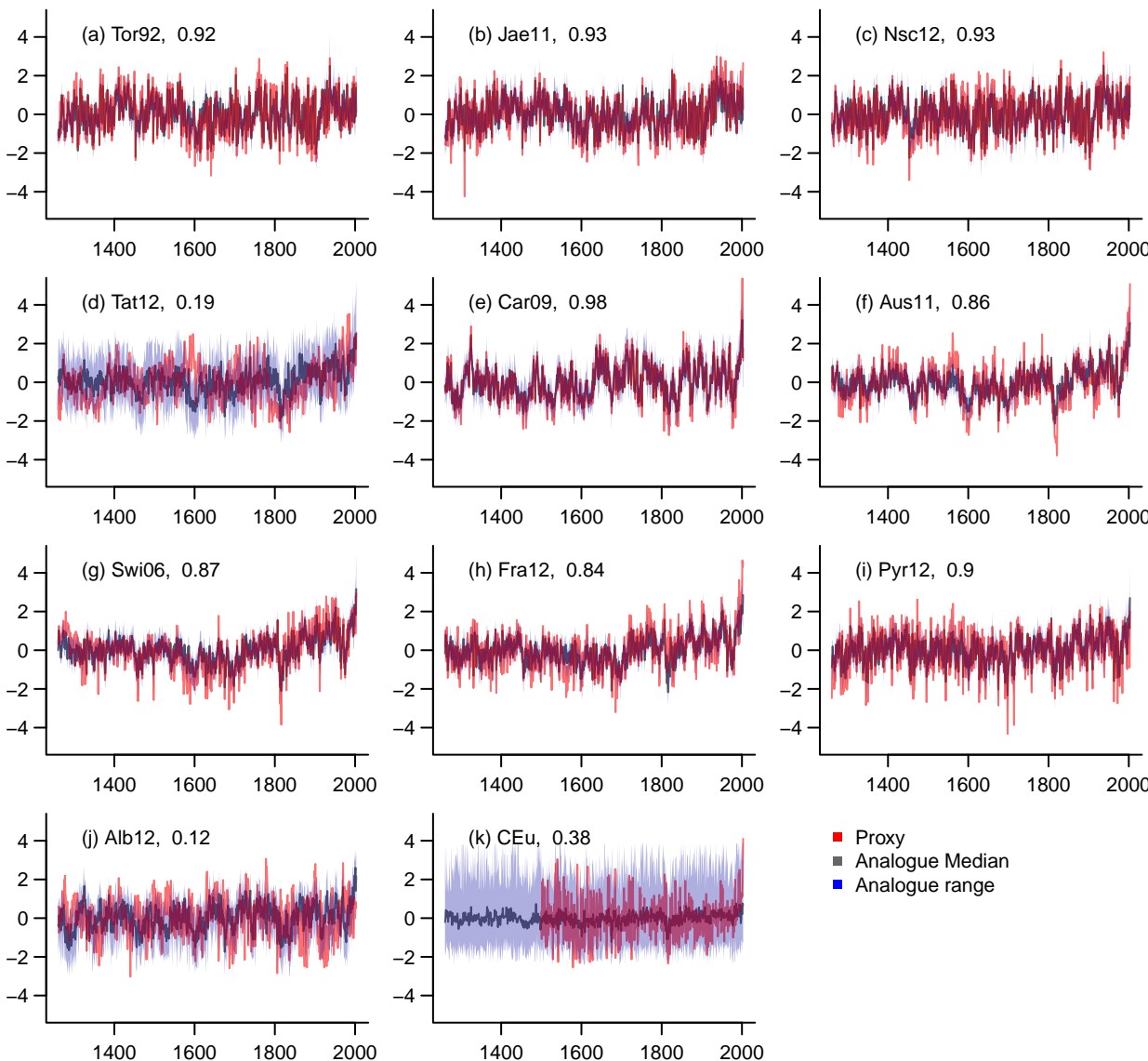

**Figure 6.** Analogue reconstruction values at the locations of the Euro 2k-proxies. Shown are the normalized proxies in red, the median of 39 analogue values in black and the full range of the 39 local analogues in blue. X-axes are years CE. Correlations between the local reconstruction median and the proxy series are given as numbers next to the proxy IDs in each panel.

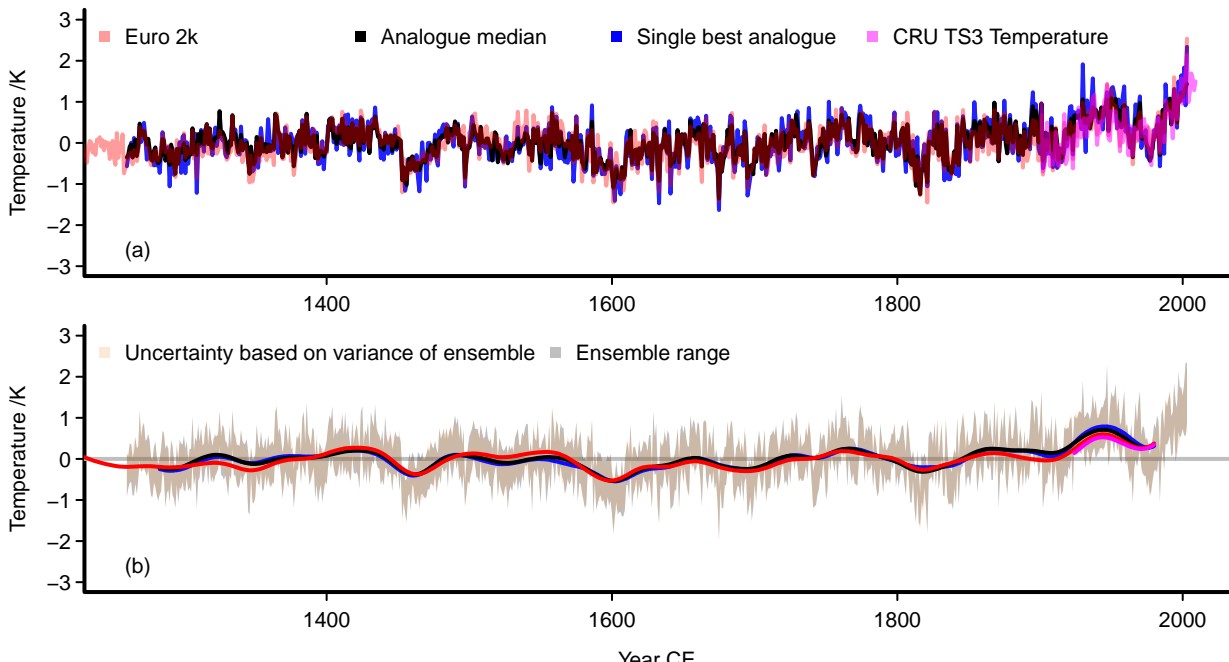

**Figure 7.** The analogue reconstruction for the 39 best analogues: **(a)**: the interannual rescaled temperature reconstruction median in black; the blue line is the single best-analogue reconstruction; the red line is the Euro 2k-reconstruction; magenta is the CRU temperature adjusted to the mean of the reconstruction-median over the CRU period. **(b)**: the unsmoothed uncertainty, estimates in light grey are the ensemble range, and the brown envelope gives a two standard deviation interval based on the variance of the 39 samples; note that both uncertainty estimates are hardly distinguishable on this scale, the panel adds the series from (a) but for 47-point Hamming filtered data. Panel (b) adds a zero line as visual assistance.

These considerations highlight that, although the analogues may be well constrained locally, this gives no indication about the strength of the relations away from the anchoring locations. Indeed, correlations with the observational CRU data are in line with the correlations between the proxies and the CRU data (not shown). Section 3.6 below shows that, indeed, the correlations in Figure 6 do not necessarily reflect how well the reconstruction captures the observed temperature elsewhere.

Figure 6k shows the comparison for the spatial average summer temperature for the Central European area (Dobrovolný et al., 2010). This mean is computed over the grid-points from 7.5E to 18.75E and 46.4N and 50.1N in the coarse resolution model data. This domain obviously represents a larger area than the data by Dobrovolný et al. (2010). There is not any identifiable variability in the uncertainty envelope. Consequently the median also shows very little variability. Nevertheless the variability is comparable between central European data for the analogue reconstruction and the original record if one considers individual members. Although the temporal variations of the median are muted, the median-record still correlates notably but not strongly with the central European data of Dobrovolný et al. (2010).

The fixed number analogue reconstruction also agrees well with the Euro 2k-reconstrution (Figure 7) as did the single best analogue approach. Indeed the median of the fixed-number analogue-ensemble correlates slightly better with the Euro 2k-

reconstruction at $r \approx 0.89$ compared to the single best analogue ($r \approx 0.82$). The variability of the median of the analogues, however, is approximately 8% smaller than the variability of the Euro 2k reconstruction and approximately 17% smaller than the variability of the single best analogue reconstruction. Similarly, while the range of the best analogue is comparable to the Euro 2k-reconstruction, the range of the 39-analogue ensemble median is strongly reduced compared to both other series. The coldest values are only slightly warmer but the warmest values are about one Kelvin colder than for the other two series. Therefore, using a set of analogues to produce a reconstruction suppresses variability. This reduction of variability for median or mean based reconstructions is expected due to the compensation of noise and within the individual members. It is well established that such a trade-off between accuracy and variability exists for analogue search algorithms (Gómez-Navarro et al., 2015b, 2017).

Although the uncertainty of the regional average for Central Europe shows a wide uncertainty for the 39 analogues, the full domain reconstruction has a rather narrow uncertainty range. The full ensemble range and a two standard deviation uncertainty based on the variance of the ensemble are nearly indistinguishable in Figure 7. The included proxies anchor the area mean reconstruction to a narrow range of variability if we choose a fixed number of analogues.

The distribution of the uncertainty estimates of the 39 analogue median is narrower than for the single best analogue, and the distribution also has smaller values than for the two estimates for the single best analogue. However in this case, the variability of the fixed number of analogues does not encompass the full range of potential analogues compliant with a specific uncertainty level. Again we note that as long as an uncertainty estimate is smaller than the proxy noise based estimate as seen in Figure 2, we think one should use the proxy noise based uncertainty.

Interannual differences between the single best-analogue reconstruction and the median of the 39-analogue reconstruction appear to be of similar size as the interannual differences between the Euro 2k-reconstruction and the 39-analogue median (not shown). The smoothed representations align quite well for the two different analogue approaches. On the other hand there are some systematic differences between the 39-analogue median and the Euro 2k-reconstruction in the smoothed version particularly in the 14th and 16th centuries and since approximately the year 1850. We generally assume that such systematic differences are due to differing sensitivities between the regression based approach of the Euro 2k-reconstruction and our analogue search. However considering the mid 16th century, the work by Wetter and Pfister (2011, 2013) may suggest that indeed our simulation pool is insufficient for this period and the Euro 2k-data more reliably captures the temperature then.

Differences between the two analogue approaches do not show such systematic differences except maybe for the early 20th century. Both analogue approaches appear to overestimate the warming trend since the early 19th century. This is more pronounced in the single best reconstruction compared to the median of the 39 analogues, for which we already noted the reduced variability.

The coldest and warmest periods are very similar in the 39-analogue reconstruction compared to the best-analogue version. Again, coldest conditions on decadal, 30-year, and centennial time-scales occur mainly in the 17th century (not shown). This holds for the median as well as the coldest and warmest analogue estimates for the periods. Regarding well dated tropical volcanic eruptions, we again find summer cooling following some events but others barely leave a signal in the European area

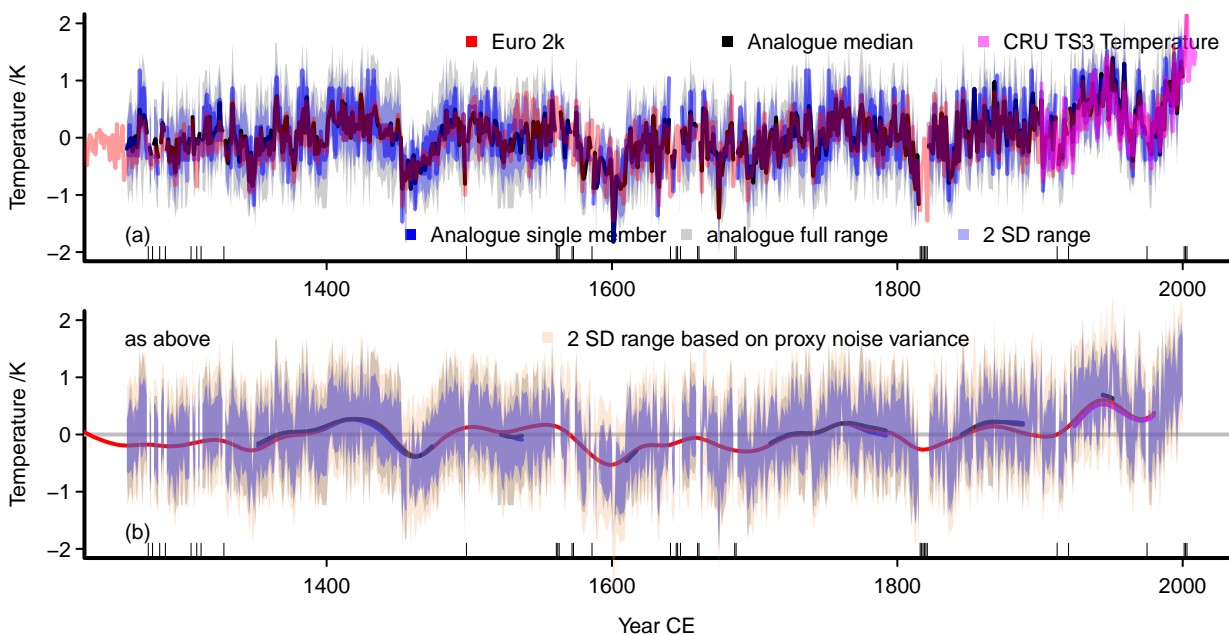

**Figure 8.** Analogue reconstruction based on an $1SD_{noi}$ uncertainty of the proxies. **(a)**: Interannual data: red, the Euro 2k-reconstruction; black, the analogue median; blue line, a single analogue member, blue shading, two standard deviation uncertainty range around the analogue median based on variability of the analogues, grey shading, the full range of analogues; marks at horizontal axis mark unsuccessful analogue searches. **(b)**: As (a) for 47-point Hamming filtered data, but we add a two standard deviation uncertainty based on the square root of the proxy noise variances in brown as also shown in Figure 2. Panel (b) adds a zero line as visual assistance.

mean data (not shown). For spatial fields, similarly, there is not a distinct signal of post-eruption summer cooling. The range of analogues even allows for some regional warming.

## 3.3 Analogues within $1SD_{noi}$

The use of a fixed number of analogues in the previous section implies that we consider for each date a different level of proxy uncertainty according to our considerations in section 2.1.2. Next, we shortly present a reconstruction for which we consider only those analogues falling within a certain uncertainty interval around all of the original proxies for each date. This will

5  result in an uneven number of analogues at each individual date. We use a fixed one noise-standard-deviation interval around the proxy values. The method is more likely to find valid analogue for all dates if we choose larger uncertainty intervals. However, larger intervals imply that the number of analogues may become exceedingly large for certain dates. As mentioned above, the one standard deviation interval has a maximal number of 2105 possible analogues, which one may already rate as too many.

10  Figure 8 displays the results for such an analogue reconstruction collecting all analogues within one noise-standard-deviation around the proxy values. Again there is good agreement between the analogue reconstruction and the Euro 2k-reconstruction.

Blue lines in the upper panels of Figure 8 show one single member of the reconstruction ensemble which also compares quite well to the Euro 2k-reconstruction.

As indicated before, if one chooses smaller uncertainty-intervals around the proxy values, it becomes more likely that the method fails to identify suitable analogues. This becomes obvious when considering the smoothed estimates. This way of constraining the analogue space quite frequently fails to provide any analogue at all. Small ticks at the time-axes of Figure 8 show that such failures appear to cluster in the 13th and 14th centuries, in the 16th and 17th centuries and in the early 19th century. A number of these are years with strong forcing from volcanic eruptions (compare Sigl et al., 2015). This is a
shortcoming of our approach to uncertainty in this section. Our results in previous sections as well as subsampling approaches (e.g., Neukom et al., 2019) do not have this specific problem.

Another period without suitable analogues occurs after the year 2000 CE. Considering the results of Jungclaus et al. (2010, e.g., their Figure 3), one might have hoped that the COSMOS-millennium simulation ensemble includes analogues also matching the recent summers. However, we do not search analogues that only fit the observed area mean warming regionally or
globally, but we search for analogues that also represent the interrelation among the proxy locations and do so within a fixed noise threshold. Thus, it is unsurprising that we fail to find analogues. The European temperature slowly leaves the temperature range observed in approximately the previous 750 years and we have only few candidate fields that may represent the warm climate after the year 2000 CE, e.g., the summer heat of the year 2003 CE (compare, e.g. Wetter and Pfister, 2013; Black et al., 2004; Stott et al., 2004; Garcia-Herrera et al., 2010). Additional gaps occur in uncertainty envelopes based on the ensemble
variance when there is only one valid analogue.

Figure 8 shows three differenct uncertainty estimates. For one, there is in both panels in grey the full range of the analogues that comply with the one standard deviation noise around the proxy values. Second, the panels show in blue a two standard deviation uncertainty based on the variance of the ensemble members at each date. The latter is in this case usually notably narrower than the full range, which reflects to a good part simply the number of available analogues. We also add in Figure
8b an assumed two standard deviation uncertainty envelope based on the proxy noise at each individual proxy location. It is slightly wider than the full range of the ensemble.

The occasional failure of the method to find analogues complicates any attempt to identify coldest centuries. That is, the validity of any identified period is limited and, thus, the exercise is of reduced value. However, the coldest decades and 30-year periods again are in the early 17th century as for our other approaches. We find the warmest periods usually centred about the
early 15th century for the period before 1850 CE, which compares well with the Euro 2k-reconstruction. However, considering only the warmest estimates of the envelope, the warmest decade occurs in the second half of the 18th century, which is more in line with the estimates of our other analogue approaches.

The lack of appropriate analogues also hampers evaluating the response to well dated tropical volcanic eruptions. For example, there are no analogues available for the year without summer 1816 CE. Otherwise, the common feature is again that some
eruptions appear to have resulted in European summer cooling while there is no identifiable imprint for other eruptions in our European area mean data (not shown). Comparing spatial fields for this reconstruction, anomalies are more homogeneous but

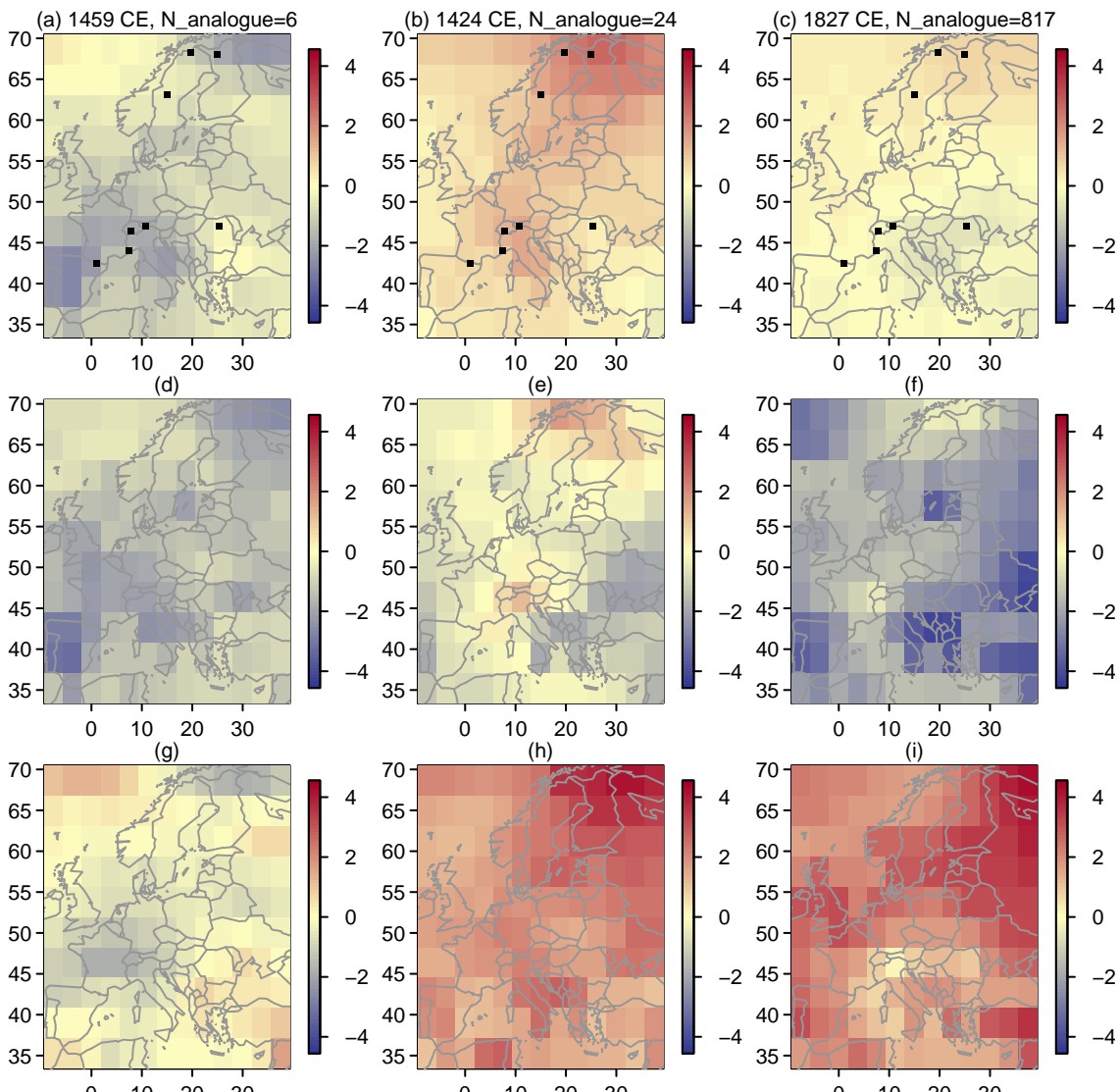

**Figure 9.** Analogue fields for three reconstructed cases with different numbers of analogues, color bars are temperature anomalies in Kelvin relative to the full period. From left to right, 1459 CE with 6 analogues, 1424 CE with 24 analogues, and 1827 CE with 817 analogues. From top to bottom, median, local minimum and local maximum. Black dots signal the proxy locations in the top row.

also smaller than for the reconstruction from 39 good analogues (not shown). While we find cooling, the wide range of the analogues also allows for notable warming for some eruptions.

Up until now, we concentrated on time-series. Figure 9 shows how the analogue reconstruction can provide diverse spatial representations for the same set of proxy-values. It can give several different reconstructions that strongly differ from each other. The example years are chosen to represent a rather cold, a rather warm, and an approximately average year, and the top

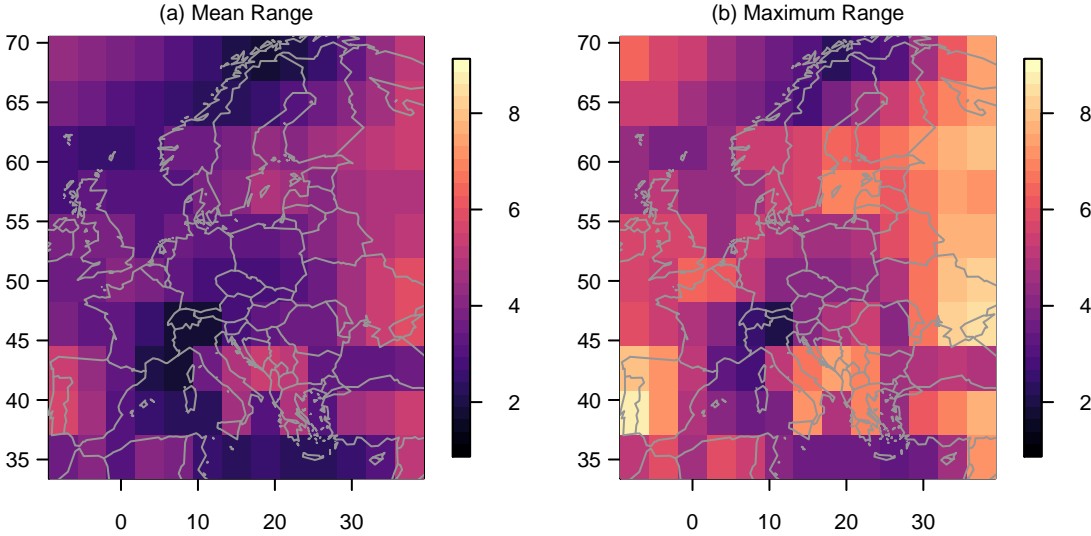

**Figure 10.** The local range of analogues over the reconstruction period: **(a)** Mean range; **(a)** Maximum range occurring over the period.

row shows the median of the found analogues for the three cases of 1459 CE, 1424 CE, and 1827 CE. Incidentally, these are also three years for which we find few, i.e. 6, reasonable, i.e. 24, and as many as 817 analogues in a one standard-deviation interval. The subsequent rows add the local minimum and maximum values respectively. Black dots in the top row show the
original proxy locations. Note that the Figure displays temperature anomalies from the mean over the full period in Kelvin.

It is surprising that, e.g., the proxies anchor the year 1827 in Turkey only within a range of up to 8 Kelvin for the more than 800 analogues. Even central Scandinavia may be rather cold or rather warm although it should be constrained by three proxy records. Indeed the best analogue for that year is close to the proxies (compare Figure 4).

The 24 analogues for the year 1424 have a tendency to warm values but again warm and cold conditions are found within a
one standard deviation interval around our proxy anchors for south-eastern and south-western Europe. On the other hand the six analogues available for the year 1459 mostly give slightly cold conditions over wide parts of the domain and especially for continental Europe.

Figure 10 reflects on the potentially very wide local range of the analogues. It shows the mean range and the maximum range of the ensembles for the field. Thereby, it summarises the local uncertainties for the analogue fields. Dependent on location, the
mean range of the ensemble is between approximately 1.7K and approximately 5.9K (Figure 10a). The mean range is generally large at the eastern border of the domain, and it becomes also large over the southern Adriatic Sea, the central Baltic Sea, and particularly at the western boundary over the Iberian Peninsula. The local maxima of ranges over time mirror the distribution of the mean ranges. Further, they emphasize how weakly constrained the reconstructions are throughout the domain (Figure 10b).

We noted for Figure 4 that it is not necessarily the case locally that individual analogues fit better in regions with multiple proxies. However, the mean ranges in Figure 10 are indeed smallest in northern Scandinavia and the Alps, and small ranges

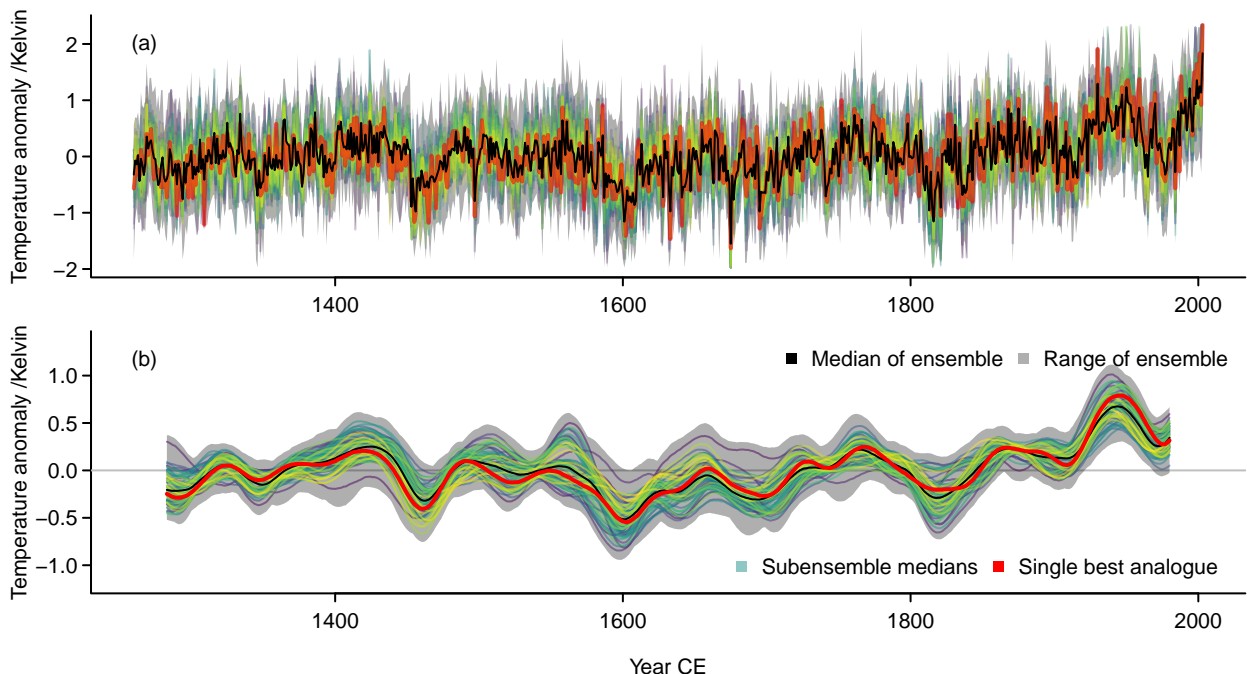

**Figure 11.** Ensemble of subsampled reconstructions: **(a)** Interannual data, **(b)** data smoothed with 47-point Hamming filter. Both panels show in grey the full range of 6500 reconstructions based on 4 proxies and only half of the simulated fields, in black the median of all 6500 reconstructions, and in red the single best analogue reconstruction based on the full data. Further colored lines are medians of the 65 subensembles using different sets of 4 proxies. Panel (b) adds a zero line as visual assistance.

extend towards the French coast of the Mediterranean. Ranges are also small along parts of the southern border of our domain. Except for this latter region, these are generally the areas where multiple proxies cluster. That is, these fields again show

5 that one can expect for the analogue search that the overall range and thereby the uncertainty estimates of the reconstruction are narrower close to clusters of proxies, if the proxies well constrain the reconstruction (compare also Franke et al., 2010; Gómez-Navarro et al., 2015b).

The fact that the fixed uncertainty analogue search commonly fails in finding suitable analogues obviously reduces its value if we are interested in complete reconstruction series. However, such deficiencies also provide valuable information about

10 how well our pool of analogues represents the variability recorded by the proxies within a certain interval of confidence. Furthermore, the occasionally large numbers of potential analogues together with their potentially locally wide range are a note of caution that field reconstructions may be of limited value locally even if the area mean is a valid representation of past mean climates.

### 3.4 Reconstruction ensemble by subsampling

Recently, Neukom et al. (2019) assess the uncertainty of an analogue search reconstruction by subsampling the available proxy data and the pool of availabe model simulation fields. As outlined above we adopt such a subsampling procedure to compare the results to our reconstructions. Figure 11a shows the range of the full ensemble as well as the medians of the subensembles for different combinations of the proxies and for an annual resolution of the data. Panel (b) presents the smoothed data. Both panels add the single best analogue reconstruction based on all data for comparison.

Individual reconstructions and the median of the subensembles differ strongly from one another and also may display strong differences to our single best analogue reconstruction using all data. However, the overall median and the single best analogue from all data agree well in their smoothed representation. Differences are most visible in the 14th to 16th centuries, the early 19th century, and the middle of the 20th century. The range of the subsampling ensemble is slightly larger than most of our discussed uncertainty estimates but is still generally smaller than the assumed two standard deviation uncertainty based on the proxy noise.

The subsampling uses only four out of eight available proxies for our domain and their coverage may be very uneven. Nevertheless, even subselecting the proxies appears to validly constrain the candidate pool with respect to the regional mean although with notable uncertainty. We do not provide further evaluation of the subsampling ensemble. In view of the results of our previous analyses, we presume that four proxies may indeed provide a constraint on the area mean, but will fail to do so locally.

### 3.5 Comparison of uncertainties

Figure 12 plots histograms for the various described uncertainty estimates of area mean reconstructions. Ensemble ranges are not necessarily symmetric around their median. Most other estimates are symmetric and we plot only positive values. The vertical line in Figure 12 shows the constant estimate for the correlation based uncertainty.

We note that the uncertainty distribution for the subsampling based ensemble range is centred at larger values compared to most of our other estimates using the full set of proxies. Including less proxies in the search is a weaker constraint on the candidate pool compared to using all proxies and therefore the range of potential analogues also likely widens. The wide range of the root mean squared error based estimate for the single best analogue is mainly due to the large errors in the late 20th century as already seen in Figure 2. Distributions of our uncertainty estimates are generally comparable to the uncertainty estimates from the Euro 2k effort.

Neither the estimates of the fixed number of analogues nor the fixed one standard deviation interval likely represent the full range of uncertainty. For most dates, the fixed number of analogues represent only part of all valid analogues according to our assumptions on local uncertainty, and the fixed one standard deviation interval is by construction a rather narrow estimate. The assumed two standard deviation uncertainty estimates based on the proxy noise are generally larger than estimates from all other approaches as seen in the green line in Figure 12. Nevertheless our results highlight that our reconstruction efforts may

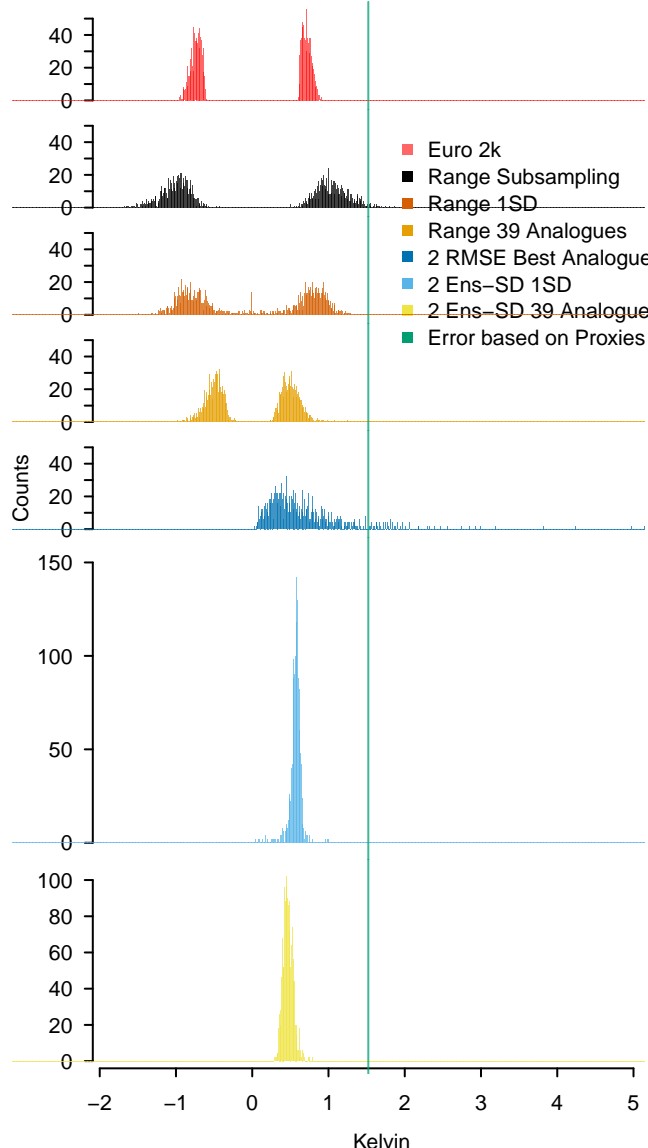

**Figure 12.** Comparison of uncertainty estimates: from top to bottom, histograms in bins of 0.01 Kelvin for: Euro 2k two standard deviation uncertainty (red), range of subsampling ensemble (black), range of fixed one noise standard deviation ensemble (dark orange), range of 39 analogues (light orange), assumed two standard deviation interval for the MSE based uncertainty estimate for the single best analogue (dark blue), ensemble variance based two standard deviation interval for the one noise standard deviation (light blue), ensemble variance based two standard deviation intervals for the 39 analogues (yellow). We only show one-sided estimates when the estimates are symmetric. The green line througout the panel marks the uncertainty estimate based on the proxy noise.

only be weakly constrained. They also indicate that many uncertainty estimates may be optimistic, if we assume the proxy noise based estimate to be indeed a relevant representation of the uncertainty due to the noise in our local information.

## 3.6 Comparison to station data

Station data allow to evaluate our reconstruction against sources of information independent of the proxies or other reconstructions. The Berkeley Earth project (BEST Muller et al., 2013) provides regionally representative series, which we use in the following for a short comparison. We choose those regionally representative series close to locations of long instrumental records. Figure 13 shows a selection of such comparisons with the median of the one standard deviation reconstruction ensemble. The appendix provides equivalent comparisons for the single best analogue and the approach using a fixed number of analogues (Appendix Figures A4 and A5).

Correlations are often of notable strength between the reconstructed median data close to locations of the long instrumental records with the regionally representative data series from the BEST project (Muller et al., 2013), see numbers in panels of Figure 13. Correlations are largest in Scandinavia and around the Alps. Both regions are where most proxy records are located.

Comparing the data series, however, indicates notable shortcomings of the reconstruction median. The reconstruction median often overestimates the recent warming trend and the median shows notably less variability than the BEST-series. The underestimation of the variability on the other hand leads occasionally to an underestimation of the most recent warm anomalies. The top-left panel for Nuuk highlights that the lack of constraints on the reconstruction can result in potentially artificial spikes in the time-series. There are also cases where both series appear to agree quite well over the period when both are available. Examples are the Central England Temperature and Montdidier. Comparisons look similar for our other two reconstruction approaches (see Appendix Figures A4 and A5).

## 4 Summary and Discussions

Earlier proxy surrogate reconstructions from the analogue method usually considered the single best match or a small set of best fits to reconstruct past climate states compliant with limited local proxy information. Testing the analogue method against a prior reconstruction for the European domain shows that the method indeed allows to reconstruct past climate variability comparably to more common approaches. It appears even to appropriately capture the intra-proxy variability and the proxy-variability over time. This holds for different implementations of the method using either a single best or multiple good analogues.

Our focus, however, is on the uncertainty of reconstructions by analogue search. The method traditionally neglects the uncertainty of the final estimate. An exception considering the Common Era of the last 2000 years is the study by Neukom et al. (2019). They use a subsampling approach to provide an ensemble of reconstructions, which allows to use the ensemble range as a measure of uncertainty of the reconstruction.

We describe alternative approaches of obtaining uncertainty estimates for analogue reconstructions, which do not require to reduce the available information from proxies and simulations. These estimates rely ultimately on the assumption, that

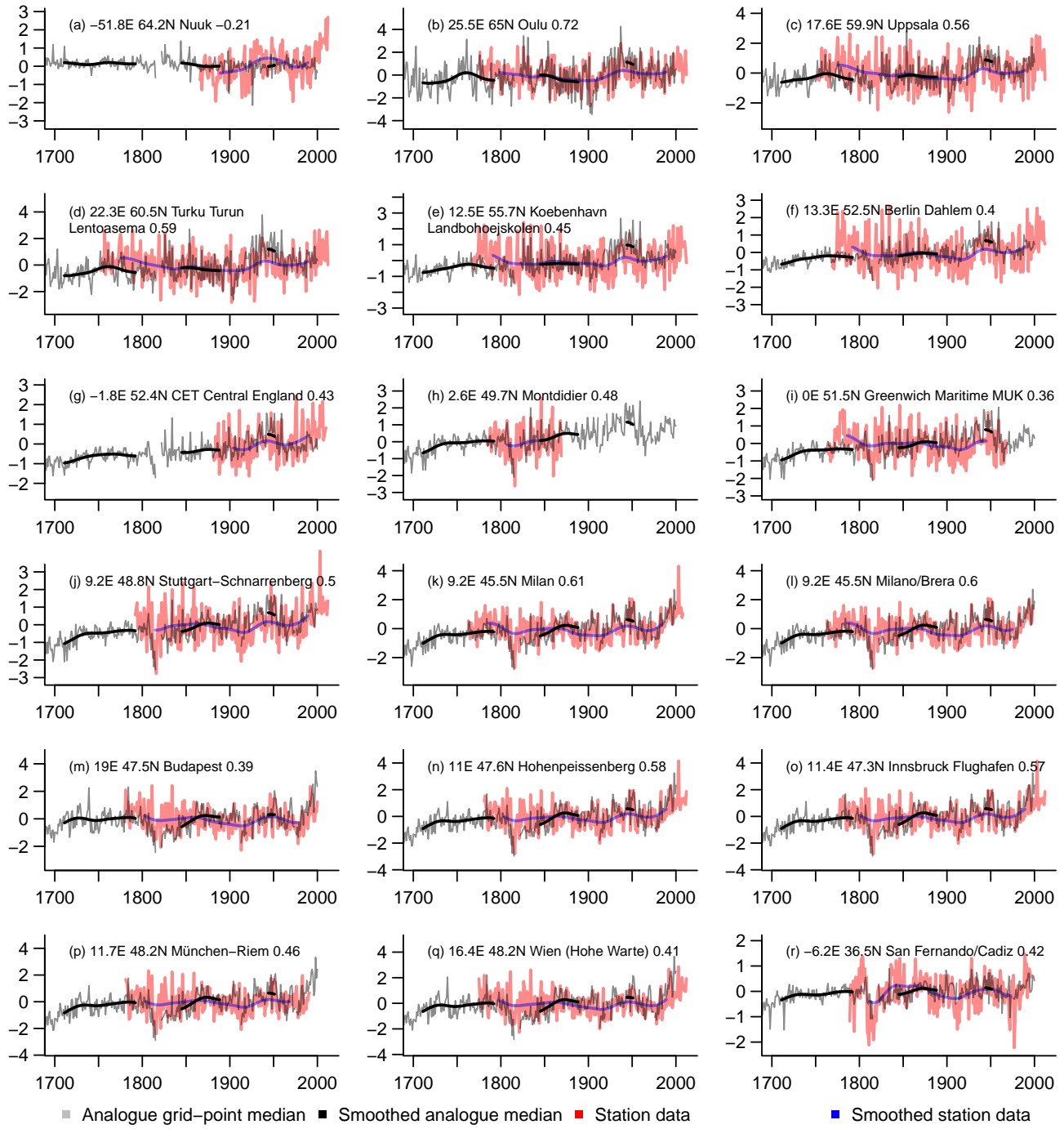

**Figure 13.** Comparison of local grid-point analogue data for the fixed one standard deviation approach with an arbitrary selection of regionally representative data from BEST. Location, station name, and correlation over available station data are at the top of the panels. Grey and black, interannual and smoothed analogue median. Red and blue, station data and its smooth. X-axes are years CE. Y-axes are temperature anomalies in Kelvin relative to the period where both datasets are available.

the calibration correlation between a proxy record and climate observations gives us information about how well the proxies represent the climate. We use these correlations to construct an estimate of the uncertainty of the area average reconstruction based on these proxies. The square root of the sum over the $Var_{noi}$, i.e. the residual noise variability for the invdidual proxies, divided by the number of proxies gives a simple uncertainty estimate for the analogue search that by construction should be an upper limit for the best analogue deviations if the best analogues are within this range. For the case of single best analogue reconstructions, we further show uncertainties based on the mean standard error between the local best analogue values and the proxy values for each reconstructed date.

We further construct two types of reconstruction ensembles based on our estimate of the local proxy uncertainty. For these ensembles, we provide two uncertainty estimates, which are their full range and an estimate based on the variance of the ensemble members. Ensemble envelopes reflect the mean uncertainty, whereas estimates based on the proxy noise generally are local uncertainties.

The uncertainty estimates from the subsampling using degraded information and the range of an ensemble from a fixed one standard deviation proxy noise uncertainty are similar to the uncertainty estimates from an earlier reconstruction of European summer temperature. However, our uncertainty estimates vary more than these earlier estimates. Most other estimates have a tendency to be samller than these earlier reconstruction uncertainties although reconstructions are comparable. The time constant estimate based on the proxy noise is larger than prior and present uncertainty estimates except for cases where the uncertainties clearly reflect that the reconstruction is a bad match to the anchoring proxy information.

We note that problems arise if we use a fixed uncertainty interval around the proxies. In this case, we are not able to obtain good analogues for some dates. Our approach is particularly unlikely to find valid analogues in the fixed uncertainty level setup for years of strong observed cooling, e.g., due to strong volcanic eruptions. This is a fundamental shortcoming of an analogue search that considers uncertainty in the way we do in this case. Our other estimates as well as the approach by Neukom et al. (2019) do not show this behaviour. More generally our results also suffer from similar shortcomings as the work by Franke et al. (2010, see also Gómez-Navarro et al., 2015b and Annan and Hargreaves, 2012), i.e. the quality of the reconstruction diminishes further away from the anchoring proxies.

We only consider complete proxy records starting at the same date with the same temporal resolution. However, the analogue method does not rely on these assumptions. It easily compensates for missing values and data with different resolutions. Gómez-Navarro et al. (2017) and Jensen et al. (2018) provide some analyses in this direction. The method however depends strongly on the pool of available analogues and the criteria for selection of analogues.

While we focussed on the temperature fields, it is easy to additionally reconstruct other variables that are compatible with the temperature proxy records, since the climate models do not only simulate surface temperature but the full climate/weather situations (compare, e.g. Diaz et al., 2016; Wahl et al., 2019). This could produce a relevant probabilistic estimate of these past situations. However, the reliability of these samples obviously depends on the strength of the link between the local temperature and other large scale fields. Similarly it is possible to obtain larger scale climate estimates compliant with the regional information, e.g., hemispheric means, and compare these to situations compliant with other proxy information. A caveat in all these considerations are the findings by Annan and Hargreaves (2012), who note that reconstructions by comparable methods

may not give the correct posterior distribution if we have a large number of proxies with small uncertainty, while if we have only few proxies with large uncertainties, the final reconstructed estimate may be not very meaningful due to a lack of accuracy.

We have to note that the reconstruction neglects possible information about the past climate forcing trajectory. This has implications for dynamical inferences, which may be misleading. While one can account for this by including the forcing reconstruction in the anchoring dataset, this reduces the pool of potential analogues. Furthermore, all results depend on the

10 consistency and quality of the pool of analogues, i.e. the simulations and the underlying sophisticated climate models. An interesting extension of our approach can be to preprocess the simulation pool data by using proxy forward models (Evans et al., 2013). This could more validly constrain the candidate pool.

Applications of the analogue method commonly only focus on the best analogue. The failure to find any analogue and the occurrence of multiple good analogues raise the issues of extrapolation and interpolation of the analogue pool and the analogue

ensemble. Interpolation of analogues may be of interest for obtaining one optimal representation for the reconstruction. More crucially, extrapolation is one solution to obtain reconstructions for situations, e.g., extremes, which are not included in the pool of potential analogues. Extrapolation of the current pool may be possible by generating synthetic analogues. Data science methods may be available to do this.

## 5  Concluding remarks

Proxy surrogate reconstructions from the analogue method often neglect that the proxies and, in turn, the reconstruction are uncertain estimates. Here, we suggest uncertainty estimates for single best-analogue reconstructions as well as analogue reconstructions from multiple good analogues. We are primarily interested in the case where we only consider analogues which fall within a certain uncertainty interval of the original proxies.

5 We compare reconstructions and uncertainty estimates to a previously published reconstruction. This evaluation suggests that the analogue approaches capture the variability as well as a composite-plus-scaling approach. The analogue reconstructions also appear to capture the intra-proxy variability and the proxy-variability. Similarly, our results suggest that our approach compares well to independent data.

If we only use analogues, which comply with the proxies within a certain uncertainty interval, the problem arises that there

10 may be no compliant candidates in the pool of simulated fields. Generally, the uncertainties and the evaluation of the local range of reconstructions suggest that the proxies only loosely constrain the reconstructions.

Upscaling the local proxies to obtain larger scale climate information holds many opportunities to infer information about past climate states. However, one has to add relevant estimates of uncertainty to provide meaningful information.

*Data availability.* The simulation data is available from the World Data Center for Climate (WDCC) at https://cera-www.dkrz.de/WDCC/ui/cerasearch/project?acronym=MILLENNIUM_COSMOS (last accessed, 21 May 2019). The Euro 2k reconstruction in the version of PAGES

5 2k Consortium (2013) and the uncerlying proxies are available from https://doi.org/10.1038/ngeo1797 or alternatively https://www.ncdc.

noaa.gov/paleo-search/study/14188 (both last accessed, 21 May 2019). The Euro 2k reconstructions of Luterbacher et al. (2016) can be found at https://www.ncdc.noaa.gov/paleo-search/study/19600 (last accessed 21 May 2019). Data for assessing the response to volcanic eruptions from Sigl et al. (2015) is available from https://doi.org/10.1038/nature14565 (last accessed 21 May 2019). We use version CRU TS 3.10 of the observational CRU-data (Harris et al., 2014; University of East Anglia Climatic Research Unit et al., 2017), which has subsequently been superseded. The current version CRU TS 4.01 is available at http://doi.org/10/gcmcz3 with further information also given at https://crudata.uea.ac.uk/cru/data/hrg/ (last visited 20 September 2018). The Berkeley Earth project data (BEST Muller et al., 2013) can be obtained from http://berkeleyearth.org/ (last accessed, 22 May 2019). Relevant results of the present study will be uploaded to the Open Science Framework at https://osf.io/embdh/.

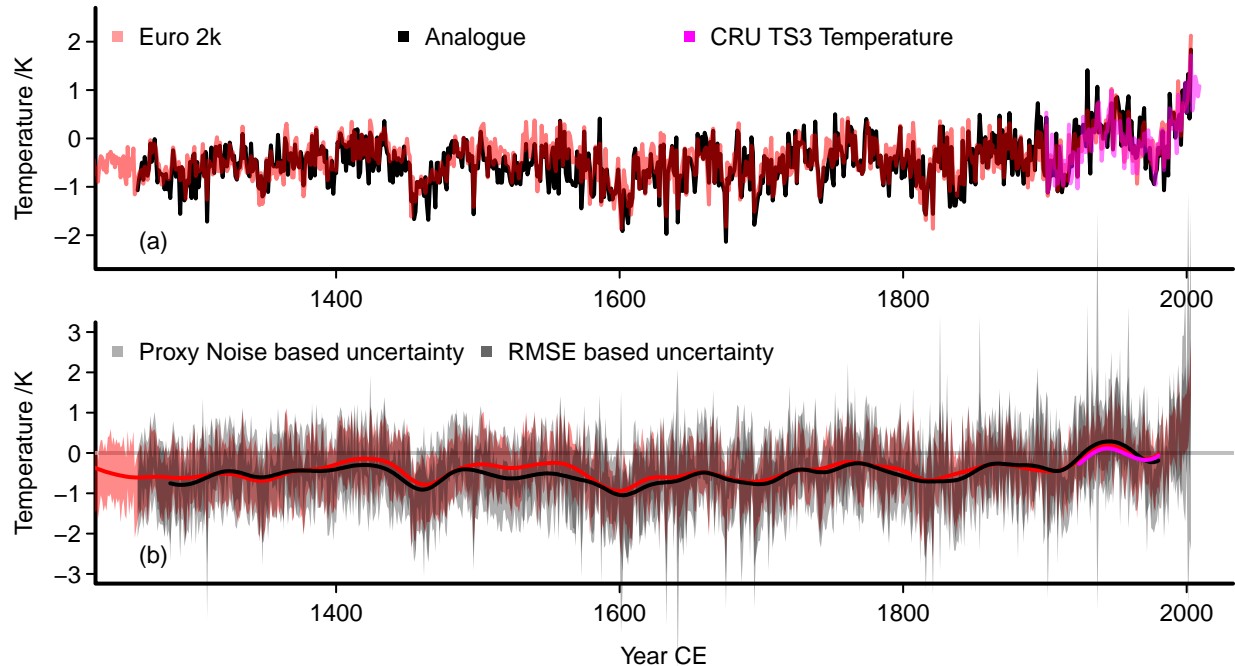

**Figure A1.** The best-analogue reconstruction as in Figure 2 but relative to the observational period 1901-2003 CE: **(a)** the interannual temperature reconstruction in black, the red line is the area mean Euro 2k-reconstruction, magenta is the observational CRU temperature adjusted to the mean of the reconstruction over its time-range. The analogue reconstruction is rescaled by the variability from one of the simulations. **(b)**: as (a) but for 47-point Hamming filtered data; we further add the uncertainty for estimates for the interannual data: red is the unsmoothed Euro 2k-uncertainty, the lighter grey envelope is a 2 standard-deviation uncertainty based on the correlation between the the proxies and the observations at the proxy locations, the darker grey envelope is a 2 standard-deviation uncertainty based on a MSE-estimate. Panel (b) adds a zero line as visual assistance.

## Appendix A: Additional Figures

This appendix provides a number of additional Figures to assisst the comparison of our reconstructions and our uncertainty estimates to previously published work.

Figure A1 shows the results from Figure 2 but relative to the climatology of the period 1901 to 2003 CE instead of the period 1260 to 2003 CE. Similarly Figure A2 highlights differences in the evaluation of the Euro 2k-reconstruction and our single best analogue reconstruction relative to the observational CRU-TS data (Harris et al., 2014).

Figure A3 adds informations on the two reconstructions for the European sector published by Luterbacher et al. (2016). The composite plus scaling reconstruction of Luterbacher et al. (2016) shows very small differences to the equivalent data of PAGES 2k Consortium (2013).

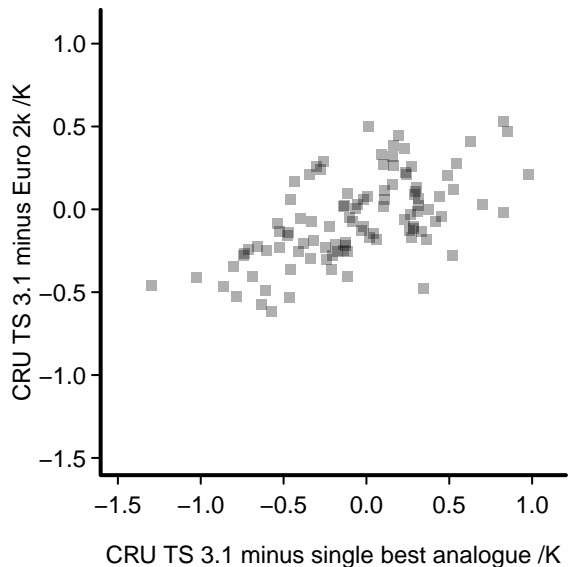

**Figure A2.** Differences between the CRU TS data and the Euro 2k reconstruction plotted against the differences between the CRU TS data and the single best analogue reconstruction.

Finally Figures A4 and A5 supplement Figure 13. Where Figure 13 compares the local analogue data of the fixed one standard deviation approach to the regional series of the BEST dataset, Figure A4 does so for the approach using a fixed number of analogues and Figure A5 provides the equivalent comparison for the single best analogue data.

## Appendix B: External code

This paper uses a nunber of external software packages. These include the Climate Data Operators (cdo, https://code.mpimet.mpg.de/project last accessed 27 November 2019). RStudio (RStudio Team, 2018) was essential. The following R (R Core Team, 2019) packages found a use: ncdf (Pierce, 2015), ncdf4 (Pierce, 2019), pracma (Borchers, 2019), caTools (Tuszynski, 2019), zoo (Zeileis and Grothendieck, 2005), dplR (Bunn, 2008), fields (Douglas Nychka et al., 2017), maps (code by Richard A. Becker et al., 2018), gtools (Warnes et al., 2018), astsa (Stoffer, 2017), psd (Barbour and Parker, 2014), knitr (Xie, 2015), kableExtra (Zhu, 2019), beeswarm (Eklund, 2016), RColorBrewer (Neuwirth, 2014), latex2exp (Meschiari, 2015), vioplot (Adler and Kelly, 2018), viridis (Garnier, 2018), pdist (Wong, 2013), foreach (Microsoft and Weston, 2017), doMC (Analytics and Weston, 2017), ddpcr (Attali, 2019), oce (Kelley and Richards, 2018), rticles (Allaire et al., 2019), and grateful (Rodriguez-Sanchez, 2017).

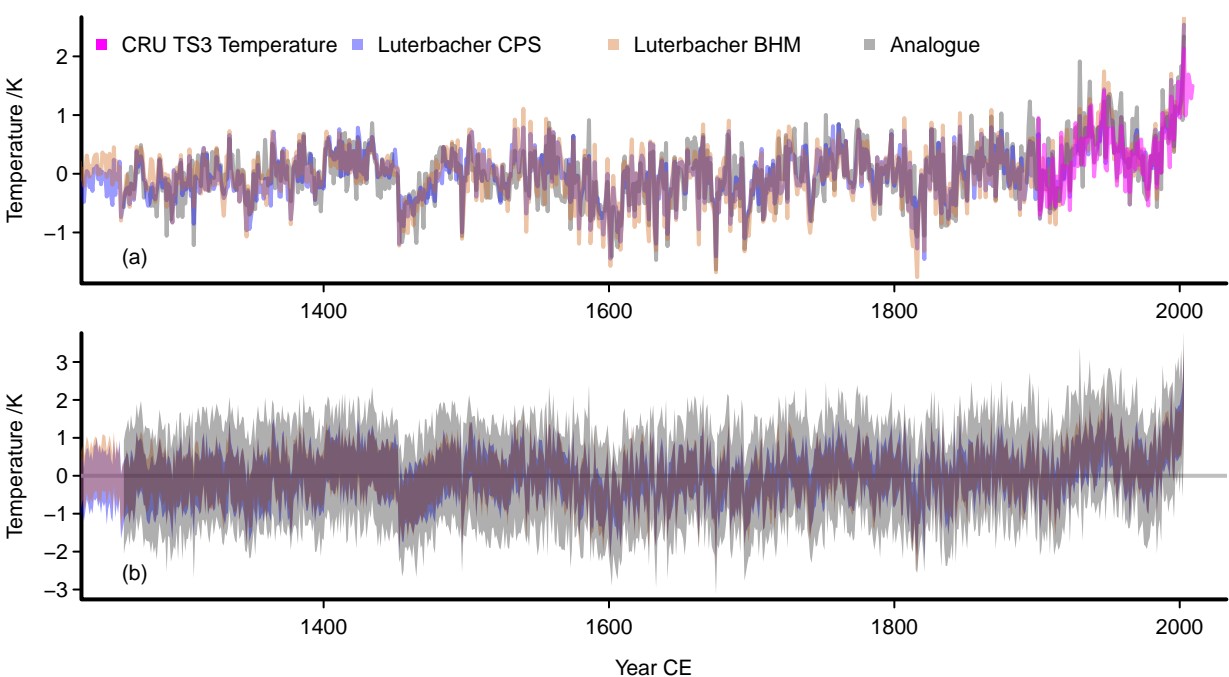

**Figure A3.** Comparison of the reconstructions of Luterbacher et al. (2016) to the best-analogue reconstruction (see also Figure 2): **(a)** the interannual single best analogue temperature reconstruction in grey; the blue is the composite plus scale (CPS) reconstruction of Luterbacher et al. (2016) and the brown line is the area mean of their Bayesian Hierarchichal Modelling reconstruction; magenta is the observational CRU temperature adjusted to the mean of the analogue reconstruction over its time-range. The analogue reconstruction is rescaled by the variability from one of the simulations. **(b)**: as (a) but for the 95% uncertainties of the reconstructed datasets. Differences between the Luterbacher CPS and the Euro 2k mean reconstructions are smaller than 0.005K. Panel (b) adds a zero line as visual assistance.

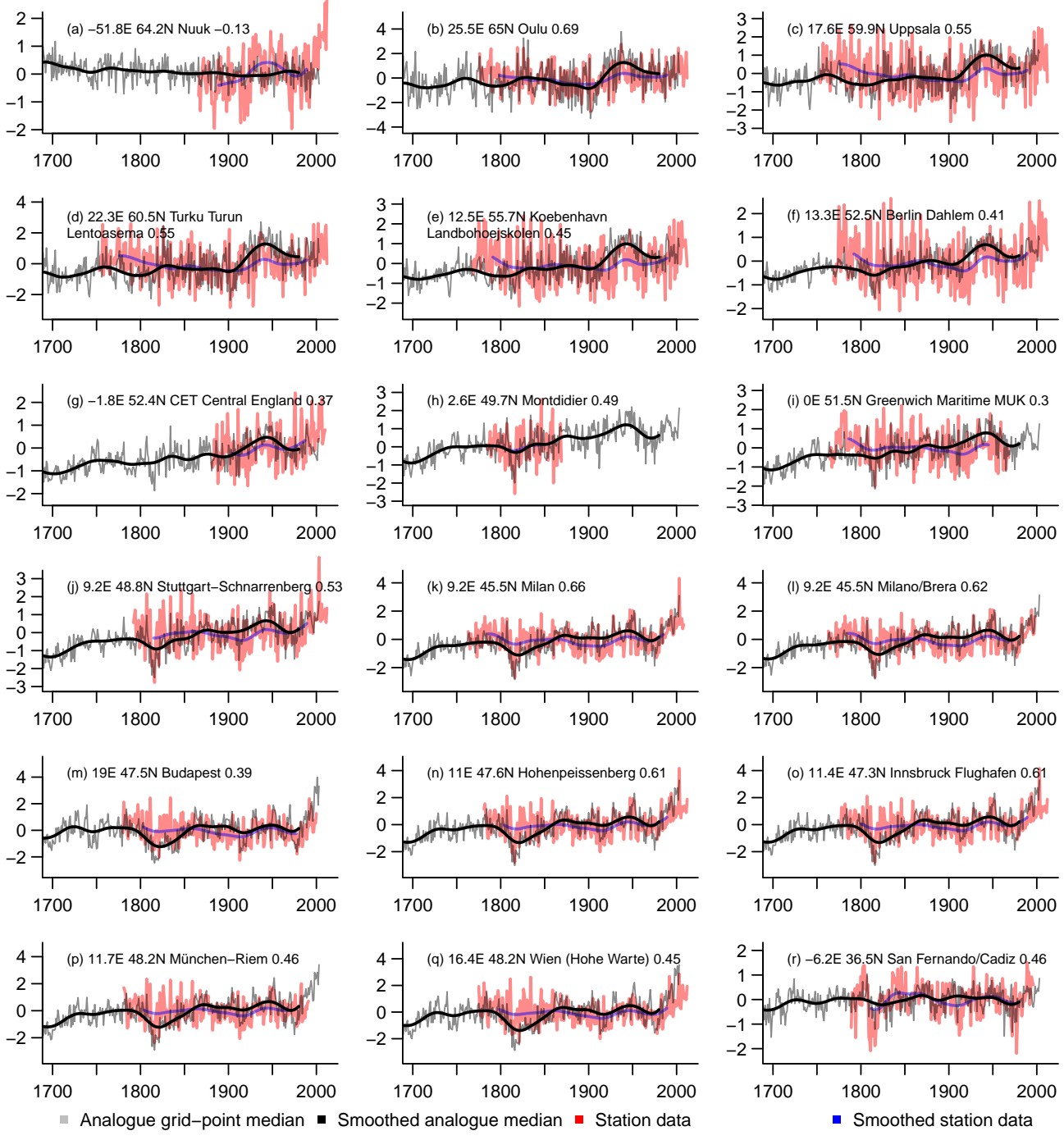

**Figure A4.** Comparison of local grid-point analogue data for the fixed number of analogues approach with an arbitrary selection of regionally representative data from BEST. Location, station name, and correlation over available station data are at the top of the panels. Grey and black, interannual and smoothed analogue median. Red and blue, station data and its smooth. X-axes are years CE. Y-axes are temperature anomalies in Kelvin relative to the period where both datasets are available.

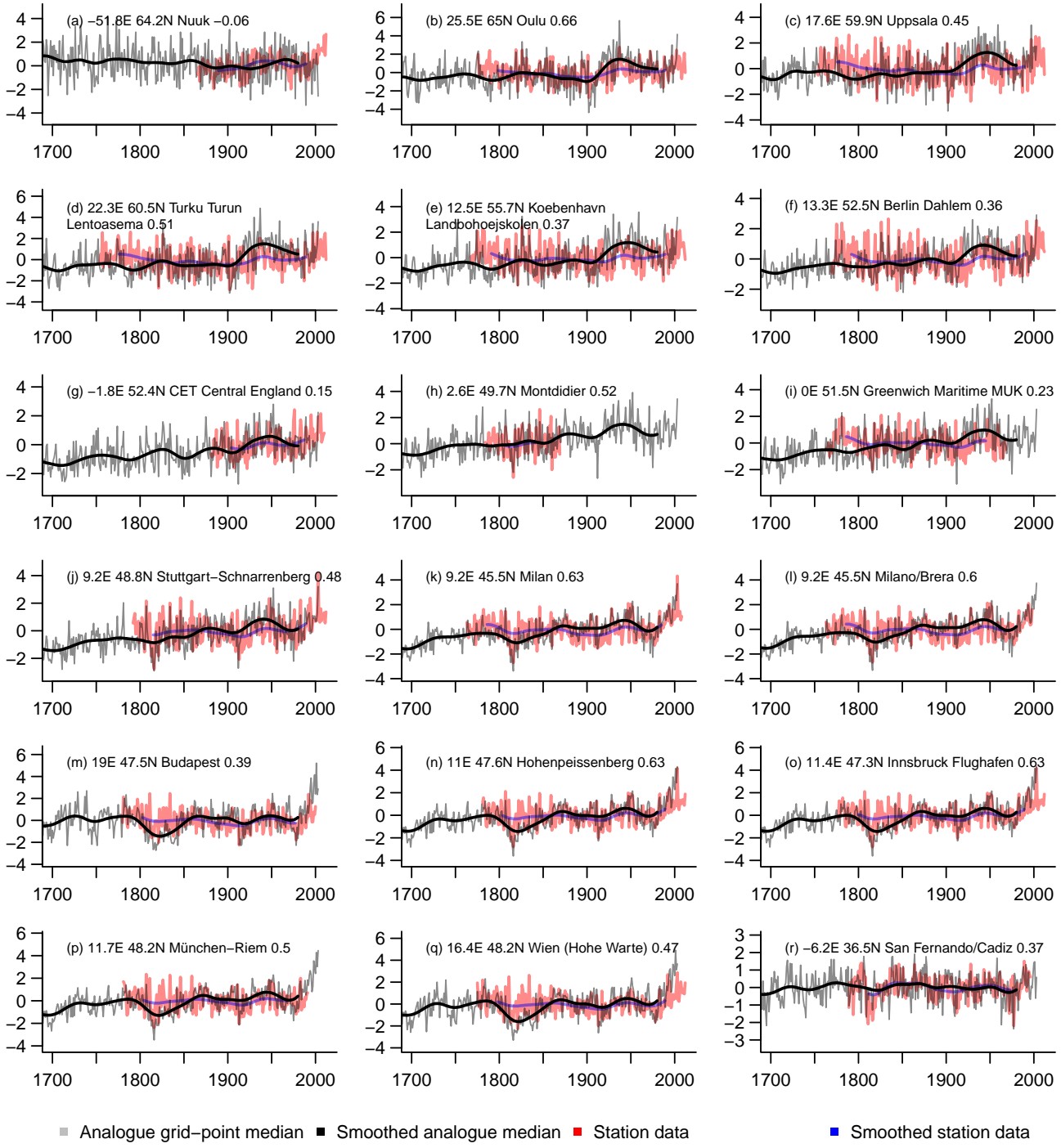

**Figure A5.** Comparison of local grid-point analogue data for the single best analogue with an arbitrary selection of regionally representative data from BEST. Location, station name, and correlation over available station data are at the top of the panels. Grey and black, interannual and smoothed analogue median. Red and blue, station data and its smooth. X-axes are years CE. Y-axes are temperature anomalies in Kelvin relative to the period where both datasets are available.

*Author contributions.* Oliver Bothe devised the analyses, performed them, and wrote the first draft. O.B. and Eduardo Zorita discussed the results and revised the manuscript.

25   *Competing interests.* The authors declare no competing interests.

*Acknowledgements.* Funding in the projects PRIME2 and PALMOD (www.palmod.de) made the completion of this study possible. This study is a contribution to PALMOD, and to the PAGES 2k Network, especially its PALEOLINK project. We acknowledge the service of the World Data Center for Climate in providing the simulation data and of the NOAA Centers for Environmental Information for providing the reconstruction data by Luterbacher et al. (2016) and PAGES 2k Consortium (2013).

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
