# Peer review of "Proxy surrogate reconstructions for Europe and the estimation of their uncertainties"

_Climate of the Past, 2019_

## Referee Comment (RC1) · Anonymous Referee #1 · 1 Aug 2019

Bothe and Zorita present a study where they investigate different ways of obtaining an uncertainty estimate for climate reconstructions using the analogue method, also known as the proxy surrogate reconstruction method. They authors describe the downside of single member reconstructions, and produce both single member and multi ensemble member reconstructions, which are compared to other reconstructions and observations. Then they go on to describe how an uncertainty can be assigned based on i) the fit of the analogue ii) assumptions on the standard deviation of the noise or iii) the ensemble spread when using a fixed number of ensemble members. Finally, the authors conclude on the pros and cons of the different approaches.

[Figure]

General comments.

This study is overall well executed, thorough and timely. However, the writing is somewhat uneven, especially in the introduction, which I have commented on in detail below, but please go through the entire manuscript as I might have run out of steam. I have few major comments about the methodology itself, but I am wondering if the method is overfitting the model data to the proxies (see specific comments below). If the writing is brushed up as well as taking my other comments into account, I think this work could be suitable for publication.

Specific comments.

P1, L1: Please rewrite this sentence. It is the combination itself that reconciles the two sources, so if this is possible "allows" is redundant. Also, if one is reconciled with the other then they are both reconciled, making "both" redundant.

P1, L3: "... to benefit from the advantages of both data sources" this is in a way a repetition from previous sentence. Why not say something about the technique? E.g. "The analogue or proxy surrogate reconstruction method is a computationally cheap data assimilation approach which samples a model ensemble based on the best match to proxy data".

P1, L9: Replace "had been" with "was"?

P1, L10: Remove "using"?

P1, L12-L14: "The approaches do not agree... " this sentence is not easy to read. Perhaps rewrite "However, the two approaches do not agree on the warmest preindustrial decades, which for the Euro 2k reconstruction is during the early 15th century, and for the analogue approach is during the early 18th century".

P1, L15: "The surrogate reconstructions..." I suggest that you early in the manuscript choose to call the reconstructions either surrogate or analogue, even if you have said it means the same - just to make it easier to read.

P1, L15: Insert comma before "but". Please use more commas to help the reader.

P1, L15-L16: Actually, I don't understand the sentence. You lose me around "even under uncertainty". Please rewrite.

P1, L20: Is "paleo-observations" the right word? Why not simply "proxy data"?

P2, L1: Replace "the search for" with "finding"?

P2, L7: Why "not only", maybe cut this?

P2, L14-L15: "The analogue method... " This sentence is hard to follow. Please rewrite.

P3, L1: Either write "Here we propose ..." or "In this study we provide...".

P3, L?? (something strange happens with the line numbers): "Here, we obtain... " (skip comma after "Here"). Please be clear on what is model and what is proxy. I suppose "pool of relevant candidate fields" is model out put and "local indices" is proxy data?

Figure 1: Please add units to axes.

P4, L15-L16: "That is, they use recent observations, which measured archives... ". I don't follow this sentence. Please rewrite.

P4, L19: "to more than one environmental condition" do you mean that a given proxy paramenter can depend on more than climate or environmental variable? Please clarify.

P4, L21-22: Is "environmental condition" the correct word, or is "climate state" more accurate.

P4, L26: I suggest you make a sheet or table with mathematical abbreviations that you use in the paper.

Table 1: So, the correlations are between the tree ring data series and the JJA CRU temperature. Please add these details to the caption. How do you deal with seasonality

of the proxy data? In Wilson et al. (2016) each proxy site is listed with different seasonal sensitivity to temperature (Table 1) and I believe you are using some of the same data.

P6, L13: "The last of the remaining eight proxy indices starts in 1260" meaning that all remaining records cover 1260 to 2003 CE?

P7, L16: "strong ensemble" or "8 member ensemble"?

P7, 2nd paragraph, L8: "Since the current manuscript is not least a proof of concept... " this formulation sounds off.

Figure2: (a) rescaled temperature? Please specify which scaling is used in the caption, so you don't have to look for it. Euro 2k is an area mean? It's hard to see that the difference to the CRU temp. Can you show this in (c)? Luterbacher et al. (2016) is discussed a lot in relation to Figure 2, it would be helpful if you show this data as well.

P9, L17: "calculated as the square root ..." why not write out the equation?

P9, c. L25: So which uncertainty is realistic? And why?

P9, L29-L30: "The coldest century was until 1648 CE in the best-analogue reconstruction but until 1678 CE in the Euro 2k record" please write the interval of the coldest century. This formulation is unclear.

P10, L1-L? (again random line numbers): When discuss interval please write them out instead of just giving the end year. It's much easier to read. Just write "the warmest century was 1353-1452 CE".

P10: About the volcanic analysis. Did you look at high latitude eruptions, e.g. Laki? How did you do the super imposed epoch analysis? Maps of field anomalies, or time series? How did you define the reference period before the eruptions?

P10, 2nd paragraph, L7-L8: "Interestingly, the analogues even appear to occasionally capture the relation between the proxies included and those excluded" couldn't this be completely random? Then it's not very interesting.

P10, 2nd paragraph, L10: Replace "1947. Then" with "1947, where".

Figure 4: What are the numbers next to the site name e.g. "(a) Tor92 0.91"? Is 0.91 the correlation?

Figure 5: Again, what are the numbers next to the site names?

P15, L4: Correlations "between 0.84 and 0.98" for proxies and and reconstructed temperature. These correlations are a good bit higher on average than the data in Table 1. Are you overfitting the data, or how can you explain this? Wouldn't you need forward proxy modeling of tree growth to give a more realistic link between model and proxy data (e.g. Tardif et al. 2019)?

P21, L8: Is it really "strange variability" since the reconstruction is unconstrained in Greenland?

References.

Tardif, R., Hakim, G. J., Perkins, W. A., Horlick, K. A., Erb, M. P., Emile-Geay, J., Anderson, D. M., Steig, E. J., and Noone, D.: Last Millennium Reanalysis with an expanded proxy database and seasonal proxy modeling, Clim. Past, 15, 1251-1273, https://doi.org/10.5194/cp-15-1251-2019, 2019.

Wilson, R., Anchukaitis, K., Briffa, K.R., Büntgen, U., Cook, E., D'Arrigo, R., Davi, N., Esper, J., Frank, D., Gunnarson, B., Hegerl, G., Helama, S., Klesse, S., Krusic, P.J., Linderholm, H.W., Myglan, V., Osborn, T.J., Rydval, M., Schneider, L., Schurer, A., Wiles, G., Zhang, P., Zorita, E., 2016. Last millennium northern hemisphere summer temperatures from tree rings: Part I: the long term context. Quat. Sci. Rev. 134, 1e18. http://dx.doi.org/10.1016/j.quascirev.2015.12.005.

---

## Referee Comment (RC2) · Anonymous Referee #2 · 6 Aug 2019

This study is an interesting contribution to the field of climate reconstruction because it adds and compares multiple way of uncertainty estimation to the widely and successfully used analog reconstruction methodology. It fits very well to the scope of Climate of the Past. Hence, I suggest publication after revisions that should make the structure clearer, condense the results including figures with time series and after putting the focus a bit more on the novelty of the uncertainty estimation than on the reconstruction.

Comments:

Introduction

- Would be worth mentioning the just published global reconstruction by Neukom et al.

[Figure]

2019, which includes an analog approach, too.

- I would find a list of the content helpful at the end of the introduction, saying that three approaches are tested: 1. best analog only, . . .

- Page 2, line 20: "guestimate" is colloquial language

Methods

- The entire structure of the study and the used error estimation should be made clearer. Can you add a schematic diagram?

- Explain clearly how you come to your three reconstruction experiments, into which the results are separated. I assume the number 39 for the minimal number of analogs in 2003 (page 5, line 20) is the reason for having 39 in section 3.2 but that is not clear to the reader.

- Page 4, line 23: "under certain assumption" Which assumptions? Please write more precise.

- Page 4, last two paragraphs: I would rather put the equations more prominent in separate lines and not in the middle of the sentence because understanding the error estimation is crucial for this study.

- Page 5, line 6: modified

- Page 5, line 15: "dates" You have not mentioned yet that you reconstruct JJA averages at annual resolution

- Page 5, line 15ff.: How do you choose the noise SD levels such as 2.57? And why if you write in line 22 that only the 1 SD criterion gives a reasonable number of analogs?

Proxies

- Is there a reason to use the gridded CRU data for proxy correlations here and the BEST data later in the paper?

- You could explain that the correlation of the excluded location in Slovakia are low because trees are limited to temperatures in another season. Otherwise it seems strange, why they appear in the PAGES data base. However, I am not sure why the Albania chronology with weakly significant negative correlation appears in the PAGES collection. Maybe, it has been removed in the more strictly screened version 2 of the data base? Having this negative correlation in mind, I do not understand why it is used for comparison/verification later in the paper? I would not expect a good match/positive correlation in the analog reconstruction.

Model simulations

- Page 7, line 5: Please explain again briefly why the "similar internal variability" of the simulation is important instead of referring to the previous section

Results

- Generally, try to shorten the results section and have a clear and consistent structure for the three experiments. I would put more focus on the uncertainty results than the reconstruction itself.

- Make clearer, how the three experiments compare and later in the discussion what we can learn from this.

- Page 9, line 2: why is the plot relative to Euro-2k and not relative to instrumental data?

- Fig. 3: It is not surprising that the analogs fit better, where you have spatial proxy clusters than isolated locations. I have not seen this discussed in the paper.

- Page 11, line 26: It is good that the analog reconstruction generally agrees with previous statistical reconstructions but they are not a reference and it is unclear which ones are closer to reality. Rather just see if they are in your uncertainty range.

- Page 14, line 30ff: Have you considered weighting the analogs with respect to their distance?

- Page 15, line 5: "visually there is good agreement" Not clear what you are talking about, other series besides Tatra and Albania?

- Page 15, line 29: Add reference to figure

- Page 15, line 35: How can you have a stronger 20th century warming trend in the reconstruction than in observations and at the same time have trouble to find analogs for exceptionally warm years such as 2003?

- Page 16, line 9: If you look at the temperature evolution after individual eruptions, this is not a superposed epoch analysis.

- Page 19, line 14 and Fig. 8: Why do you show a mean and not the median in this case? The mean should be influenced by the number of averaged analogs and the numbers are highly different in this case.

- Page 19, line 28ff: Why is the comparison with instrumental data just done for the 1 SD reconstruction?

Concluding remarks

- Please avoid 1-sentence paragraphs

Figures

- Generally, please think of a way to reduce the number of figures with time series. Both, the number of really necessary panels in each figure and figures in total. E.g. it is probably not required to see the annual resolution reconstruction for the full period for all three experiments or do multiple smoothing have to be presented?

- I find the uncertainty ranges often impossible to see (e.g. Fig. 2a). I cannot recognize the "envelope", you are talking of. As this a main focus of the paper, please try to find a way to plot uncertainty better visible, e.g. just a smoothed version for the entire period and a subperiod at annual resolution.

---

## Referee Comment (RC3) · Anonymous Referee #3 · 23 Aug 2019

**Abstract**

This study proposes a new climate reconstructions for Europe for nearly the full last millennium. The approach is based on the Analog Method, also known in the literature as Proxy Surrogate Reconstruction. One of the main novelties of this manuscript is how the authors extend the methodology to explicitly account for uncertainties. The authors present several reconstructions and compare them to the Euro 2k reconstruction, as well as independent data from the BEST project. Similarities, differences, advantages and caveats are discussed through the manuscript.

[Figure]

**General comment**

Most classical reconstruction methods produce a single reconstruction which does not explicitly account for uncertainty, although it is acknowledged that it populates this type of data-sets. This is problematic because uncertainty is not only ubiquitous, but it is heterogeneous both in time and space. This is an important limitation that precludes the proper assessment of the limitations of the knowledge we can gather from climate reconstructions. In this sense, I think this study is important and necessary to improve one prominent tool to produce such reconstructions, the Analog Method.

The design of the study is sensible, and I have mostly minor comments regarding details I could not fully understand and therefore might deserve clarification. Should not be for the issue I discuss below, I would recommend publication after minor revision.

There is however and important aspect that has to be improved in the manuscript under the light of very recent bibliography published even after this discussion was started. There exists a published extension to the Analog Method that allows to estimate uncertainties. This is part of a recent publication with a more general aim (Neukom et al., 2019). There, authors briefly introduce and apply a methodology which largely differs from the one presented here, but that aims at the same purpose: explicitly assess uncertainties in climate field reconstructions with the Analog Method. I think this work should somehow account for the existence of this already published method. The level of modification applied to the manuscript depends on the authors. At the minimum, the differences between approaches should be discussed (for example, the approach opted by Neukom et al. (2019) does not produce missing values, being in principle an important advantage). At best, the approach adopted by Neukom et al. (2019) could be implemented here as well, and a comparison could be done between both methods. In my opinion, the latter would greatly improve the interest of this manuscript, but it is perhaps a major modification of the work that falls beyond its original scope. I leave it up to the authors and I would not be disappointed if they decide not to tackle this task.

**Minor comments**

1. Page 2, Line 20: I think the correct citation is Gómez-Navarro et al. (2014)

2. Fig 1: Maybe excluded locations could be shown with grey symbols, as well as the are representative for Central Europe. The location of these proxies is relevant for example to understand Figure 5.

3. Page 4, Lines 28-29. I think it is more correct to say that, only when $\text{Var}_{res}$ and $\text{Var}_{sig}$ are uncorrelated, the total variance is the sum of both (because in that case the covariance term vanishes).

4. Page 5, Line 6: typo (modfied)

5. Page 5, Lines 17–21: I do not understand where the 2.57 comes from. How it is related to the minimum number of 39 proxies? Please clarify.

6. Page 5, Lines 22-23: why is it the only one? why 2105 is special? why not 1.5 $\text{SD}_{noi}$?

7. Overall, in the two paragraphs aforementioned, it lies the core of the two reconstructions carried out. I think this is important, and it should be made more explicit that the two approaches represent different method used for real below. Perhaps this can be made more explicit with some structural element, such as an un-ordered list or similar.

8. Page 6, Table 1: I assume this is exactly the correlation used to define the $\text{SD}_{noi}$ in each proxy location, right? If so, this could be clarified in the main text (especially in Section 2.1.2).

9. Page 6, lines 6–8: The criterion to exclude two proxies is not very clear. What is meant by "relevant portion of variance"? In Fig. 5 we learn that the reconstruction

in these sites is poor. Would it be better if these sites were part of the network. Surely the answer is yes. I understand that the amount of climate information we get is poorer than in the other locations, but still we could benefit for having *some* information. At worst, if the proxies were pure noise, it would not be necessarily worse than not having information at all. In other words, I think having poor information is better than having none, and it's not fully obvious to me why proxies should be excluded from the analysis based on relatively low correlation alone.

10. Page 6 (but relevant for the whole study): why do you restrict the reconstruction to the period 1260 to 2003? The reconstruction could have been applied further back in time. The number of proxies varies in time, but this could be even beneficial for this study, focused on the validation of new methodologies. It would show how the estimates of the uncertainty presented here are sensible to a varying number of proxies. I feel that this choice has unnecessarily limited the scope of the manuscript.

11. Page 7, Table 2: it could be interesting to write the total number of analogues, i.e. the pool size. It would make more meaningful the number of proxies used to produce ensembles. For example, having 817 analogues (as in Fig. 8) has a clearer meaning when you add that they are 817 out of, let's say, 25000. It shows that you are still selecting a relatively minor number of relatively good analogues.

12. Page 7, Lines 6–7: I think having a consistent bias through the pool is not necessarily good, as it seems to be implied by the wording. It ensures that the bias are translated into the reconstruction. This is partly avoided using structurally different models to build the pool. I do not mean that the authors should necessarily rebuild the reconstruction with a larger set of models, but I think that at least they should not imply that using a single model is somehow beneficial.

13. Page 8, Fig. 2: I think a line marking the 0 K anomalies would help to read the series. This pertains mostly panels b and c, where the sign of the anomaly is

important, but difficult to appreciate without such a line. This argument applies to Figs 6 and 7 as well.

14. Page 8, Fig. 2: It's not fully clear to me what this figure (as well as Figs 6 and 7) show. Does "summary" mean spatial average?

15. Page 9, Line 10: please change "degree Kelvin" to "Kelvin". Please review it, as there are other locations where I saw this in the manuscript.

16. Page 9, Lines 9–16. The order of these two paragraphs can be exchanged. It's a bit unusual and therefore confusing to discuss Fig. 2c before Fig. 2b.

17. Page 10, Lines 10–15: I think the fact that the reconstruction underestimate the intra-location variability is a problem of the pool, not the Analog Method itself. Do the authors think that this could be improved if higher resolution models were used to build the pool?

18. Page 11, Fig. 3: The list of locations in the caption is misleading (the name and the ID are written all together). It seems a detail, but it puzzled me for a while until I realised that Tor92 and Torneträsk are not two proxies, but the ID and the name of the same one. You could easily remove this by using for instance parenthesis to separate name from ID or vice versa.

19. Page 11, Line 26: "The general agreement between the Euro 2k and the analogue..." this reads odd at this point, as the reader does not know where to find the information the authors are referring to. It turns out that this comparison is introduced later, in Figure 6 in Page 14.

20. Page 15: Lines 17–23: The reduced variance could be quantified (how much is notably smaller variance in Line 19?). Further, the lost of variance when more analogues are considered is common in this approach, and generally in any statistical approach, i.e. there is a bias-variance trade off. It could be noted here

that this has been comprehensively discussed in the bibliography of the Analog Method.

21. Page 15, Line 32: do the authors have a theory on what could be the reason for such systematic differences? Are they meaningful, can they be used to discuss merits or problems in the reconstructions? Or are they rather low-frequency random fluctuations highly sensitive to method parameters?

22. Page 16: Lines 25–26: The presence of missing values in years with volcanic eruptions is a major caveat of the method, as those are typically the years most interesting in climate studies. Here it would be specially relevant my comment about a comparison with the method presented by Neukom et al. (2019).

23. Page 16, Lines 27–31: I do not see why it is "unsurprising" this lack of analogues for the recent period. The pool contains this warming as well, so the search should not present more problems for this period than in any other.

24. Page 19, Lines 18–20: Maybe I'm miss-evaluating this, but I think that anchoring the reconstruction within a range of 8 K is a poor result. It shows that the 800 analogues are indeed poorly constrained in this region, so we have little idea of how the actual climate was in that period and region. More generally, I have the concern that the spread shown for example in Fig. 7 might provide an optimistic measure of the actual uncertainty. Fig 7e for instance shows the range in the spatial average, which is about 2 K. But this is after spatial average, where regional differences can cancel out! I wonder how large is the range in each location. This might perhaps be illustrated with a map of (temporally averaged) ranges? Eventually, my guess is that using as many as 800 analogues or more, really far away from "the best" is, as outlined by the authors, too much.

**References**

Neukom, R., Steiger, N., Gómez-Navarro, J. J., Wang, J., and Werner, J. P.: No evidence for globally coherent warm and cold periods over the preindustrial Common Era, Nature, 571, 550, https://doi.org/10.1038/s41586-019-1401-2, https://www.nature.com/articles/s41586-019-1401-2, 2019.
* * *

---

## Author Comment (AC1) · 8 Oct 2019

Dear referees, dear editor,

We want to thank the referees and the editor for evaluating our manuscript and providing such encouraging comments.

Below we respond to the reviewer comments. We will modify the manuscript to shift the focus towards the uncertainties. We yet are undecided whether we explicitly include a bootstrap uncertainty estimate similar to Neukom et al. (2019).

We are sorry that we took so long for our final response.

On behalf of the authors

Sincerely yours,

Oliver Bothe

**Editor**

*Please for the revisions, you might add in the paper, that compared to PAGES2k, Luter-bacher et al. 2016 excluded the Tatra and Albania proxies from their analysis as they lack significant correlations with European summer temperature variability.*

**Response** We make this change and add: Already Luterbacher et al. (2016) noted this and, therefore, did not consider these two proxies in their reconstruction effort.

**Referee 1**

*Bothe and Zorita present a study where they investigate different ways of obtaining an uncertainty estimate for climate reconstructions using the analogue method, also known as the proxy surrogate reconstruction method. They authors describe the down-side of single member reconstructions, and produce both single member and multi ensemble member reconstructions, which are compared to other reconstructions and observations. Then they go on to describe how an uncertainty can be assigned based on i) the fit of the analogue ii) assumptions on the standard deviation of the noise or iii) the ensemble spread when using a fixed number of ensemble members. Finally, the authors conclude on the pros and cons of the different approaches.*

**General comments.**

*This study is overall well executed, thorough and timely. However, the writing is some-what uneven, especially in the introduction, which I have commented on in detail below, but please go through the entire manuscript as I might have run out of steam. I have few major comments about the methodology itself, but I am wondering if the method is overfitting the model data to the proxies (see specific comments below). If the writing is brushed up as well as taking my other comments into account, I think this work could be suitable for publication.*

**Response:** We thank the reviewer for their positive evaluation.

Our revisions try to improve on the writing.

Regarding the overfitting: See our response below.

**Specific comments.**

*P1, L1: Please rewrite this sentence. It is the combination itself that reconciles the two sources, so if this is possible "allows" is redundant. Also, if one is reconciled with the other then they are both reconciled, making "both" redundant.*

**Response:** We change this to: Combining proxy information and climate model simulations reconciles these sources of information about past climates.

*P1, L3: ". . . to benefit from the advantages of both data sources" this is in a way a repetition from previous sentence. Why not say something about the technique? E.g. "The analogue or proxy surrogate reconstruction method is a computationally cheap data assimilation approach which samples a model ensemble based on the best match to proxy data".*

**Response:** We are not convinced that this simply repeats the content of the previous sentence but follow the suggestion of the referee: The analogue or proxy surrogate reconstruction method is a computationally cheap data assimilation approach, which searches in a pool of simulated climate states the best fit to proxy data.

*P1, L9: Replace "had been" with "was"?*

**Response:** We change this accordingly.

*P1, L10: Remove "using"?*

**Response:** We change this accordingly.

*P1, L12-L14: "The approaches do not agree. . . " this sentence is not easy to read. Perhaps rewrite "However, the two approaches do not agree on the warmest preindustrial decades, which for the Euro 2k reconstruction is during the early 15th century, and for the analogue approach is during the early 18th century".*

**Response:** We change this to read: However, the approaches disagree on the

warmest preindustrial decade, which is in the early 15th century for the Euro 2k reconstruction and in the early 18th century for the analogue approach.

*P1, L15: "The surrogate reconstructions. . . " I suggest that you early in the manuscript choose to call the reconstructions either surrogate or analogue, even if you have said it means the same - just to make it easier to read.*

**Response:** We try to be consistent in our writing. Therefore, we here change the sentence to: The reconstructions from the analogue method . . .

*P1, L15: Insert comma before "but". Please use more commas to help the reader.*

**Response:** We try to use more commas to ease reading the manuscript.

*P1, L15-L16: Actually, I don't understand the sentence. You lose me around "even under uncertainty". Please rewrite.*

**Response:** We change this to read: The reconstructions from the analogue method also represent the local variations of the observed proxies. Local uncertainties of the temperature reconstructions tend to be large in areas that are poorly covered by the proxy records.

*P1, L20: Is "paleo-observations" the right word? Why not simply "proxy data"?*

**Response:** We regard it to be the right expression but change the occurrences according to the referee's suggestion.

*P2, L1: Replace "the search for" with "finding"?*

**Response:** We replace this with "searching".

*P2, L7: Why "not only", maybe cut this?*

**Response:** There are further applications. Therefore, we would like to keep the sentence as it is.

*P2, L14-L15: "The analogue method. . . " This sentence is hard to follow. Please*

*rewrite.*

**Response:** We rewrite this: The analogue method is a computationally cheap means to contrast information from simulations and reconstructions in the sense of data assimilation. However, it is methodologically less sophisticated than full data assimilation procedures (compare, e.g. Tardif et al., 2019).

*P3, L1: Either write "Here we propose ..." or "In this study we provide...".*

**Response:** We rewrite: Here we propose a reconstruction ...

*P3, L?? (something strange happens with the line numbers): "Here, we obtain... " (skip comma after "Here"). Please be clear on what is model and what is proxy. I suppose "pool of relevant candidate fields" is model out put and "local indices" is proxy data?*

**Response:** We are sorry for the random line numbers. We use the RMarkdown template and under certain unclear conditions this happens, but can easily be repaired, which we unfortunately did not do.

We rewrite: Here we obtain large-scale fields of summer temperature based on a pool of relevant candidate fields from model simulations and a set of local proxy data records as predictors for the period 1260 to 2003 of the Common Era (CE).

*Figure 1: Please add units to axes.*

**Response:** We clarify the Figure.

*P4, L15-L16: "That is, they use recent observations, which measured archives... ". I don't follow this sentence. Please rewrite.*

**Response:** We rewrite: That is, they use recent observations made on properties of paleo-archives. These archives, in turn, recorded the past environmental conditions...

*P4, L19: "to more than one environmental condition" do you mean that a given proxy*

*paramenter can depend on more than climate or environmental variable? Please clarify.*

**Response:** We rewrite: ... that the archives recorded signals from more than one climate or environmental variable (e.g. temperature and precipitation ...

*P4, L21-22: Is "environmental condition" the correct word, or is "climate state" more accurate.*

**Response:** We think our choice of words is valid, but we rewrite: Correlations provide a simple measure of the relation between proxy-observations and the climatic environment over a period when reliable (instrumental) observations of the climatic variability exist.

*P4, L26: I suggest you make a sheet or table with mathematical abbreviations that you use in the paper.*

**Response:** We add another table.

*Table 1: So, the correlations are between the tree ring data series and the JJA CRU temperature. Please add these details to the caption. How do you deal with seasonality of the proxy data? In Wilson et al. (2016) each proxy site is listed with different seasonal sensitivity to temperature (Table 1) and I believe you are using some of the same data.*

**Response:** 1. We add the details to the caption. 2. As we compare our reconstruction to the Euro 2k reconstruction, we consider the seasonal attribution as used by the publications for this reconstruction. That is, we do not test whether the relation of the proxies is strongest for summer. For the attribution to summer compare Luterbacher et al. (2016) and PAGES 2k Consortium (2013).

*P6, L13: "The last of the remaining eight proxy indices starts in 1260" meaning that all remaining records cover 1260 to 2003 CE?*

**Response:** Yes. We clarify this: The latest start date of any of the remaining eight

proxy indices is the year 1260 CE. Thus, all eight records cover the period 1260 to 2003 CE.

*P7, L16: "strong ensemble" or "8 member ensemble"?*

**Response:** We clarify: there exists a multi-model ensemble of simulations for the last 1100 years. A number of additional simulations comply with the PMIP3 protocol but are not included in the effort

*P7, 2nd paragraph, L8: "Since the current manuscript is not least a proof of concept... " this formulation sounds off.*

**Response:** We remove the sentence.

*Figure2: (a) rescaled temperature? Please specify which scaling is used in the caption, so you don't have to look for it. Euro 2k is an area mean? It's hard to see that the difference to the CRU temp. Can you show this in (c)? Luterbacher et al. (2016) is discussed a lot in relation to Figure 2, it would be helpful if you show this data as well.*

**Response:** 1. We clarify the rescaling in the caption. 2. Euro 2k is an area mean. 3. We are unclear what the referee is referring to with respect to the CRU data. 4. Since the CRU data are only shown for qualitative comparisons, we rather would not do this. However, we did add such a comparison to the respective panel. 5. We add an Appendix to show additional Figures including one plotting the data by Luterbacher et al. (2016).

*P9, L17: "calculated as the square root ..." why not write out the equation?*

**Response:** We thought it was clear enough, but now show the equation.

*P9, c. L25: So which uncertainty is realistic? And why?*

**Response:** We shortly describe the characteristics of both uncertainty estimates. Both describe realistically part of the uncertainty.

*P9, L29-L30: "The coldest century was until 1648 CE in the best-analogue reconstruction but until 1678 CE in the Euro 2k record" please write the interval of the coldest century. This formulation is unclear.*

**Response:** We clarify the sentence.

*P10, L1-L? (again random line numbers): When discuss interval please write them out instead of just giving the end year. It's much easier to read. Just write "the warmest century was 1353-1452 CE".*

**Response:** We clarify this.

*P10: About the volcanic analysis. Did you look at high latitude eruptions, e.g. Laki? How did you do the super imposed epoch analysis? Maps of field anomalies, or time series? How did you define the reference period before the eruptions?*

**Response:** We only considered tropical eruptions. We clarify this: We consider a subselection of tropical eruption events in 1286, 1345, 1458, 1601, 1641, 1695, 1809, and 1815. We performed a superposed epoch analysis but do not show the results. We considered fields and area averages. We chose the five calendar years before an eruption year as reference period, which is a common approach (compare, e.g. Sigl et al., 2015).

*P10, 2nd paragraph, L7-L8: "Interestingly, the analogues even appear to occasionally capture the relation between the proxies included and those excluded" couldn't this be completely random? Then it's not very interesting.*

**Response:** We modify this: The analogues even appear to occasionally capture the relation between the proxies included and those excluded, which obviously might be by chance.

*P10, 2nd paragraph, L10: Replace "1947. Then" with "1947, where".*

**Response:** We do so.

*Figure 4: What are the numbers next to the site name e.g. "(a) Tor92 0.91"? Is 0.91 the correlation?*

**Response:** Yes. We clarify the caption.

*Figure 5: Again, what are the numbers next to the site names?*

**Response:** We again clarify the caption by mentioning that these are the correlations.

*P15, L4: Correlations "between 0.84 and 0.98" for proxies and and reconstructed temperature. These correlations are a good bit higher on average than the data in Table 1. Are you overfitting the data, or how can you explain this? Wouldn't you need forward proxy modeling of tree growth to give a more realistic link between model and proxy data (e.g. Tardif et al. 2019)?*

**Response:** The correlations are between the proxy locations and the medians of 39 analogues. Indeed they are high, but we would hope that the proxies included in our search constrain the search effectively and give good reconstruction results. This holds especially for the median, which is a filter for the data of the reconstruction ensemble members. The aim of data assimilation is to match the observations, i.e. the proxies, closely with the simulated data. Nevertheless, we are unable to exclude the possibility that our data constrains the pool too much and therefore may fail in a prediction excercise.

These correlations indicate only agreement with the proxy records not necessarily with the true temperature. Anyway, locally high correlations do not indicate high skill elsewhere. Indeed, correlations with the observational CRU data are in line with the correlations between the proxies and the CRU data as one may expect from these high correlation coefficients. The comparison to the BEST-data shows, this does not necessarily reflect on how well the reconstruction captures the observed temperature elsewhere.

Regarding the use of proxy system models: An optimal approach would incorporate a

calibrated proxy system model to pre-process the simulation data. Indeed, any reconstruction approach can benefit from pre-processing data with calibrated proxy forward models.

Regarding overfitting, there are no parameters in the analog-setting that are calibrated for a better fit to the predictand. The number of analogs chosen or the distance metric chosen are not optimized for a better fit.

*P21, L8: Is it really "strange variability" since the reconstruction is unconstrained in Greenland?*

**Response:** We clarify this: The top-left panel for Nuuk highlights that the lack of constraints on the reconstruction can result in potentially artificial spikes in the time-series.

**Referee 2**

*This study is an interesting contribution to the field of climate reconstruction because it adds and compares multiple way of uncertainty estimation to the widely and successfully used analog reconstruction methodology. It fits very well to the scope of Climate of the Past. Hence, I suggest publication after revisions that should make the structure clearer, condense the results including figures with time series and after putting the focus a bit more on the novelty of the uncertainty estimation than on the reconstruction.*

**Response:** We thank the referee for the positive evaluation.

Our revisions try to clarify the structure of the manuscript, put the emphasis on the uncertainty, and clarify the Figures.

However, we indeed think that time-series plots are in the most cases the most appropriate visualisation.

**Comments:**

**Introduction**

*- Would be worth mentioning the just published global reconstruction by Neukom et al. 2019, which includes an analog approach, too.*

**Response:** Of course. Until the publication of Neukom et al. (2019) we were not aware of their work. In view of the comments of referee 3 we will discuss their approach.

*- I would find a list of the content helpful at the end of the introduction, saying that three approaches are tested: 1. best analog only, . . .*

**Response:** We add a short paragraph outlining the manuscript.

*- Page 2, line 20: "guestimate" is colloquial language*

**Response:** We regard it an appropriate term, but slightly change the sentence.

**Methods**

*- The entire structure of the study and the used error estimation should be made clearer. Can you add a schematic diagram?*

**Response:** We try to clarify the structure of the methods section and to be more explicit about the error estimation. However, we do not think a schematic diagram is necessary at this point.

*- Explain clearly how you come to your three reconstruction experiments, into which the results are separated. I assume the number 39 for the minimal number of analogs in 2003 (page 5, line 20) is the reason for having 39 in section 3.2 but that is not clear to the reader.*

**Response:** We clarify this part.

*- Page 4, line 23: "under certain assumption" Which assumptions? Please write more precise.*

**Response:** This refers to the assumptions mentioned in the next paragraphs. We clarify the sentence.

*- Page 4, last two paragraphs: I would rather put the equations more prominent in separate lines and not in the middle of the sentence because understanding the error estimation is crucial for this study.*

**Response:** We follow this recommendation.

*- Page 5, line 6: modified*

**Response:** We thank the referee for spotting this.

*- Page 5, line 15: "dates" You have not mentioned yet that you reconstruct JJA averages at annual resolution*

**Response:** We clarify this now at this location and in section 2.1.1.

*- Page 5, line 15ff.: How do you choose the noise SD levels such as 2.57? And why if you write in line 22 that only the 1 SD criterion gives a reasonable number of analogs?*

**Response:** We clarify this now. We choose 2.57SD as it gives a reasonable minimum number for a set of good analogues. We choose 1SD as it gives a reasonable maximum number of analogues for a fixed SD level reconstruction.

**Proxies**

*- Is there a reason to use the gridded CRU data for proxy correlations here and the BEST data later in the paper?*

**Response:** We use the regionally representative series from BEST and we use these for periods before widespread instrumental data is available. We use the CRU data as correlation target as it is commonly used.

*- You could explain that the correlation of the excluded location in Slovakia are low because trees are limited to temperatures in another season. Otherwise it seems strange, why they appear in the PAGES data base. However, I am not sure why the Albania chronology with weakly significant negative correlation appears in the PAGES collection. Maybe, it has been removed in the more strictly screened version 2 of the data base? Having this negative correlation in mind, I do not understand why it is used for comparison/verification later in the paper? I would not expect a good match/positive correlation in the analog reconstruction.*

**Response:** Already Luterbacher et al. (2016) removed these two series from their reconstruction effort. We may remove the relevant panels in our revised version but did not yet decide on this.

We cannot recall why the EuroMed 2k network included both chronologies in their initial reconstruction approach. The original publication for the Albanian record (Seim et al., 2012, https://doi.org/10.3354/cr01076) identifies a significant negative relation to temperature with, however, only small correlation coefficients. We might presume that initially EuroMed 2k considered this to be enough for this data sparse region.

**Model simulations**

*- Page 7, line 5: Please explain again briefly why the "similar internal variability" of the simulation is important instead of referring to the previous section*

**Response:** We clarify this now.

**Results**

*- Generally, try to shorten the results section and have a clear and consistent structure for the three experiments. I would put more focus on the uncertainty results than the reconstruction itself.*

**Response:** We will try to be more concise in our writing while preserving the relevance of the manuscript. We will try to shift the focus to the uncertainty in our revisions.

*- Make clearer, how the three experiments compare and later in the discussion what we can learn from this.*

**Response:** Our revisions try to be clearer about the differences between the three setups.

*- Page 9, line 2: why is the plot relative to Euro-2k and not relative to instrumental data?*

**Response:** The idea was to compare the reconstruction and its uncertainty against a previous reconstruction based on the same data. If we put everything relative to a 20th century data, the potentially different trends over the 20th century will make such a comparison harder. We consider changing the visualisation and aim to clarify this in our revisions.

*- Fig. 3: It is not surprising that the analogs fit better, where you have spatial proxy clusters than isolated locations. I have not seen this discussed in the paper.*

**Response:** We are not sure about the point of the reviewer. We are going to add more discussion on the different data availability. However, considering Figure 3, it is not necessarily the case that the analogues fit better in, e.g., the Alps or Scandinavia.

*- Page 11, line 26: It is good that the analog reconstruction generally agrees with previous statistical reconstructions but they are not a reference and it is unclear which ones are closer to reality. Rather just see if they are in your uncertainty range.*

**Response:** Our revisions aim to be clearer about the evaluation of our reconstruction against the data. However, we would argue in this case that the convergence of both approaches is an important aspect. Indeed, such convergence is, in our view, one strength of the recent work by the PAGES 2k Consortium (2019) and Neukom et al. (2019).

*- Page 14, line 30ff: Have you considered weighting the analogs with respect to their distance?*

**Response:** We considered weighting the analogs. Indeed, weighting may provide us with a clear posterior distribution. However, weighting the analogues by their distance,

in our understanding, to some extent would counter our approach of using analogues that relate to a certain uncertainty level of the proxies.

*- Page 15, line 5: "visually there is good agreement" Not clear what you are talking about, other series besides Tatra and Albania?*

**Response:** We clarify in our revisions: Visually there is good agreement between the proxies included in our analogue search and our reconstructed local series. Usually, the range of reconstructed values is relatively narrow for these proxies.

*- Page 15, line 29: Add reference to figure*

**Response:** We modify and clarify this: Thus, Figure 5 shows that the included proxies anchor the reconstruction to a very narrow range of variability if we choose a fixed number of analogues.

*- Page 15, line 35: How can you have a stronger 20th century warming trend in the reconstruction than in observations and at the same time have trouble to find analogs for exceptionally warm years such as 2003?*

**Response:** We refer to the warming trend from the early 19th century onwards and thereby the mean warming over time for this period. The lack of analogues is due to the exceptional warm years in the early 21st century. We do not find analogues for these, the specific interrelation among the proxy records, and within a narrow one standard deviation uncertainty range.

*- Page 16, line 9: If you look at the temperature evolution after individual eruptions, this is not a superposed epoch analysis.*

**Response:** Thank you for highlighting our lack of clarity. First, we indeed did a valid Superposed Epoch Analysis but considered also individual evolutions. Second, as we only mention the individual evolutions here, we skip the reference to the superposed epoch analysis.

*- Page 19, line 14 and Fig. 8: Why do you show a mean and not the median in this case? The mean should be influenced by the number of averaged analogs and the numbers are highly different in this case.*

**Response:** The referee is correct. We redid the analysis with the median. We are yet undecided what to show in the revised manuscript because visual differences are negligible and differences in results are small. We will discuss our decision.

*- Page 19, line 28ff: Why is the comparison with instrumental data just done for the 1 SD reconstruction?*

**Response:** We considered the fixed number 1SD reconstruction as essential part of the work and therefore did it only for this. We consider including equivalent Figures for the other two approaches in an appendix.

**Concluding remarks**

*- Please avoid 1-sentence paragraphs*

**Response:** We will do so.

**Figures**

*- Generally, please think of a way to reduce the number of figures with time series. Both, the number of really necessary panels in each figure and figures in total. E.g. it is probably not required to see the annual resolution reconstruction for the full period for all three experiments or do multiple smoothing have to be presented?*

**Response:** We will reconsider all Figures.

*- I find the uncertainty ranges often impossible to see (e.g. Fig. 2a). I cannot recognize the "envelope", you are talking of. As this a main focus of the paper, please try to find a way to plot uncertainty better visible, e.g. just a smoothed version for the entire period and a subperiod at annual resolution.*

**Response:** We will reconsider all visualisations and try to put maximum emphasis on the uncertainty ranges.

**Referee 3**

*This study proposes a new climate reconstructions for Europe for nearly the full last millennium. The approach is based on the Analog Method, also known in the literature as Proxy Surrogate Reconstruction. One of the main novelties of this manuscript is how the authors extend the methodology to explicitly account for uncertainties. The authors present several reconstructions and compare them to the Euro 2k reconstruction, as well as independent data from the BEST project. Similarities, differences, advantages and caveats are discussed through the manuscript.*

**General comment**

*Most classical reconstruction methods produce a single reconstruction which does not explicitly account for uncertainty, although it is acknowledged that it populates this type of data-sets. This is problematic because uncertainty is not only ubiquitous, but it is heterogeneous both in time and space. This is an important limitation that precludes the proper assessment of the limitations of the knowledge we can gather from climate reconstructions. In this sense, I think this study is important and necessary to improve one prominent tool to produce such reconstructions, the Analog Method.*

**Response:** We thank the referee for their evaluation and rating of our manuscript.

*The design of the study is sensible, and I have mostly minor comments regarding details I could not fully understand and therefore might deserve clarification. Should not be for the issue I discuss below, I would recommend publication after minor revision.*

**Response:** We thank the referee.

*There is however and important aspect that has to be improved in the manuscript under the light of very recent bibliography published even after this discussion was started.*

*There exists a published extension to the Analog Method that allows to estimate uncertainties. This is part of a recent publication with a more general aim (Neukom et al., 2019). There, authors briefly introduce and apply a methodology which largely differs from the one presented here, but that aims at the same purpose: explicitly assess uncertainties in climate field reconstructions with the Analog Method. I think this work should somehow account for the existence of this already published method. The level of modification applied to the manuscript depends on the authors. At the minimum, the differences between approaches should be discussed (for example, the approach opted by Neukom et al. (2019) does not produce missing values, being in principle an important advantage). At best, the approach adopted by Neukom et al. (2019) could be implemented here as well, and a comparison could be done between both methods. In my opinion, the latter would greatly improve the interest of this manuscript, but it is perhaps a major modification of the work that falls beyond its original scope. I leave it up to the authors and I would not be disappointed if they decide not to tackle this task.*

**Response:** As the referee notes, we became aware of Neukom et al. (2019) after the discussion phase started. In view of their publication, we have to modify various parts of the manuscript. We will thoroughly discuss the differences between our approach and their approach.

At this point it is unlikely that we produce a bootstrap uncertainty estimate following Neukom et al.

The approach of Neukom et al. combines two sources of uncertainty. These are differing pools of candidate fields and differing proxy coverage. The former is to some extent included in our consideration of a set of fields, which agree with the initial uncertainty. The latter is not included in our approaches. Neukom et al. describe the uncertainty if we have less information available than we have. We describe the uncertainty due to the uncertainty in the proxy anchors.

**Minor comments**

*1. Page 2, Line 20: I think the correct citation is Gómez-Navarro et al. (2014)*

**Response:** Gómez-Navarro et al. (2017) discuss the main differences between the analogue search and offline data assimilation approaches.

*2. Fig 1: Maybe excluded locations could be shown with grey symbols, as well as the are representative for Central Europe. The location of these proxies is relevant for example to understand Figure 5.*

**Response:** We make these modifications.

*3. Page 4, Lines 28-29. I think it is more correct to say that, only when Varres and Varsig are uncorrelated, the total variance is the sum of both (because in that case the covariance term vanishes).*

**Response:** This is what we intended to express. We will clarify our writing.

*4. Page 5, Line 6: typo (modfied)*

**Response:** We thank the referee for spotting this.

*5. Page 5, Lines 17–21: I do not understand where the 2.57 comes from. How it is related to the minimum number of 39 proxies? Please clarify.*

**Response:** We clarify this in the revised manuscript. 2.57SD is equivalent to a 99% interval. 39 is the smallest number of analogues found at any date. We therefore later choose this as the size of our fixed number reconstruction ensemble.

*6. Page 5, Lines 22-23: why is it the only one? why 2105 is special? why not 1.5 $SD_{noi}$?*

**Response:** Considering fixed $SD_{noi}$ intervals, the number of valid analogues increases. It may become soon unfeasibly large. We think that the 2105 analogues

for a $1SD_{noi}$ interval are still reasonable. Therefore, we only consider a $1SD_{noi}$ interval for the fixed SD reconstruction.

*7. Overall, in the two paragraphs aforementioned, it lies the core of the two reconstructions carried out. I think this is important, and it should be made more explicit that the two approaches represent different method used for real below. Perhaps this can be made more explicit with some structural element, such as an un-ordered list or similar.*

**Response:** We try to clarify the methods section in our revised manuscript.

*8. Page 6, Table 1: I assume this is exactly the correlation used to define the $SD_{noi}$ in each proxy location, right? If so, this could be clarified in the main text (especially in section 2.1.2).*

**Response:** Yes it is. We will clarify this in the main text.

*9. Page 6, lines 6–8: The criterion to exclude two proxies is not very clear. What is meant by "relevant portion of variance"? In Fig. 5 we learn that the reconstruction in these sites is poor. Would it be better if these sites were part of the network. Surely the answer is yes. I understand that the amount of climate information we get is poorer than in the other locations, but still we could benefit for having some information. At worst, if the proxies were pure noise, it would not be necessarily worse than not having information at all. In other words, I think having poor information is better than having none, and it's not fully obvious to me why proxies should be excluded from the analysis based on relatively low correlation alone.*

**Response:** We clarify this in the revised manuscript.

Already Luterbacher et al. (2016) excluded these proxies because they lack a clear temperature signal.

The referee is correct that, generally, a pure noise record should not be worse than having no information at all. However, this is not the worst case. The worst case would be a record which biases the distance measure in our analogue search towards

a different state.

That is, we follow the common approach to only include proxies with a signal beyond a certain level. We are aware that this has been a controversial decision in the past but we regard it valid in view of past practices.

*10. Page 6 (but relevant for the whole study): why do you restrict the reconstruction to the period 1260 to 2003? The reconstruction could have been applied further back in time. The number of proxies varies in time, but this could be even beneficial for this study, focused on the validation of new methodologies. It would show how the estimates of the uncertainty presented here are sensible to a varying number of proxies. I feel that this choice has unnecessarily limited the scope of the manuscript.*

**Response:** We thank the referee for their confidence in our approach. The referee is correct that in principle there are no reasons to stop in 1260. However, stopping there, in our opinion, eases the interpretation of results since thereby only an equal number of proxies enters the uncertainty estimation.

We will clarify this in the revised manuscript.

*11. Page 7, Table 2: it could be interesting to write the total number of analogues, i.e. the pool size. It would make more meaningful the number of proxies used to produce ensembles. For example, having 817 analogues (as in Fig. 8) has a clearer meaning when you add that they are 817 out of, let's say, 25000. It shows that you are still selecting a relatively minor number of relatively good analogues.*

**Response:** We clarify this in the revised manuscript.

*12. Page 7, Lines 6–7: I think having a consistent bias through the pool is not necessarily good, as it seems to be implied by the wording. It ensures that the bias are translated into the reconstruction. This is partly avoided using structurally different models to build the pool. I do not mean that the authors should necessarily rebuild the reconstruction with a larger set of models, but I think that at least they should not imply*

*that using a single model is somehow beneficial.*

**Response:** We clarify this in the revised manuscript.

*13. Page 8, Fig. 2: I think a line marking the 0 K anomalies would help to read the series. This pertains mostly panels b and c, where the sign of the anomaly is important, but difficult to appreciate without such a line. This argument applies to Figs 6 and 7 as well.*

**Response:** Our revisions generally try to make our Figures clearer. We do add a zero line to panel c) but are yet undecided about other panels, because it even more increases the number of elements per panel.

*14. Page 8, Fig. 2: It's not fully clear to me what this figure (as well as Figs 6 and 7) show. Does "summary" mean spatial average?*

**Response:** "Summary" means summary of the main results of the reconstruction. We modify the captions.

*15. Page 9, Line 10: please change "degree Kelvin" to "Kelvin". Please review it, as there are other locations where I saw this in the manuscript.*

**Response:** We do so.

*16. Page 9, Lines 9–16. The order of these two paragraphs can be exchanged. It's a bit unusual and therefore confusing to discuss Fig. 2c before Fig. 2b.*

**Response:** We restructure the section in our revisions.

*17. Page 10, Lines 10–15: I think the fact that the reconstruction underestimate the intra-location variability is a problem of the pool, not the Analog Method itself. Do the authors think that this could be improved if higher resolution models were used to build the pool?*

**Response:** We did not investigate this feature. We only state it without attributing it to

the method. There are a number of explanations on which we only very shortly touch here. First, the noisy proxy series may overestimate the true intra-location variability. Second, the simulations may be too smooth in space. This, thirdly, might be due to the low resolution and simulations with higher resolutions might help then. Fourth, the chosen distance measure may result in such a feature dependent on the characteristics of the simulation pool, which however should usually not be the case.

We clarify our statement in the revised version.

*18. Page 11, Fig. 3: The list of locations in the caption is misleading (the name and the ID are written all together). It seems a detail, but it puzzled me for a while until I realised that Tor92 and Torneträsk are not two proxies, but the ID and the name of the same one. You could easily remove this by using for instance parenthesis to separate name from ID or vice versa.*

**Response:** We clarify the caption.

*19. Page 11, Line 26: "The general agreement between the Euro 2k and the analogue..." this reads odd at this point, as the reader does not know where to find the information the authors are referring to. It turns out that this comparison is introduced later, in Figure 6 in Page 14.*

**Response:** We try to clarify in our revisions what is meant at this point.

*20. Page 15: Lines 17–23: The reduced variance could be quantified (how much is notably smaller variance in Line 19?). Further, the lost of variance when more analogues are considered is common in this approach, and generally in any statistical approach, i.e. there is a bias-variance trade off. It could be noted here that this has been comprehensively discussed in the bibliography of the Analog Method.*

**Response:** We will make these discussions clearer in the revised version.

*21. Page 15, Line 32: do the authors have a theory on what could be the reason for such systematic differences? Are they meaningful, can they be used to discuss*

*merits or problems in the reconstructions? Or are they rather low-frequency random fluctuations highly sensitive to method parameters?*

**Response:** In the case of the mid 16th century deviation there are indications that the Euro 2k more validly captures the extremes in this period (Wetter and Pfister, 2011, https://doi.org/10.5194/cp-7-1307-2011, 2013, https://doi.org/10.5194/cp-9-41-2013), which may again indicate a period where the simulation pool is insufficient. Generally though, we would assume that it is mainly random due to the different sensitivities of the methods.

*22. Page 16: Lines 25–26: The presence of missing values in years with volcanic eruptions is a major caveat of the method, as those are typically the years most interesting in climate studies. Here it would be specially relevant my comment about a comparison with the method presented by Neukom et al. (2019).*

**Response:** We will discuss this.

*23. Page 16, Lines 27–31: I do not see why it is "unsurprising" this lack of analogues for the recent period. The pool contains this warming as well, so the search should not present more problems for this period than in any other.*

**Response:** We will clarify this.

Indeed, the pool includes this period but we do not only require a similar mean state but also a similar interrelation between locations, which makes it more likely that the limited size of the pool does not include such a case.

*24. Page 19, Lines 18–20: Maybe I'm miss-evaluating this, but I think that anchoring the reconstruction within a range of 8 K is a poor result. It shows that the 800 analogues are indeed poorly constrained in this region, so we have little idea of how the actual climate was in that period and region. More generally, I have the concern that the spread shown for example in Fig. 7 might provide an optimistic measure of the actual uncertainty. Fig 7e for instance shows the range in the spatial average, which is about 2*

*K. But this is after spatial average, where regional differences can cancel out! I wonder how large is the range in each location. This might perhaps be illustrated with a map of (temporally averaged) ranges? Eventually, my guess is that using as many as 800 analogues or more, really far away from "the best" is, as outlined by the authors, too much.*

**Response:** We consider visualising the mean and the range of the temporal temperature ranges.

---

## Author Response (AR1)

Dear referees, dear editor,

We want to thank the referees and the editor for evaluating our manuscript and providing such encouraging comments.

Below we respond to the reviewer comments and list our main changes.

On behalf of the authors

Sincerely yours,

Oliver Bothe

**List of main changes:**

- \* Abstract
  - Clarified writing
- \* Introduction
  - Clarified writing
  - Added discussion of Neukom et al., 2019
- \* Methods & Data
  - Clarified writing
  - Clarified structure
  - Added table
  - Modified Figure 1
  - Added information on Neukom et al., 2019
- \* Results
  - Clarified writing
  - Clarified structure
  - · Changed visualization of the results by reducing the number of time-series plots and adding other Figures
  - Added short description of results from a subsampling approach following Neukom et al., 2019
  - Added comparison of different uncertainty estimates
- \* Summary & Discussions
  - Clarified writing
  - Clarified structure
- \* Appendix
  - Added appendix
  - Added supplementary figures to appendix

**Editor**

Please for the revisions, you might add in the paper that, compared to PAGES2k, Luterbacher et al. 2016 excluded the Tatra and Albania proxies from their analysis as they lack significant correlations with European summer temperature variability.

**Response** We make this change and add: Already Luterbacher et al. (2016) noted this and, therefore, did not consider these two proxies in their reconstruction effort. That is, we, as Luterbacher et al. (2016), exclude these proxies because there is not a clear relation to temperature.

**Referee 1**

Bothe and Zorita present a study where they investigate different ways of obtaining an uncertainty estimate for climate reconstructions using the analogue method, also known as the proxy surrogate reconstruction method. They authors describe the downside of single member reconstructions, and produce both single member and multi ensemble member reconstructions, which are compared to other reconstructions and observations. Then they go on to describe how an uncertainty can be assigned based on i) the fit of the analogue ii) assumptions on the standard deviation of the noise or iii) the ensemble spread when using a fixed number of ensemble members. Finally, the authors conclude on the pros and cons of the different approaches.

**General comments.**

This study is overall well executed, thorough and timely. However, the writing is somewhat uneven, especially in the introduction, which I have commented on in detail below, but please go through the entire manuscript as I might have run out of steam. I have few major comments about the methodology itself, but I am wondering if the method is overfitting the model data to the proxies (see specific comments below). If the writing is brushed up as well as taking my other comments into account, I think this work could be suitable for publication.

**Response:** We thank the reviewer for their positive evaluation.

We hope that our revisions do improve the writing.

Regarding the overfitting: See our response below.

**Specific comments.**

P1, L1: Please rewrite this sentence. It is the combination itself that reconciles the two sources, so if this is possible "allows" is redundant. Also, if one is reconciled with the other then they are both reconciled, making "both" redundant.

**Response:** We change this to: Combining proxy information and climate model simulations reconciles these sources of information about past climates.

*P1, L3: ". . . to benefit from the advantages of both data sources" this is in a way a repetition from previous sentence. Why not say something about the technique? E.g. "The analogue or proxy surrogate reconstruction method is a computationally cheap data assimilation approach which samples a model ensemble based on the best match to proxy data".*

**Response:** We are not convinced that this simply repeats the content of the previous sentence but follow the suggestion of the referee: The analogue or proxy surrogate reconstruction method is a computationally cheap data assimilation approach, which searches in a pool of simulated climate states the best fit to proxy data.

P1, L9: Replace "had been" with "was"?

**Response:** We change this accordingly.

P1, L10: Remove "using"?

**Response:** We change this accordingly.

P1, L12-L14: "The approaches do not agree. . . " this sentence is not easy to read. Perhaps rewrite "However, the two approaches do not agree on the warmest preindustrial decades, which for the Euro 2k reconstruction is during the early 15th century, and for the analogue approach is during the early 18th century".

Response: We change and shorten this: However, the approaches disagree on the warmest preindustrial periods.

P1, L15: "The surrogate reconstructions..." I suggest that you early in the manuscript choose to call the reconstructions either surrogate or analogue, even if you have said it means the same - just to make it easier to read.

**Response:** We try to be consistent in our writing. Therefore, we here change the sentence to: The reconstructions from the analogue method ...{}

P1, L15: Insert comma before "but". Please use more commas to help the reader.

**Response:** Our revisions try to use more commas to ease reading the manuscript.

P1, L15-L16: Actually, I don't understand the sentence. You lose me around "even under uncertainty". Please rewrite.

**Response:** We change this to read: The reconstructions from the analogue method also represent the local variations of the observed proxies.

P1, L20: Is "paleo-observations" the right word? Why not simply "proxy data"?

Response: We regard it to be the right expression but change the occurrences according to the referee's suggestion.

P2, L1: Replace "the search for" with "finding"?

**Response:** We replace this with "searching" and slightly modify the sentence.

P2, L7: Why "not only", maybe cut this?

**Response:** The part now reads: The analogue method found subsequent applications in downscaling and upscaling of climate information ...

P2, L14-L15: "The analogue method. . . " This sentence is hard to follow. Please rewrite.

**Response:** The full paragraph is rewritten.

P3, L1: Either write "Here we propose ..." or "In this study we provide...".

Response: We rewrite: "Here, we propose"

P3, L?? (something strange happens with the line numbers): "Here, we obtain..." (skip comma after "Here"). Please be clear on what is model and what is proxy. I suppose "pool of relevant candidate fields" is model out put and "local indices" is proxy data?

**Response:** We are sorry for the random line numbers. We use the RMarkdown template and under certain unclear conditions this happens, but can easily be repaired, which we unfortunately did not do.

We rewrite: Here, we obtain annually resolved large-scale fields of seasonal mean summer (June, July, August, JJA) temperature based on a pool of relevant candidate fields and a set of local data indices as predictors for the period 1260 to 2003 of the Common Era (CE).

Figure 1: Please add units to axes.

**Response:** We clarify the Figure.

P4, L15-L16: "That is, they use recent observations, which measured archives...". I don't follow this sentence. Please rewrite.

**Response:** We rephrase the full paragraph.

P4, L19: "to more than one environmental condition" do you mean that a given proxy paramenter can depend on more than climate or environmental variable? Please clarify.

**Response:** We rewrite: The most obvious source of uncertainty is that the archives recorded signals from more than one climate or environmental variable (e.g. temperature and precipitation; compare Evans et al., 2013; Tolwinski-Ward et al., 2013; Evans et al., 2014; Tolwinski-Ward et al., 2015).

P4, L21-22: Is "environmental condition" the correct word, or is "climate state" more accurate.

**Response:** We think our choice of words is valid, but we rewrite: Correlations provide a simple measure of the relation between proxy-observations and the climatic environment over a period when reliable (instrumental) observations of the climatic variability exist.

P4, L26: I suggest you make a sheet or table with mathematical abbreviations that you use in the paper.

**Response:** We add another table.

Table 1: So, the correlations are between the tree ring data series and the JJA CRU temperature. Please add these details to the caption. How do you deal with seasonality of the proxy data? In Wilson et al. (2016) each proxy site is listed with different seasonal sensitivity to temperature (Table 1) and I believe you are using some of the same data.

**Response:** 1. We add the details to the caption. 2. As we compare our reconstruction to the Euro 2k reconstruction, we consider the seasonal attribution as used by the publications for this reconstruction. That is, we do not test whether the relation of the proxies is strongest for summer. For the attribution to summer compare Luterbacher et al. (2016) and PAGES 2k Consortium (2013).

**P6, L13: "The last of the remaining eight proxy indices starts in 1260" meaning that all remaining records cover 1260 to 2003 CE?**

**Response:** Yes. We clarify this: We describe results for the period 1260 to 2003, although two of the Euro 2k proxy series extend back to the year 138 BC, and the analogue approach is suited to use variable numbers of proxies. The latest start date of any of the used eight proxy indices is the year 1260 CE, and, thus, all eight records cover the period 1260 to 2003 CE. We decide against using uneven numbers of proxies and against extending the reconstruction further back to ease the comparison of the results and our different uncertainty estimates.

P7, L16: "strong ensemble" or "8 member ensemble"?

**Response:** We clarify: there exists a multi-model ensemble of climate simulations for the last approximately 1100 years. A number of additional simulations comply with the PMIP3 protocol but are not included in the effort

P7, 2nd paragraph, L8: "Since the current manuscript is not least a proof of concept..." this formulation sounds off.

**Response: We remove the sentence.**

Figure2: (a) rescaled temperature? Please specify which scaling is used in the caption, so you don't have to look for it. Euro 2k is an area mean? It's hard to see that the difference to the CRU temp. Can you show this in (c)? Luterbacher et al. (2016) is discussed a lot in relation to Figure 2, it would be helpful if you show this data as well.

**Response:** 1. We clarify the rescaling in the caption. 2. Euro 2k is an area mean. 3. We are unclear what the referee is referring to with respect to the CRU data. 4. We add an Appendix to show additional Figures and do a comparison of differences there. 5. The Appendix also includes a Figure showing the data by Luterbacher et al. (2016).

P9, L17: "calculated as the square root ..." why not write out the equation?

**Response:** We thought it was clear enough, but now show the equation.

**P9, c. L25: So which uncertainty is realistic? And why?**

**Response:** We shortly describe the characteristics of both uncertainty estimates. Both describe realistically part of the uncertainty: Both uncertainty measures for the analogue reconstruction describe different but not mutually exclusive parts of the uncertainty of the reconstruction. The variance based envelope estimates the reconstruction uncertainty based on the local agreement between proxies and observations over the period when instrumental data is available. Thus, it is unlikely that the uncertainty of the reconstruction at any time is smaller than this estimate because we can assume that the quality of the proxies is best in the recent period. The proxy based noise uncertainty estimate includes local information but extrapolates these over the period without instrumental data. On the other hand, the mean square error captures the misfit between the uncertain proxies and the final reconstruction product. Where it is smaller than the variance based estimate, we would call it unrealistic. When it exceeds this estimate, it is preferable.

*P9, L29-L30: "The coldest century was until 1648 CE in the best-analogue reconstruction but until 1678 CE in the Euro 2k record" please write the interval of the coldest century. This formulation is unclear.*

**Response:** We clarify the description of these types of results.

P10, L1-L? (again random line numbers): When discuss interval please write them out instead of just giving the end year. It's much easier to read. Just write "the warmest century was 1353-1452 CE".

**Response:** We change and shorten the description of these types of results.

P10: About the volcanic analysis. Did you look at high latitude eruptions, e.g. Laki? How did you do the super imposed epoch analysis? Maps of field anomalies, or time series? How did you define the reference period before the eruptions?

**Response:** We only considered tropical eruptions. The paragraph now reads: We now consider the response to volcanic forcing, as volcanoes are considered to be the most important external forcing over the pre-industrial period. They are also the best constrained past climate forcing for the last 500 to 2000 years (e.g., Sigl et al., 2015; Wilson et al., 2016). The period of our reconstructions includes only a few of the large tropical eruptions of the last millennium. We consider a subselection of tropical eruption events in 1286, 1345, 1458, 1601, 1641, 1695, 1809, and 1815. We performed a superposed epoch analysis but we do not graphically show the results. We considered fields and area averages. We chose the five calendar years before an eruption year as reference period, which is a common approach (compare, e.g. Sigl et al., 2015).

P10, 2nd paragraph, L7-L8: "Interestingly, the analogues even appear to occasionally capture the relation between the proxies included and those excluded" couldn't this be completely random? Then it's not very interesting.

**Response:** We modify this: The analogues even appear to occasionally capture the relation between the proxies included and those excluded, which obviously might be by chance.

P10, 2nd paragraph, L10: Replace "1947. Then" with "1947, where".

Response: We do so.

Figure 4: What are the numbers next to the site name e.g. "(a) Tor92 0.91"? Is 0.91 the correlation?

**Response:** Yes. We clarify the caption.

Figure 5: Again, what are the numbers next to the site names?

**Response:** We again clarify the caption by mentioning that these are the correlations.

P15, L4: Correlations "between 0.84 and 0.98" for proxies and and reconstructed temperature. These correlations are a good bit higher on average than the data in Table 1. Are you overfitting the data, or how can you explain this? Wouldn't you need forward proxy modeling of tree growth to give a more realistic link between model and proxy data (e.g. Tardif et al. 2019)?

**Response:** The correlations are between the proxy locations and the medians of 39 analogues. Indeed they are high, but we would hope that the proxies included in our search constrain the search effectively and give good reconstruction results. This holds especially for the median, which is a filter for the data of the reconstruction ensemble members. The aim of data assimilation is to match the observations, i.e.~the proxies, closely with the simulated data. Nevertheless, we are unable to exclude the possibility that our data constrains the pool too much and therefore may fail in a prediction excercise.

These correlations indicate only agreement with the proxy records not necessarily with the true temperature. Anyway, locally high correlations do not indicate high skill elsewhere. Indeed, correlations with the observational CRU data are in line with the correlations between the proxies and the CRU data as one may expect from these high correlation coefficients. The comparison to the BEST-data shows, this does not necessarily reflect on how well the reconstruction captures the observed temperature elsewhere.

Regarding the use of proxy system models: An optimal approach would incorporate a calibrated proxy system model to preprocess the simulation data. Indeed, any reconstruction approach can benefit from pre-processing data with calibrated proxy forward models.

Regarding overfitting, there are no parameters in the analog-setting that are calibrated for a better fit to the predictand. The number of analogs chosen or the distance metric chosen are not optimized for a better fit.

P21, L8: Is it really "strange variability" since the reconstruction is unconstrained in Greenland?

**Response:** We clarify this: The top-left panel for Nuuk highlights that the lack of constraints on the reconstruction can result in potentially artificial spikes in the time-series.

**Referee 2**

This study is an interesting contribution to the field of climate reconstruction because it adds and compares multiple way of uncertainty estimation to the widely and successfully used analog reconstruction methodology. It fits very well to the scope of Climate of the Past. Hence, I suggest publication after revisions that should make the structure clearer, condense the results including figures with time series and after putting the focus a bit more on the novelty of the uncertainty estimation than on the reconstruction.

**Response:** We thank the referee for the positive evaluation.

Our revisions try to clarify the structure of the manuscript, put the emphasis on the uncertainty, and clarify the Figures.

We try to reduce the number of time-series plots, but we nevertheless feel that they are the most appropriate visualisation in many cases.

**Comments:**

**Introduction**

- Would be worth mentioning the just published global reconstruction by Neukom et al. 2019, which includes an analog approach, too.

**Response:** Of course. Until the publication of Neukom et al. (2019) we were not aware of their work. In view of the comments of referee 3 we will discuss their approach.

- I would find a list of the content helpful at the end of the introduction, saying that three approaches are tested: 1. best analog only, . . .

**Response:** We add a short paragraph outlining the manuscript.

- Page 2, line 20: "guestimate" is colloquial language

**Response:** We regard it an appropriate term, but nevertheless remove the phrase.

**Methods**

- The entire structure of the study and the used error estimation should be made clearer. Can you add a schematic diagram?

**Response:** We try to clarify the structure of the manuscript and in particular of the methods section. Our revisions are more explicit about the error estimation. However, we do not think a schematic diagram is necessary at this point.

- Explain clearly how you come to your three reconstruction experiments, into which the results are separated. I assume the number 39 for the minimal number of analogs in 2003 (page 5, line 20) is the reason for having 39 in section 3.2 but that is not clear to the reader.

Response: The revisions clarify this part.

- Page 4, line 23: "under certain assumption" Which assumptions? Please write more precise.

**Response:** This refers to the assumptions mentioned in the next paragraphs. We clarify the sentence and restructure the section.

- Page 4, last two paragraphs: I would rather put the equations more prominent in separate lines and not in the middle of the sentence because understanding the error estimation is crucial for this study.

Response: We follow this recommendation.

- Page 5, line 6: modified

**Response:** We thank the referee for spotting this.

- Page 5, line 15: "dates" You have not mentioned yet that you reconstruct JJA averages at annual resolution

Response: We clarify this now at this location and in section 2.1.1.

- Page 5, line 15ff.: How do you choose the noise SD levels such as 2.57? And why if you write in line 22 that only the 1 SD criterion gives a reasonable number of analogs?

**Response:** Our revisisions try to clarify our reasoning about our different approaches. We choose 2.57SD as it gives a reasonable minimum number for a set of good analogues. We choose 1SD as it gives a reasonable maximum number of analogues for a fixed SD level reconstruction.

**Proxies**

- Is there a reason to use the gridded CRU data for proxy correlations here and the BEST data later in the paper?

**Response:** We use the regionally representative series from BEST and we use these for periods before widespread instrumental data is available. We use the CRU data as correlation target as it is commonly used.

- You could explain that the correlation of the excluded location in Slovakia are low because trees are limited to temperatures in another season. Otherwise it seems strange, why they appear in the PAGES data base. However, I am not sure why the Albania chronology with weakly significant negative correlation appears in the PAGES collection. Maybe, it has been removed in the more strictly screened version 2 of the data base? Having this negative correlation in mind, I do not understand why it is used for comparison/verification later in the paper? I would not expect a good match/positive correlation in the analog reconstruction.

**Response:** Already Luterbacher et al. (2016) removed these two series from their reconstruction effort. We do not remove the relevant panels this round of revisions as we think they still provide information.

We cannot recall why the EuroMed 2k network included both chronologies in their initial reconstruction approach. The original publication for the Albanian record (Seim et al., 2012, https://doi.org/10.3354/cr01076) identifies a significant negative relation to temperature with, however, only small correlation coefficients. We might presume that initially EuroMed 2k considered this to be enough for this data sparse region.

**Model simulations**

- Page 7, line 5: Please explain again briefly why the "similar internal variability" of the simulation is important instead of referring to the previous section

Response: We add an explanation.

**Results**

- Generally, try to shorten the results section and have a clear and consistent structure for the three experiments. I would put more focus on the uncertainty results than the reconstruction itself.

**Response:** Our revisions try to improve the structure and to be more concise in the description of the results while at the same time preserving the relevance of the manuscript and incorporating all referees' comments.

The revisions put slightly more focus on the uncertainty.

- Make clearer, how the three experiments compare and later in the discussion what we can learn from this.

**Response:** Our revisions try to be clearer about the differences between the three setups.

- Page 9, line 2: why is the plot relative to Euro-2k and not relative to instrumental data?

**Response:** The idea was to compare the reconstruction and its uncertainty against a previous reconstruction based on the same data. We add comparisons to the observational data in the appendix.

- Fig. 3: It is not surprising that the analogs fit better, where you have spatial proxy clusters than isolated locations. I have not seen this discussed in the paper.

**Response:** We are not sure about the point of the reviewer. We are going to add more discussion on the different data availability. However, considering Figure 3, it is not necessarily the case that the analogues fit better in, e.g., the Alps or Scandinavia. We extend on this slightly in discussing the current Figure 4 and the new Figure 10.

- Page 11, line 26: It is good that the analog reconstruction generally agrees with previous statistical reconstructions but they are not a reference and it is unclear which ones are closer to reality. Rather just see if they are in your uncertainty range.

**Response:** Our revisions aim to be clearer about the evaluation of our reconstruction against the data. However, we would argue in this case that the convergence of both approaches is an important aspect. Indeed, such convergence is, in our view, one strength of the recent work by the PAGES 2k Consortium (2019) and Neukom et al. (2019). We compare more to the observational data.

- Page 14, line 30ff: Have you considered weighting the analogs with respect to their distance?

**Response:** We considered weighting the analogs. Indeed, weighting may provide us with a clear posterior distribution. However, weighting the analogues by their distance, in our understanding, to some extent would counter our approach of using analogues that relate to a certain uncertainty level of the proxies.

- Page 15, line 5: "visually there is good agreement" Not clear what you are talking about, other series besides Tatra and Albania?

**Response:** We rewrite the description of these results.

- Page 15, line 29: Add reference to figure

**Response:** We rewrite the description of these results.

- Page 15, line 35: How can you have a stronger 20th century warming trend in the reconstruction than in observations and at the same time have trouble to find analogs for exceptionally warm years such as 2003?

**Response:** We refer to the warming trend from the early 19th century onwards and thereby the mean warming over time for this period. The lack of analogues is due to the exceptional warm years in the early 21st century. We do not find analogues for these, the specific interrelation among the proxy records, and within a narrow one standard deviation uncertainty range.

- Page 16, line 9: If you look at the temperature evolution after individual eruptions, this is not a superposed epoch analysis.

**Response:** Thank you for highlighting our lack of clarity. First, we indeed did a valid Superposed Epoch Analysis but considered also individual evolutions. Second, as we only mention the individual evolutions here, we skip the reference to the superposed epoch analysis.

- Page 19, line 14 and Fig. 8: Why do you show a mean and not the median in this case? The mean should be influenced by the number of averaged analogs and the numbers are highly different in this case.

**Response:** The referee is correct. We redid the analysis with the median. We show the median now in the revised manuscript as this is more correct as highlighted by the referee. Visual differences are negligible and differences in results are small.

- Page 19, line 28ff: Why is the comparison with instrumental data just done for the 1 SD reconstruction?

**Response:** We considered the fixed number 1SD reconstruction as essential part of the work and therefore did it only for this. Equivalent Figures for the other two approaches are now in the appendix.

**Concluding remarks**

- Please avoid 1-sentence paragraphs

Response: We will do so.

**Figures**

- Generally, please think of a way to reduce the number of figures with time series. Both, the number of really necessary panels in each figure and figures in total. E.g. it is probably not required to see the annual resolution reconstruction for the full period for all three experiments or do multiple smoothing have to be presented?

**Response:** We reconsidered all Figures. Thereby, we reduced the number of panels showing time-series.

- I find the uncertainty ranges often impossible to see (e.g. Fig. 2a). I cannot recognize the "envelope", you are talking of. As this a main focus of the paper, please try to find a way to plot uncertainty better visible, e.g. just a smoothed version for the entire period and a subperiod at annual resolution.

**Response:** Our revisions reconsider all visualisations and try to put maximum emphasis on the uncertainty ranges. A new subsection shortly compares the uncertainty estimates.

**Referee 3**

This study proposes a new climate reconstructions for Europe for nearly the full last millennium. The approach is based on the Analog Method, also known in the literature as Proxy Surrogate Reconstruction. One of the main novelties of this manuscript is how the authors extend the methodology to explicitly account for uncertainties. The authors present several reconstructions and compare them to the Euro 2k reconstruction, as well as independent data from the BEST project. Similarities, differences, advantages and caveats are discussed through the manuscript.

**General comment**

Most classical reconstruction methods produce a single reconstruction which does not explicitly account for uncertainty, although it is acknowledged that it populates this type of data-sets. This is problematic because uncertainty is not only ubiquitous, but it is heterogeneous both in time and space. This is an important limitation that precludes the proper assessment of the limitations of the knowledge we can gather from climate reconstructions. In this sense, I think this study is important and necessary to improve one prominent tool to produce such reconstructions, the Analog Method.

**Response:** We thank the referee for their evaluation and rating of our manuscript.

The design of the study is sensible, and I have mostly minor comments regarding details I could not fully understand and therefore might deserve clarification. Should not be for the issue I discuss below, I would recommend publication after minor revision.

**Response:** We thank the referee.**

There is however and important aspect that has to be improved in the manuscript under the light of very recent bibliography published even after this discussion was started. There exists a published extension to the Analog Method that allows to estimate uncertainties. This is part of a recent publication with a more general aim (Neukom et al., 2019). There, authors briefly introduce and apply a methodology which largely differs from the one presented here, but that aims at the same purpose: explicitly assess uncertainties in climate field reconstructions with the Analog Method. I think this work should somehow account for the existence of this already published method. The level of modification applied to the manuscript depends on the authors. At the minimum, the differences between approaches should be discussed (for example, the approach opted by Neukom et al. (2019) does not produce missing values, being in principle an important advantage). At best, the approach adopted by Neukom et al. (2019) could be implemented here as well, and a comparison could be done between both methods. In my opinion, the latter would greatly improve the interest of this manuscript, but it is perhaps a major modification of the work that falls beyond its original scope. I leave it up to the authors and I would not be disappointed if they decide not to tackle this task.

**Response:** As the referee notes, we became aware of Neukom et al. (2019) after the discussion phase started. In view of their publication, we have to modify various parts of the manuscript. We will thoroughly discuss the differences between our approach and their approach.

We add a short additional results section for a subsampling approach following Neukom et al.

The approach of Neukom et al.~combines two sources of uncertainty. These are differing pools of candidate fields and differing proxy coverage. The former is to some extent included in our consideration of a set of fields, which agree with the initial uncertainty. The latter is not included in our approaches. Neukom et al.~describe the uncertainty if we have less information available than we have. We describe the uncertainty due to the uncertainty in the proxy anchors.

**Minor comments**

1. Page 2, Line 20: I think the correct citation is Gómez-Navarro et al. (2014)

**Response:** Gómez-Navarro et al. (2017) discuss the main differences between the analogue search and offline data assimilation approaches.

2. Fig 1: Maybe excluded locations could be shown with grey symbols, as well as the are representative for Central Europe. The location of these proxies is relevant for example to understand Figure 5.

Response: We make these modifications.

3. Page 4, Lines 28-29. I think it is more correct to say that, only when  $Var_{res}$  and  $Var_{sig}$  are uncorrelated, the total variance is the sum of both (because in that case the covariance term vanishes).

Response: This is what we intended to express. Our revisions clarify this.

4. Page 5, Line 6: typo (modfied)

**Response:** We thank the referee for spotting this.

5. Page 5, Lines 17–21: I do not understand where the 2.57 comes from. How it is related to the minimum number of 39 proxies? Please clarify.

Response: Our revisions aim to clarify the description of our approach and our reasoning that lead us to this implementation.

Specifically, 2.57SD is equivalent to a 99% interval. 39 is the smallest number of analogues found at any date. We therefore later choose this as the size of our fixed number reconstruction ensemble.

6. Page 5, Lines 22-23: why is it the only one? why 2105 is special? why not 1.5  $SD_{noi}$ ?

**Response:** Considering fixed  $SD_{noi}$  intervals, the number of valid analogues increases. It may become soon unfeasibly large. We think that the 2105 analogues for a  $1SD_{noi}$  interval are still reasonable. Therefore, we only consider a  $1SD_{noi}$  interval for the fixed SD reconstruction.

7. Overall, in the two paragraphs aforementioned, it lies the core of the two reconstructions carried out. I think this is important, and it should be made more explicit that the two approaches represent different method used for real below. Perhaps this can be made more explicit with some structural element, such as an un-ordered list or similar.

**Response:** Our revisions aim to clarify the methods section.

8. Page 6, Table 1: I assume this is exactly the correlation used to define the  $SD_{noi}$  in each proxy location, right? If so, this could be clarified in the main text (especially in section 2.1.2).

Response: Yes it is. We clarify this in the revised manuscript.

9. Page 6, lines 6–8: The criterion to exclude two proxies is not very clear. What is meant by "relevant portion of variance"? In Fig. 5 we learn that the reconstruction in these sites is poor. Would it be better if these sites were part of the network. Surely the answer is yes. I understand that the amount of climate information we get is poorer than in the other locations, but still we could benefit for having some information. At worst, if the proxies were pure noise, it would not be necessarily worse than not having information at all. In other words, I think having poor information is better than having none, and it's not fully obvious to me why proxies should be excluded from the analysis based on relatively low correlation alone.

**Response:** We clarify this in the revised manuscript.

Already Luterbacher et al. (2016) excluded these proxies because they lack a clear temperature signal.

The referee is correct that, generally, a pure noise record should not be worse than having no information at all. However, this is not the worst case. The worst case would be a record which biases the distance measure in our analogue search towards a different state.

That is, we follow the common approach to only include proxies with a signal beyond a certain level. We are aware that this has been a controversial decision in the past but we regard it valid in view of past practices.

10. Page 6 (but relevant for the whole study): why do you restrict the reconstruction to the period 1260 to 2003? The reconstruction could have been applied further back in time. The number of proxies varies in time, but this could be even beneficial for this study, focused on the validation of new methodologies. It would show how the estimates of the uncertainty presented here are sensible to a varying number of proxies. I feel that this choice has unnecessarily limited the scope of the manuscript.

**Response:** We thank the referee for their confidence in our approach. The referee is correct that in principle there are no reasons to stop in 1260. However, stopping there, in our opinion, eases the interpretation of results since thereby only an equal number of proxies enters the uncertainty estimation.

We clarify this in the revised manuscript.

11. Page 7, Table 2: it could be interesting to write the total number of analogues, i.e. the pool size. It would make more meaningful the number of proxies used to produce ensembles. For example, having 817 analogues (as in Fig. 8) has a clearer meaning when you add that they are 817 out of, let's say, 25000. It shows that you are still selecting a relatively minor number of relatively good analogues.

**Response:** We clarify this in the revised manuscript.

12. Page 7, Lines 6–7: I think having a consistent bias through the pool is not necessarily good, as it seems to be implied by the wording. It ensures that the bias are translated into the reconstruction. This is partly avoided using structurally different models to build the pool. I do not mean that the authors should necessarily rebuild the reconstruction with a larger set of models, but I think that at least they should not imply that using a single model is somehow beneficial.

**Response:** We make these modifications in the revised manuscript.

13. Page 8, Fig. 2: I think a line marking the 0 K anomalies would help to read the series. This pertains mostly panels b and c, where the sign of the anomaly is important, but difficult to appreciate without such a line. This argument applies to Figs 6 and 7 as well.

**Response:** We do not generally add zero lines to panels, because this increases the number of elements per panel and, thereby, possibly reduces their readability. We do add zero lines to panels of smoothed series (e.g., the mentioned panel 2b). The mentioned panel 2c was replaced by a different visualisation in a new Figure.

14. Page 8, Fig. 2: It's not fully clear to me what this figure (as well as Figs 6 and 7) show. Does "summary" mean spatial average?

Response: "Summary" means summary of the main results of the reconstruction. We modify the captions.

15. Page 9, Line 10: please change "degree Kelvin" to "Kelvin". Please review it, as there are other locations where I saw this in the manuscript.

**Response: We do so.**

16. Page 9, Lines 9–16. The order of these two paragraphs can be exchanged. It's a bit unusual and therefore confusing to discuss Fig. 2c before Fig. 2b.

**Response:** We restructure the section in our revisions.

17. Page 10, Lines 10–15: I think the fact that the reconstruction underestimate the intra-location variability is a problem of the pool, not the Analog Method itself. Do the authors think that this could be improved if higher resolution models were used to build the pool?

**Response:** We did not investigate this feature. We only state it without attributing it to the method. There are a number of explanations on which we only very shortly touch here. First, the noisy proxy series may overestimate the true intra-location variability. Second, the simulations may be too smooth in space. This, thirdly, might be due to the low resolution and simulations with higher resolutions might help then. Fourth, the chosen distance measure may result in such a feature dependent on the characteristics of the simulation pool, which however should usually not be the case.

We clarify our statement in the revised version.

18. Page 11, Fig. 3: The list of locations in the caption is misleading (the name and the ID are written all together). It seems a detail, but it puzzled me for a while until I realised that Tor92 and Torneträsk are not two proxies, but the ID and the name of the same one. You could easily remove this by using for instance parenthesis to separate name from ID or vice versa.

**Response:** We clarify the caption.

19. Page 11, Line 26: "The general agreement between the Euro 2k and the analogue..." this reads odd at this point, as the reader does not know where to find the information the authors are referring to. It turns out that this comparison is introduced later, in Figure 6 in Page 14.

**Response:** We try to clarify in our revisions what is meant at this point.

20. Page 15: Lines 17–23: The reduced variance could be quantified (how much is notably smaller variance in Line 19?). Further, the lost of variance when more analogues are considered is common in this approach, and generally in any statistical approach, i.e. there is a bias-variance trade off. It could be noted here that this has been comprehensively discussed in the bibliography of the Analog Method.

**Response:** We add more descriptions to highlight this point.

21. Page 15, Line 32: do the authors have a theory on what could be the reason for such systematic differences? Are they meaningful, can they be used to discuss merits or problems in the reconstructions? Or are they rather low-frequency random fluctuations highly sensitive to method parameters?

**Response:** In the case of the mid 16th century deviation there are indications that the Euro 2k more validly captures the extremes in this period (Wetter and Pfister, 2011, https://doi.org/10.5194/cp-7-1307-2011, 2013, https://doi.org/10.5194/ cp-9-41-2013), which may again indicate a period where the simulation pool is insufficient. Generally though, we would assume that it is mainly random due to the different sensitivities of the methods.

22. Page 16: Lines 25–26: The presence of missing values in years with volcanic eruptions is a major caveat of the method, as those are typically the years most interesting in climate studies. Here it would be specially relevant my comment about a comparison with the method presented by Neukom et al. (2019).

**Response:** The revisions emphasize this shortcoming of the approach.

23. Page 16, Lines 27–31: I do not see why it is "unsurprising" this lack of analogues for the recent period. The pool contains this warming as well, so the search should not present more problems for this period than in any other.

**Response:** The revisions make this point more clearly.

Indeed, the pool includes this period but we do not only require a similar mean state but also a similar interrelation between locations, which makes it more likely that the limited size of the pool does not include such a case.

24. Page 19, Lines 18–20: Maybe I'm miss-evaluating this, but I think that anchoring the reconstruction within a range of 8 K is a poor result. It shows that the 800 analogues are indeed poorly constrained in this region, so we have little idea of how the actual climate was in that period and region. More generally, I have the concern that the spread shown for example in Fig. 7 might provide an optimistic measure of the actual uncertainty. Fig 7e for instance shows the range in the spatial average, which is about 2 K. But this is after spatial average, where regional differences can cancel out! I wonder how large is the range in each location. This might perhaps be illustrated with a map of (temporally averaged) ranges? Eventually, my guess is that using as many as 800 analogues or more, really far away from "the best" is, as outlined by the authors, too much.

**Response:** We now show the mean and the maximum of the temporal temperature ranges in a Figure. We discuss this more explicitly in the revised manuscript.

[revised manuscript text omitted]

The analogue method found subsequent applications

not only in downscaling of climate information (e.g., Zorita and von Storch, 1999). In the paleoclimate-context, Graham et al. (2007) renan the method into Proxy Surrogate Reconstruction method and use the analogy between proxy-observations and simulated

- 15 climate states. Subsequently a number of authors use the approach for climate index and climate field reconstructions of past climate states (e.g., Franke et al., 2010; Trouet et al., 2009; Gómez-Navarro et al., 2015b, 2017; Jensen et al., 2018; Talento et al., 2019) in downscaling and upscaling of climate information (e.g., Zorita and von Storch, 1999; Schenk and Zorita, 2012). Modern analogue techniques of varying complexity are also common in paleoecology-paleoecology follow a similar idea (e.g., Graumlich, 1993; Jackson and Williams, 2004).
- 20 Our understanding of past climate changes depends on the consilience of our different avenues of evidence like simulations and reconstructions. The analogue method is a computationally cheap means to contrast information from both simulations and reconstructions in the sense of data assimilation though methodologically less sophisticated. The method The approach allows to reconcile the spatially sparse information from environmental and documentary proxy data with spatially complete and dynamically consistent though possibly biased information from observational data or long climate simulations
- 25 (Graham et al., 2007; Trouet et al., 2009; Guiot et al., 2010; Franke et al., 2010; Luterbacher et al., 2010; Schenk and Zorita, 2012; Góme This can provide a in the sense of data assimilation (Graham et al., 2007; Trouet et al., 2009; Guiot et al., 2010; Franke et al., 2010; Luterbacher et al., 2010; Schenk and Zorita, 2012; Góme It can provide an initial dynamic understanding of past climate variability in terms of a guesstimate. Gómez-Navarro et al. (2017) provide a short comparison with more complex data assimilation-techniques. Annan and Hargreaves (2012) test a particle-filter method
- 30 in a perfect model setting and. However, it is less sophisticated than full data assimilation procedures (compare, e.g. Tardif et al., 2019, and discussions in Gómez-Navarro et al., 2017). Graham et al. (2007) call reconstructions by analogue "Proxy Surrogate Reconstructions" in an early paleoclimatological application. Later studies use the approach for climate index and climate field reconstructions

(e.g., Franke et al., 2010; Trouet et al., 2009; Gómez-Navarro et al., 2015b, 2017; Jensen et al., 2018; Talento et al., 2019; Neukom et al.,

35

The analogue method is generally found to perform well, e.g., for area averaged indices and also at the locations of the used predictors (compare, e.g., Franke et al., 2010). However, reducing the number of predictors prominently worsens the skill at remote locations, and reconstruction skill accumulates at the predictor locations (Franke et al., 2010; Gómez-Navarro et al., 2015b). Annan and Hargreaves (2012) find a trade-off between accuracy and reliability of reconstructions dependent on quality and

5 quantity of the available proxy-records. Since simple analogue search approaches They test a particle-filter method. As simple analogue searches and particle filter methods share common assumptions, this trade-off also applies for analogue search reconstructions.

Franke et al. (2010) show the very good agreement of their proxy surrogate reconstruction in terms of the area averaged indices and also at the locations of instrumental data used as predictors. However, reducing the number of predictors prominently

- 10 worsens the skill at remote locations. Gómez-Navarro et al. (2015b) show further evidence for the accumulation of skill at the predictor locations (see also Annan and Hargreaves, 2012)
[revised manuscript text omitted]

- 10 observations, which measured archives, which in turn recorded the past environmental conditions (see, Evans et al., 2013). The observations rely on proxy data, which may be documentary notations but more often are measurements of biological, geological, or chemical properties of the archivesenvironment. Such proxy representations of the past conditions are naturally uncertain. The most obvious source of uncertainty is the sensitivity of the archives (e.g., trees) to that the archives recorded signals from more than one environmental condition
- 15 (e.g., Evans et al., 2013; Tolwinski-Ward et al., 2013; Evans et al., 2014; Tolwinski-Ward et al., 2015). climate or environmental

| Expression                          | Description                                   |
|-------------------------------------|-----------------------------------------------|
| $r_{\sim}$                          | Correlation coefficient                       |
| $\stackrel{R^2}{\sim}$              | Squared correlation coefficient               |
| MSEres                              | Residual Mean squared error                   |
| MSE tot                             | Total Mean squared error                      |
| Varres                              | Residual Variance                             |
| $\underbrace{Var_{tot}}{Var_{tot}}$ | Total Variance                                |
| $\underbrace{Var_{sig}}{Var_{sig}}$ | Variance of the signal                        |
| Varnoi            | Variance of the noise                         |
| Varnoi                              | Variance of the noise of an individual record |
| $\underbrace{Var_{sim}}$            | Variance of a simulation record               |
| $\underbrace{Var_{i}}_{i}$          | Variance of an indidividual time series       |
| $\widetilde{SD}$                    | Standard deviation                            |
| $\underline{SD}_{noi}$              | Standard deviation of the noise               |

variable (e.g. temperature and precipitation; compare Evans et al., 2013; Tolwinski-Ward et al., 2013; Evans et al., 2014; Tolwinski-Ward

In the following, we describe our thinking on the uncertainty of an analogue reconstruction. We first provide general derivations before describing the three reconstruction approaches (i) best analogue, (ii) fixed number of analogues, and (iii)

5 fixed uncertainty level. Our derivation of the uncertainty estimates relies on a number of assumptions, which we detail in the next paragraphs. Table 1 lists all mathematical expressions used in the following.

**Derivation of the uncertainty estimates**

Correlations provide a simple measure of the relation between proxy-observations and an environmental condition the climatic environment over a period when reliable (instrumental) observations of the environmental condition exist. From the correlation

- 10 coefficients, and under certain simplifying assumptions, climatic variability exist. We assume we can derive the uncertainty in representing the local climate by the of how well a local proxy record as described in the followingrepresents the local climate from the correlation coefficients. We denote this uncertainty hereafter as proxy uncertainty. We use correlations between the proxy records and the observational gridded CRU-data (Harris et al., 2014, Version CRU TS 3.10). Table 2 in section 2.2 lists the used proxies and their correlations to the observational data. These listed correlations enter our considerations on
- 15 uncertainty.

In our present approach, we consider normalized proxy data. That is the variance of an individual proxy *i* is  $Var_i = 1$ . We also consider normalized simulated records, and their local variance then also is  $Var_{sim} = 1$ . Our goal is to derive a simple criterion for the similarity between proxy patterns and simulated (analogue) patterns that takes into account the inherent uncertainty in the proxy records.

Assuming one can interpret the squared correlation coefficient ( $R^2$ ) as explained variance, one can profit from the equivalence  $R^2 = 1 - MSE_{res}/MSE_{tot} = 1 - Var_{res}/Var_{tot}$

**5 $R^2 = 1 - MSE_{res}/MSE_{tot} = 1 - Var_{res}/Var_{tot}$**

if we take the considered mean squared errors (MSE) as unbiased. The subscripts are res for residual and tot for total.

We can take the total variance  $Var_{tot}$  to be equal to the variance of the sum of a signal (subscript *sig*) and the residual noise. If we assume these are uncorrelated, we obtain  $1 - R^2 = Var_{noi}/(Var_{sig} + Var_{noi})$ .

**$1 - R^2 = Var_{noi} / (Var_{sig} + Var_{noi})$**

10 We replaced the residual variance by the noise variance (subscript *noi*) and reorganised the equation.

**If** Because we consider normalized data, the total variance becomes one,  $Var_{tot} = 1$ . For a simulated climate record in a grid-cell of a climate model, there is no uncertainty and, then, it is indeed  $Var_{tot} = Var_{sig} = 1$ , i.e. the total variance is pure signal. For the case of a normalized proxy we take  $Var_{tot} = 1 = Var_{sig} + Var_{noi}$  and thus  $\frac{1 - R^2 = Var_{noi}}{R^2 = Var_{noi}}$ .

In our present approach, we consider normalized proxy data, i.e.,  $Var_i = 1$  for an individual proxy *i*. We also consider 15 normalized simulated records, i.e.  $Var_{sim} = 1$ . Our goal is to replace a simple criterion of similarty between proxy patterns and simulated (analogue) patterns with a new criterion that also takes into account the inherent uncertainty in the proxy records . Candidate analogues then may provide a credible envelope on the analogue reconstruction dependent on the available data. With simulated

$$1 - R^2 = Var_{noi}$$

20 This is an expression for the noise variance of one local proxy record.

We want to use the local estimates of the proxy noise to formulate a criterion for finding analogues in simulated field records from climate simulations. Because we use simulated records with unit variance, the we can consider the following as a noise standard deviation becomes  $SD_{noi} = \sqrt{1 - R^2}$ .

 $\underbrace{SD_{noi}=\sqrt{1-R^2}}_{\underset{\scriptstyle \leftarrow}{\sum}}$

25 Based on these assumptions, there are a number of possible approaches to obtain uncertainties of ways to obtain uncertainty estimates for a reconstruction by analogue, which we describe next.

One possibility to define this modfied similarity criterion is to assume that

**Different reconstructions and uncertainty estimates**

First, we consider the case of a reconstruction from the single best analogue. We use the normalized data for this reconstruction.

30 For this case, we assume that we can obtain one standard deviation uncertainties as the square root of the sum over the

individual proxy noise variances  $(Var_{noi})$  divided by the number N of proxies:  $\sqrt{\sum_{1}^{N} (1 - Var_{noi})/N}$ . These are only an approximation of the uncertainty. If we want to plot the time-series in temperature units, we have to rescale these estimates. We do this simply by multiplying the noise variances in the square root by the grid-point variance from a selected simulation. Our visualisations for the single best analogue reconstruction add an alternative uncertainty envelope. This is given by the mean squared error between the proxy-values and the best-analogue values at the closest grid-point.

From our point of view, the real benefit of our derivation of uncertainty is to use only analogues which comply with a certain tolerance criterion. That is, a second way towards an uncertainty estimate assumes that we can obtain a similarity criterion between proxy data and simulation pool by considering the noise standard deviation represents a noise tolerance value for every proxy included in our analoguesearch for an individual proxy as local noise tolerance threshold. A candidate field has

5

10 to comply with all local thresholds to be considered a valid analogue. We then can limit our analogue search to only those analogues within a certain tolerance range at each location, i.e. within plus and minus one, two, or three  $SD_{noi}$  around the proxy value.

Alternatively, we can use the individual values for all proxies to construct a maximally tolerated Euclidean distance. The obvious caveat of this latter approach is that the analoguesmay locally lie outside the tolerance range of some of the proxy

- 15 records although the Euclidean distance is smaller than the maximally tolerated value. On the other hand, the criterion that the analogue should lie within each individual proxy tolerance may exclude the overall best analogue according to the minimal Euclidean distance. We consider this downside acceptableIn the following we only consider analogues within traditional 90%, 95%, 99% and 99.9% intervals. We consider two cases: (A) we use a fixed number of analogues, and (B) we use a fixed noise level *SDnoi*. For the fixed number approach, we ad hoc require that there are at least ten valid analogues for all years.
- 20 GenerallyFor a defined noise tolerance criterion, there may be at best a few locally tolerable analogues for a certain dateaccording to a defined tolerance criterion. We find for our application that a . For example, if we consider a criterion of one  $SD_{noi}$  tolerance provides no tolerable analogue, that is a ~68%-interval, this criterion is so strict that we do not find any tolerable analogues for 35 dates. Similarly 1.64  $SD_{noi}$  and 1.96  $SD_{noi}$  years in our period of interest. Similarly ~1.64  $SD_{noi}$  (90%) and ~1.96  $SD_{noi}$  (95%) criteria still imply that we find less than ten analogues for one year (2003 CE).
- 25 Obviously, the real benefit of the proposed method is to use only analogues, which comply with a certain tolerance criterion. In the following, we choose a tolerance criterion of  $2.57 SD_{noi}$  However, we want to provide a reconstruction at each date for the full period. We restrict the number of analogues for all dates to a constant number, which in the period 1260 to 2003 CE and want to consider a fixed number of analogues. We find that among the tested levels, a tolerance criterion of  $2.57 SD_{noi}$ , i.e. a 99% interval, is the smallest number of available analogues at any date within noise level that provides more than 10
- 30 analogues for every year in the full period. If we include the year 2003, the The minimal number of analogues is 39.39 for this criterion if we include the year 2003. It increases to 156 excluding the year 2003. We do not test additional noise levels between  $\sim 1.96 SD_{noi}$  and  $2.57 SD_{noi}$  as we further, ad hoc, decide that 39 analogues is still a reasonably small number of analogues for the reconstruction with a constant number of analogues. Thus, our reconstruction with a constant number of analogues uses 39 analogues.

However, the Considering a fixed standard-deviation criterion, the number of valid analogues can become large for individual years. For example, the largest number of analogues for a single year for a one-standard deviation criterion is the only one that gives is 2105 in our approach. We regard this still a subjectively reasonable maximal number of 2105 possible analogues. Thus, subsequently, we also we choose a one  $SD_{noi}$  interval to discuss results for a fixed one  $SD_{noi}$  interval. Both sets of results

5 are also compared to a criterion. As the previous paragraphs highlight, such a  $1SD_{noi}$  criterion will fail to find analogues for certain years.

We later show the results for these reconstructions in comparison to the single best-analogue reconstruction. For ensembles of analogues, uncertainty estimates are the full range of the ensemble and an uncertainty envelope based on the intra-ensemble variance.

- 10 Our time-series plots present a number of uncertainty envelopes. The first one is motivated by the considerations detailed above. If we show normalized series, we assume that the square root of the sum over the individual proxy noise variances  $(Var_{not_i})$  divided by the number of proxies represent one standard deviation uncertainties. However, for plotting temperature series, we have to rescale these estimates. We do this simply by multiplying the noise variances in the square root by a selected grid-point varianceAs a side note, we could also use the individual local values for all proxies to construct a maximally tolerated
- 15 Euclidean distance. The obvious caveat of this approach is that the analogues may locally lie outside the tolerance range of some of the proxy records although the Euclidean distance is smaller than the maximally tolerated value. On the other hand, the criterion that the analogue should lie within each individual proxy tolerance may exclude the overall best analogue according to the minimal Euclidean distance. We consider this downside acceptable and only consider these. Furthermore, we do not weight the analogues, e.g., according to their distance, because our approach of explicitly considering the uncertainty in the

20 proxies already accounts for the mismatch between proxies and candidate pool. Additionally, for ensembles of analogues, the full range of the ensemble is plotted, and another envelope bases on the intra-ensemble variance. Finally, for single best-analogue reconstructions, a credible envelope is given by the MSE between the normalized proxy-values and the normalized 
[revised manuscript text omitted]

We use data centered on the full period 1260 to 2003 CE and the data is normalized with the standard deviation over the same period. Jungclaus et al. (2010) provide details on the simulations (see also data references in Table 23). We use simulation output from the ensemble members including all forcing components for the period 800 to 2005 CE (Table 2). 3). Thereby we

10 have a pool of 9648 candidate fields. Forcings are solar, volcanic, greenhouse gas, orbital, and land use; the carbon dioxide concentration was calculated interactively (compare Jungclaus et al., 2010).

**3** Results**

**3.1 Single best-analogue reconstruction**

Figure ?? summarises Figures 2 and 3 compare the single best-analogue reconstruction to the Euro 2k-reconstruction and

- 15 the observational data relative to the full period 1260 to 2003 CE. There is generally very good agreement between the Euro 2k-2k-reconstruction and the analogue reconstruction but the latter appears to overestimate the warming since the early 19th century (Figure 2a). Note that the observational data is plotted relative to the mean of the Euro 2k-reconstruction over the observational period and solely provides a qualitative comparison. We evaluate our analogue reconstruction against the Euro 2k reconstruction, because we regard the former reconstruction as the main benchmark for the analogue uncertainty estimation.
- 20 Appendix Figure A1 makes the comparison relative to the period of the observational data. We note that differences between the observations and the reconstructions are larger for the best analogue approach compared to the Euro-2k reconstruction (Figure 3a and Appendix Figure A2).

The analogue reconstruction shows rather small centennial variations as does the Euro 2k-reconstruction (Figure 2). We note that the Bayesian Hierarchichal Modelling (BHM) reconstruction by Luterbacher et al. (2016) shows larger variations

- 25 compared to their composite-plus-scaling reconstruction in the early part of the last millennium prior to our study period. The larger warming since about 1800 in the analogue reconstruction is in line with a slightly larger warming in the BHMreconstruction by Luterbacher et al. (2016). Appendix Figure A3 shows a comparison of the best analogues reconstruction to the two European summer temperature reconstructions of Luterbacher et al. (2016). This complements Figure 2 where we show the comparison to the Euro 2k-reconstruction of PAGES 2k Consortium (2013).
- 30 The difference plot in Figure ??e shows the size of the interannual Figure 3a shows differences between different data sets as swarm plots. Swarm plots are categorical scatter plots, where the data points are adjusted to avoid overlap between points. Thereby, swarm plots provide information on the distribution of the data plotted. The differences between the Euro 2k composite-plus-scaling reconstruction and the best-analogue reconstruction highlight again their reasonable agreement (Figure

---

## Author Response (AR2)

Dear editor, dear referees,

many thanks for your positive evaluation of our revised manuscript.

Below we address the referee's comments.

On behalf of the authors

Yours sincerely

Oliver Bothe

**Referee 2:**

**General Comment**

*The manuscript addresses the important issue of the uncertainty estimation in analog-based climate reconstructions, a frequently used reconstruction methods in the recent past. Overall, the paper is well written and improved considerably from the original discussion paper. However, I would prefer a more condensed results section, which is still very extensive. It would be sufficient to focus more on common features and differences by comparing the approaches instead of explaining the same information for each approach repeatedly, e.g. naming the coldest, warmest decades for each approach again. Furthermore, the readability of some figures needs improvements (see below). Overall, I suggest publication after some minor corrections.*

Response: We thank the referee for these comments. We tried to be more concise in our results section. Therefore we introduce a new subsection for the characteristics of all three reconstructions. However, we failed to condense the section notably.

We address the readability of Figures below.

*Page 4, line 15: Did you check if the standard deviations of the model simulation agree with observational data sets?*

Response: There is no possibility to check the full period standard deviations against an observational dataset. Over the period of overlap from 1901 to 2003, the modelled standard deviation are more often larger than the observational standard deviation than they are smaller. The ratio between modelled and observed standard deviations are between 0.75 to 1.25 in approximately 65% of the grid cells of the observational data.

*Page 5, line 6: I would not claim that mixed signals are the "most obvious" source of uncertainty, maybe better "one possible".*

Response: We change this to "A prominent source".

*Page 5, line 11: Not clear why MSE is mentioned here, too and not the only the standard variance definition.*

Response: We thought that the step using the MSE helps to motivate why the variance relation holds. We remove it on suggestion of the reviewer.

*Page 9, line 6: Can you explain how the Albanian and Slovakian proxy records can be in the PAGES2k data set if they do not explain temperature variability? Additionally, if they do not contain a climate signal, I do not understand why you show them later in the paper as kind of independent validation/comparison?*

Response: We cannot fully recall why the Pages 2k consortium and the EuroMed 2k consortium included these two proxies. However, we note that Büntgen et al. (https://doi.org/10.1073/pnas.1211485110) describe the Slovakian proxy as being sensitive to May-June temperature, which may have influenced the decision by the mentioned consortia. Similarly, Seim et al. (https://doi.org/10.3354/cr01076) find some weakly significant negative relations to temperature in their chronology.

We kept these records for comparisons as we also compare to the PAGES2k European reconstruction.

We have now removed these comparisons from the manuscript.

*Figure 2: very hard to read the figure, e.g. I cannot really see the grey envelope. Please consider other smoothing or colors.*

Response: We have clarified the figure.

*Page 12, line 24: This was already explained in line 9 on page 11.*

Response: We are sorry, but we cannot follow this comment by the referee. P12L24 and P9L11 do not refer to the same features.

*Page 13, line 10-19: This paragraph belongs in the methods section.*

Response: This was already described in the methods section. We reduce the redundancy but keep some of this information here, as it is relevant for the comparison at hand.

*Figure 4, its caption and Fig. 5: Labels not consistent, sometimes 3-letter abbreviations with numbers and sometimes without numbers.*

Response: We changed these labels to be consistent.

*Figure 4 and page 16, line 10: Why are proxies without a climate signal included (see above)?*

Response: We kept these records for comparisons as we also compare to the PAGES2k European reconstruction.

We now removed these comparisons.

*Page 16, line 11: This convergence rather gives us confidence in the method than in the understanding of past climate. The selection of input data and the multi-decadal to centennial variability contained in these proxies, probably influences on the reconstructions, too.*

Response: We agree and add a note on the confidence in the method.

We agree with the referee that the selection of input data and their variability influence the reconstruction. We do not add any discussions on this point.

*Figure 6: In my opinion, this figure is not really needed. Just reporting correlation coefficients would be sufficient.*

Response: We understand the referees concern but choose to keep the Figure.

Our aim here is to highlight how well constrained the local data is and how wide the range is for the Central European mean.

*Figure 7: The light grey ensemble range is not visible at all.*

Response: We replace the grey shading by grey lines to clarify the Figure.

*Figure 11: Why is the experiment with the fixed number of 39 analogs not included for comparison?*

Response: We now add this data to panel (b).

*Page 25, line 4: typo in "available"*

Response: We thank the referee for spotting this. We correct this.

*Page 27, line 19: Confusing that reconstructions are not made for the European domain only (but global?) and are now compared to Nuuk. This has never be mentioned and discussed before.*

Response: The referee is obviously correct. This is an artefact of earlier tests with a larger domain that we surprisingly never caught. Since this was already in the initial submission, we keep the panel in the three comparison Figures but modify the text to read:

Revision-1-Version: Comparing the data series, however, indicates notable shortcomings of the reconstruction median. The reconstruction median often overestimates the recent warming trend and the median shows notably less variability than the BEST-series. The underestimation of the variability on the other hand leads occasionally to an underestimation of the most recent warm anomalies. The top-left panel for Nuuk highlights that the lack of constraints on the reconstruction can result in potentially artificial spikes in the time-series.

New-Revision-2-Version: Comparing the data series, however, indicates notable shortcomings of the reconstruction median. The reconstruction median often overestimates the recent warming trend and the median shows notably less variability than the BEST-series. The underestimation of the variability on the other hand leads occasionally to an underestimation of the most recent warm anomalies. There are also cases where both series appear to agree quite well over the period when both are available. Examples are the Central England Temperature and Montdidier. The top-left panel extends our comparison longitudinally beyond our European reconstruction domain. The data for Nuuk on Greenland highlights that the lack of constraints outside of the reconstruction domain can result in potentially artificial spikes in the time-series.

**List of changes in manuscript:**

Abstract:

Introduction:

- minor edits

Methods & Data:

- changed derivation of uncertainties
- edited description of uncertainties
- added information on PAGES2k uncertainties
- minor edits

Results:

- removed comparison to excluded data
- clarified a number of Figures including increases of their panel sizes, changed colors, and added or removed elements
- introduced additional subsection on characteristics of the reconstructions
- minor edits

Summary and Discussions:

Concluding remarks:

Appendix:

[revised manuscript text omitted]